# Complex-valued Neurons Can Learn More but Slower than Real-valued Neurons via Gradient Descent

**Jin-Hui Wu, Shao-Qun Zhang, Yuan Jiang, Zhi-Hua Zhou**
National Key Laboratory for Novel Software Technology, Nanjing University, China
School of Artificial Intelligence, Nanjing University, China
{wujh,zhangsq,jiangy,zhouzh}@lamda.nju.edu.cn

## Abstract

Complex-valued neural networks potentially possess better representations and performance than real-valued counterparts when dealing with some complicated tasks such as acoustic analysis, radar image classification, etc. Despite empirical successes, it remains unknown theoretically when and to what extent complex-valued neural networks outperform real-valued ones. We take one step in this direction by comparing the learnability of real-valued neurons and complex-valued neurons via gradient descent. We show that a complex-valued neuron can efficiently learn functions expressed by any one real-valued neuron and any one complex-valued neuron with convergence rate $O(t^{-3})$ and $O(t^{-1})$ where $t$ is the iteration index of gradient descent, respectively, whereas a two-layer real-valued neural network with finite width cannot learn a single non-degenerate complex-valued neuron. We prove that a complex-valued neuron learns a real-valued neuron with rate $\Omega(t^{-3})$, exponentially slower than the $O(e^{-ct})$ rate of learning one real-valued neuron using a real-valued neuron with a constant $c$. We further verify and extend these results via simulation experiments in more general settings.

## 1 Introduction

Complex-valued neural networks (CVNNs) utilize neuron models and operations in the complex-valued domain and are good at handling many complicated scenarios. Pioneering works successfully apply CVNNs to various areas, such as synthetic aperture radar image classification [1], acoustic analysis [2], and magnetic resonance image reconstruction [3]. In these applications, input signals naturally contain phase information. CVNNs seem more suitable than real-valued neural networks (RVNNs) in phase-dependent tasks since empirical experiments and intuitive explanations suggest that CVNNs can possess better data representations of phase information and grasp the phase-rotational dynamics more accurately [4, 5].

Beyond the seemingly promising performance of CVNNs, many efforts have been devoted to the theoretical understanding of CVNNs. Most existing works demonstrate some desirable properties of CVNNs such as universal approximation [6, 7], the minimum width for universal approximation [8], boundedness and complete stability [9], most critical points not being local minimum [10], and local-minimum-free conditions [11]. Some recent works demonstrate the approximation advantage of CVNNs in phase-invariant tasks by proving that neuromorphic networks with complex-valued operations can approximate radial functions with exponentially fewer parameters than RVNNs [11, 12]. However, those studies do not explain why CVNNs outperform RVNNs mostly in phase-dependent tasks, particularly when considering the fact that there are functions that can be efficiently approximated but not efficiently learned with gradient methods [13, 14]. Indeed, the general functional difference between CVNNs and RVNNS remains unknown.

37th Conference on Neural Information Processing Systems (NeurIPS 2023).

In this paper, we take one step towards understanding when and to what extent one can benefit from common learning paradigms using CVNNs rather than RVNNs. More specifically, we attempt to identify the superiority and inferiority of CVNNs through the following fundamental questions

*When CVNNs outperform RVNNs via gradient descent?*

*Can we learn everything with CVNNs without paying additional price?*

We theoretically study the above two questions by focusing on learning a single neuron by optimizing the expected square loss via gradient descent under the setting of low-dimensional inputs and no bias term. Learning a single neuron, a simple and widely investigated learning problem [15, 16, 17, 18], is helpful to understand the difference between learning RVNNs and learning CVNNs since the neural operations inside a fundamental neuron model include the key factors of neural network learning. Furthermore, we conduct simulation experiments to verify our theories and extend them to more general settings of high-dimensional inputs and with bias terms. Our contributions are summarized in Table 1 and further explained as follows.

- **Complex-valued neurons can learn more than real-valued neurons.** We prove that using gradient descent, a single complex-valued neuron can efficiently learn functions expressed by any one real-valued neuron and any one complex-valued neuron with convergence rate $O(t^{-3})$ and $O(t^{-1})$ in Theorems 1 and 2, respectively. In contrast, we show the lower bound of expressing a non-degenerate complex-valued neuron with a two-layer RVNN in Theorem 4, which implies that a two-layer RVNN with finite width cannot learn a single non-degenerate complex-valued neuron. These results provide positive responses to the first question from at least two perspectives. Firstly, CVNNs outperform RVNNs when dealing with phase-sensitive tasks. Secondly, CVNN is always a conservative choice when we are unwilling to take the risk of failure.

- **Complex-valued neurons learn slower than real-valued neurons.** We present a lower bound $\Omega(t^{-3})$ for learning functions expressed by any one real-valued neuron using a complex-valued neuron via gradient descent in Theorem 6. This conclusion, together with the well-known linear convergence of learning functions expressed by any one real-valued neuron using a real-valued neuron [19], implies that CVNNs suffer from slower convergence than RVNNs when handling simple phase-independent tasks. This phenomenon answers the second question and reveals the additional price for learning everything with CVNNs.

Table 1: A summary of our contributions. The first column lists the target neurons. The second and third columns represent the convergence rates of learning the target neurons using real-valued neurons and complex-valued neurons via gradient descent, respectively.

| Target Neurons | Real-valued Neurons | Complex-valued Neurons |
|---|---|---|
| Real-valued Neuron | $O(\mathrm{e}^{-ct})$ [19] | $\Theta(t^{-3})$ (Theorems 1 and 6) |
| Complex-valued Neuron | Cannot Learn (Theorem 4) | $O(t^{-1})$ (Theorem 2) |

The rest of this paper is organized as follows. Section 2 introduces related works. Section 3 details our settings and notations. Section 4 demonstrates that complex-valued neurons can learn more than real-valued neurons. Section 5 proves that complex-valued neurons learn slower than real-valued neurons. Section 6 concludes our work with prospects.

## 2 Related Works

**Complex-valued Neural Networks.** CVNNs originate in the 1990s when parameters of networks and the commonly used back-propagation algorithm are generalized to the complex-valued domain [20, 21, 22]. The motivation of CVNNs is at least threefold. From the representation perspective, CVNNs consider the phase information and model complex-valued problems more efficiently and properly than RVNNs [23, 24, 25]. From the computation perspective, a complex-valued neuron is capable of solving the exclusive-or problem and the detection of symmetry, whereas a real-valued neuron cannot [26]. From the biological perspective, the recently proposed flexible transmitter neuron [27], which has a natural complex implementation, formulates the communication between pre-synapse and post-synapse precisely rather than considering only the pre-synapse in traditional MP neuron [28]. CVNNs achieve better performance in versatile applications, especially those with naturally phase-related signals, such as radio frequency signals [29], sonar signals [30], and audio signals [31]. We refer to two surveys for more detailed discussions [4, 5].

From the aspect of theories, several works provide preliminary support for CVNNs by proving fundamental properties of CVNNs, such as shallow CVNNs are universal approximators [6, 7], most critical points are not spurious local minimum [10, 11], and CVNNs are bounded and completely stable [9]. These theoretical insights only consider CVNNs without comparison with RVNNs. Another line of research verifies the superiority of CVNNs by comparing the approximation complexity of RVNNs and CVNNs and finding that CVNNs can express radial functions more efficiently [11, 12]. This line of work only takes approximation into account and does not explicitly consider learning processes, which is of more interest in practice. This work takes the first step toward analyzing and comparing the learning behaviors of CVNNs and RVNNs.

**Neuron Learning.** Neuron learning is the simplest case of neural network learning, and existing works mainly focus on learning real-valued neurons. Some studies demonstrate the possibility of learning one real-valued neuron or a network using meticulously designed algorithms [32, 33, 34]. Later, researchers investigate the learnability of neurons using standard gradient methods. An exponential convergence rate is established for learning one real-valued neuron with a real-valued neuron under different assumptions [19, 35, 36, 37, 38]. We consider the problem of learning between one real-valued neuron and one complex-valued neuron, as well as learning one complex-valued neuron using a complex-valued neuron. The heterogeneity between real-valued and complex-valued neurons makes the analysis of optimization behaviors more complicated.

# 3 Preliminaries

**Notations.** Suppose that the input dimension is an even number. For any vector $\boldsymbol{x} \in \mathbb{R}^{2d}$, we denote $x_i$ as the $i$-th coordinate of $\boldsymbol{x}$. Let $\boldsymbol{x}_{\mathbb{C}} = (x_1; \ldots; x_d) + (x_{d+1}; \ldots; x_{2d})\mathrm{i} \in \mathbb{C}^d$ be the folded complex-valued representation of $\boldsymbol{x}$, and $\overline{\boldsymbol{x}}_{\mathbb{C}} = (x_1; \ldots; x_d) - (x_{d+1}; \ldots; x_{2d})\mathrm{i}$ is the complex conjugate of $\boldsymbol{x}_{\mathbb{C}}$. For any two vectors $\boldsymbol{w}, \boldsymbol{v} \in \mathbb{R}^{2d}$, $\theta_{\boldsymbol{w},\boldsymbol{v}} = \arccos(\boldsymbol{w}^\top \boldsymbol{v} \|\boldsymbol{w}\|^{-1}\|\boldsymbol{v}\|^{-1}) \in [0, \pi]$ denotes the angle between $\boldsymbol{w}$ and $\boldsymbol{v}$. For any $x \in \mathbb{R}$, $\tau(x) = \max\{0, x\}$ indicates the ReLU activation function. Let $\mathrm{Re}(z)$ denote the real part of a complex number $z$. For any $z \in \mathbb{C}$ and $\psi \in [0, \pi/2]$, $\sigma_\psi(z)$ denotes the real part of the symmetrical version of zReLU activation function [39], i.e.,

$$\sigma_\psi(z) = \left\{ \begin{array}{ll} \mathrm{Re}(z), & \theta_z \in [-\psi, \psi], \\ 0, & \text{otherwise}, \end{array} \right.$$

where $\theta_z$ represents the argument of $z$. For any proposition $p$, we use $\mathbb{I}(p)$ to represent the indicator function of $p$, i.e., $\mathbb{I}(p) = 1$ if $p$ is true and $\mathbb{I}(p) = 0$ otherwise. A table of frequently used notations is provided at the beginning of Appendix A.

**Learning a Single Neuron.** We consider learning a target neuron with a learning neuron. A neuron generally takes the form $\boldsymbol{x} \to \sigma_\psi(\boldsymbol{w}; \boldsymbol{x})$, where the weight $\boldsymbol{w} \in \mathbb{R}^{2d}$ and phase $\psi \in [0, \pi/2]$ indicate learnable parameters, and we omit the bias term for technical reasons. This general formulation includes a real-valued neuron with ReLU activation $\boldsymbol{x} \to \tau(\boldsymbol{w}^\top \boldsymbol{x})$ and a complex-valued neuron with zReLU activation $\boldsymbol{x} \to \sigma_\psi(\boldsymbol{w}_{\mathbb{C}}^\top \overline{\boldsymbol{x}}_{\mathbb{C}})$ as special cases. For any target neuron with parameters $(\boldsymbol{v}, \psi_v)$, the learning process consists of finding a neuron with parameters $(\boldsymbol{w}, \psi_w)$ to minimize the expected square loss

$$L(\boldsymbol{w}, \psi_w) = \frac{1}{2}\mathbb{E}_{\boldsymbol{x} \sim \mathcal{N}(\mathbf{0},\mathbf{I})} \left[ (\sigma_{\psi_w}(\boldsymbol{w}; \boldsymbol{x}) - \sigma_{\psi_v}(\boldsymbol{v}; \boldsymbol{x}))^2 \right], \tag{1}$$

where the learnable parameter $\psi_w$ occurs only when the learning neuron is complex-valued, and the input $\boldsymbol{x}$ follows the Gaussian distribution $\mathcal{N}(\mathbf{0}, \mathbf{I})$.

**Learning Algorithm and Initialization.** We utilize the projected gradient descent as the learning algorithm, where the projection guarantees the constraint on phase $\psi \in [0, \pi/2]$. To minimize a function $f(\boldsymbol{x})$ with an initialization $\boldsymbol{x}_0$, projected gradient descent iteratively updates weights along the negative gradient direction and projects the updated weights onto the constraint set, i.e., $\boldsymbol{x}_{t+1} = P_{\mathcal{Q}}(\boldsymbol{x}_t - \eta_t \nabla_{\boldsymbol{x}} f(\boldsymbol{x}_t))$, where $\eta_t$ represents the step size, $\mathcal{Q}$ denotes the constraint set, and $P_{\mathcal{Q}}$ indicates the projection operator defined by $P_{\mathcal{Q}}(\boldsymbol{x}_0) = \arg\min_{\boldsymbol{x} \in \mathcal{Q}} \|\boldsymbol{x} - \boldsymbol{x}_0\|$. We initialize weights of neurons with Gaussian distribution, which includes most standard initialization schemes in practice [40]. The learnable parameter of the zReLU activation is initialized with $\mathcal{U}(0, \pi/2)$, i.e., the uniform distribution on $[0, \pi/2]$.

# 4 Complex-valued Neurons Can Learn More

In this section, we provide theoretical support for the learning advantage of complex-valued neurons by providing two positive learning scenarios for complex-valued neurons and one negative learning result for real-valued neurons. This section is organized as follows. Subsections 4.1 and 4.2 confirm the learning power of complex-valued neurons, by verifying that a complex-valued neuron can efficiently learn functions expressed by any one real-valued neuron and any one complex-valued neuron, respectively. Subsection 4.3 points out the limited learning capability of real-valued neurons, by proving that a two-layer RVNN with finite width cannot learn a single non-degenerate complex-valued neuron.

## 4.1 Learning One Real-valued Neuron with a Complex-valued Neuron

We first investigate the case of learning one real-valued neuron with ReLU activation using a complex-valued neuron with zReLU activation, where the expected square loss in Eq. (1) becomes

$$L_{\mathrm{cr}}(\boldsymbol{w}, \psi) = \frac{1}{2}\mathbb{E}_{\boldsymbol{x} \sim \mathcal{N}(\mathbf{0}, \mathbf{I})}\left[\left(\sigma_\psi(\boldsymbol{w}_{\mathbb{C}}^\top \overline{\boldsymbol{x}}_{\mathbb{C}}) - \tau(\boldsymbol{v}^\top \boldsymbol{x})\right)^2\right], \tag{2}$$

where we abbreviate the phase parameter $\psi_w$ as $\psi$ since the target real-valued neuron does not have a phase parameter, $\boldsymbol{w} \in \mathbb{R}^{2d}$ and $\boldsymbol{v} \in \mathbb{R}^{2d}$ represent the weight vectors of the complex-valued neuron and the real-valued neuron, respectively. We assume $\|\boldsymbol{v}\| = 1$ without loss of generality. Then we present the first theorem for complex-valued neuron learning.

**Theorem 1.** *Let $d = 1$. Suppose that $\boldsymbol{w}_0 \sim \mathcal{N}(0, I_2)$ and $\psi_0 \sim \mathcal{U}(0, \pi/2)$. Let $\{(\boldsymbol{w}_t, \psi_t)\}_{t=0}^\infty$ denote the parameter sequence of the complex-valued neuron generated by projected gradient descent when optimizing $L_{\mathrm{cr}}$, the expected loss of learning a real-valued neuron using a complex-valued neuron. If the step size $\eta_t = \eta \in (0, 1/(12\pi))$, then we have*

$$\Pr\left[L_{\mathrm{cr}}(\boldsymbol{w}_t, \psi_t) \leqslant \frac{8000}{\eta^3 t^3} + \left(1 - \frac{\eta}{48}\right)^{t+1-32/\eta}\right] \geqslant \frac{1}{32}.$$

Theorem 1 shows that a complex-valued neuron can efficiently learn the functions expressed by any one real-valued neuron with convergence rate $O(t^{-3})$ using projected gradient descent. It should be mentioned that we do not attempt to decrease the large constants in the theorem, as they do not hurt the constant probability and convergence rate.

The constant probability, rather than high probability, comes from the intrinsical difference between real-valued neurons and complex-valued neurons. A real-valued neuron activates half of the phase domain, whereas a complex-valued neuron may only activate a small part as controlled by the parameter $\psi$, which makes the expected loss a piecewise function. When the initialization of $\boldsymbol{w}$ falls into the opposite direction of $\boldsymbol{v}$ and $\psi$ is small, the activated regions of the real-valued and complex-valued neurons are not overlapped. Such a bad initialization happens with a constant probability and encourages the complex-valued neuron to decrease phase to minimize the loss. As a result, the phase of the complex-valued neuron will shrink to zero, which leads to a constant expected square loss and the failure of learning.

**Challenges.** Although $(\boldsymbol{w}, \psi) = (\boldsymbol{v}, \pi/2)$ is an obvious global minimum of the expected loss with $L_{\mathrm{cr}} = 0$, the convergence conclusion in Theorem 1 is non-trivial. As one can see in the proof, the landscape of the expected loss possesses a stationary point $(\boldsymbol{w}, \psi) = \mathbf{0}$. If we initialize $\boldsymbol{w} = -k\boldsymbol{v}$ with $k > 0$, then it is easy to verify that $\boldsymbol{w}$ converges to $\mathbf{0}$ and $\psi$ decreases to $0$ when the step size is sufficiently small. This implies that the landscape is not convex and the spurious stationary point is an attractor. The existence of this spurious stationary point becomes a critical obstacle in the proof and provides another reason for the hardness of a high-probability conclusion.

The proof idea of Theorem 1 mainly consists of estimating the first-order derivatives and finding an ideal region with both constant probability and convergence guarantees. We provide a proof sketch as follows. Firstly, we analyze the optimization behaviors of $\boldsymbol{w}$ and $\psi$ in all pieces of the loss function separately. Then we identify an ideal region with desirable gradient properties: the gradient $\nabla_\psi L_{\mathrm{cr}}(\boldsymbol{w}, \psi)$ can be bounded by $O((\psi - \pi/2)^2)$, which implies that $\psi - \pi/2$ decreases with an inversely propositional rate. Meanwhile, gradient descent on $\boldsymbol{w}$ performs like a contraction mapping with fixed point $\boldsymbol{v}$ and Lipschitz constant $1 - \Theta(\psi)$, i.e., $\boldsymbol{w}$ converges to $\boldsymbol{v}$ linearly when $\psi$

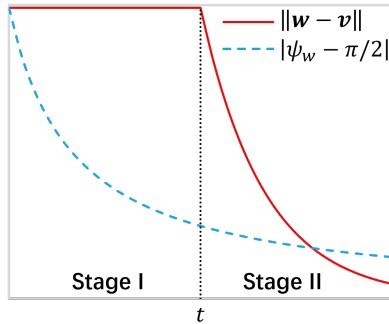 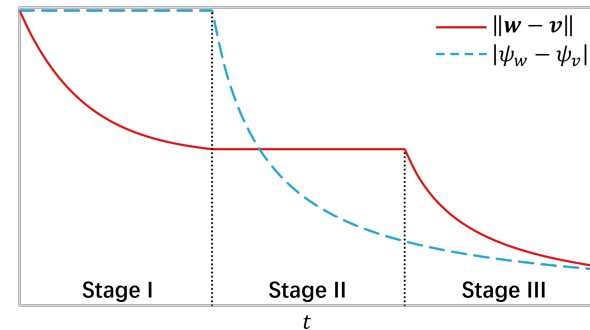

(a) Convergence stages of Theorem 1.  (b) Convergence stages of Theorem 2.

Figure 1: Subfigures (a) and (b) demonstrate the convergence stages of Theorems 1 and 2, respectively. The horizontal axis represents the iteration index of gradient descent. The black dotted line denotes the separation of convergence stages.

is large enough. Based on these observations, our convergence analysis consists of two stages, as shown in Fig. 1(a). In Stage I, the phase $\psi$ converges towards the global minimum, and the weight $\boldsymbol{w}$ remains in the ideal region. When the phase grows above some threshold, one enters Stage II where the weight converges linearly and the phase maintains its slow convergence rate. Finally, we estimate the order of loss and provide a lower bound of the probability of falling into the ideal region with Gaussian initialization to complete the proof. Detailed proofs are available in Appendix B.

## 4.2 Learning One Complex-valued Neuron with a Complex-valued Neuron

We proceed to consider learning one complex-valued neuron using a complex-valued neuron. In this case, the expected square loss in Eq. (1) can be rewritten as

$$L_{\mathrm{cc}}(\boldsymbol{w}, \psi) = \frac{1}{2}\mathbb{E}_{\boldsymbol{x}\sim\mathcal{N}(\mathbf{0},\mathbf{I})}\left[\left(\sigma_\psi(\boldsymbol{w}_\mathbb{C}^\top\overline{\boldsymbol{x}}_\mathbb{C}) - \sigma_\phi(\boldsymbol{v}_\mathbb{C}^\top\overline{\boldsymbol{x}}_\mathbb{C})\right)^2\right],$$

where $(\boldsymbol{v}, \phi)$ denotes the parameter of the target complex-valued neuron, and $(\boldsymbol{w}, \psi)$ is the learnable parameter. Without loss of generality, we still assume $\|\boldsymbol{v}\| = 1$. Here, we use gradient descent with vanishing step size $x_{t+1} = x_t - \eta_t\nabla f(x_t)$, where the positive step size $\eta_t$ satisfies $\eta_t \to 0$ as $t \to \infty$. Then we present the second theorem for complex-valued neuron learning.

**Theorem 2.** *Let $d = 1$, and $\psi_v \in [7\pi/20, 2\pi/5]$. Suppose that $\boldsymbol{w}_0 \sim \mathcal{N}(\mathbf{0}, I_2)$ and $\psi_{w,0} \sim \mathcal{U}(0, \pi/2)$. Let $\{(\boldsymbol{w}_t, \psi_{w,t})\}_{t=0}^\infty$ denote the parameter sequence of the complex-valued neuron generated by projected gradient descent when optimizing $L_{\mathrm{cc}}$, the expected loss of learning a complex-valued neuron using a complex-valued neuron. If we utilize vanishing step size $\eta_t = \min\{c_1, c_2/t\}$ with $c_1 \leqslant 1/3000$ and $c_2 \geqslant 20$, then*

$$\Pr\left[L_{\mathrm{cc}}(\boldsymbol{w}_t, \psi_{w,t}) \leqslant \frac{400c_2^3}{c_1 t}\right] \geqslant 10^{-5}.$$

Theorem 2 demonstrates that a complex-valued neuron can efficiently learn functions expressed by any one complex-valued neuron with convergence rate $O(t^{-1})$ and constant probability.

**Challenges.** It is observed that the $O(t^{-1})$ convergence rate in Theorem 2 is slower than the $O(t^{-3})$ convergence rate in Theorem 1. The deceleration of convergence comes from the intrinsic difficulties of learning functions expressed by any one complex-valued neuron. These difficulties become the main challenges in the analysis and can be understood from at least two perspectives. Firstly, there emerge new spurious stationary points. As one can see in the proof, the gradient with respect to $\psi_w$ becomes 0 once $\psi_w$ reaches $\pi/2$ and $\boldsymbol{w}$ is close to $\boldsymbol{v}$, i.e., $(\boldsymbol{w}, \psi_w) = (\boldsymbol{v}, \pi/2)$ is a spurious stationary point. Secondly, the landscape of the loss function is no longer smooth. For both $\boldsymbol{w}$ and $\psi_w$, the local landscape around the global minimum is roughly an absolute function, which declares the non-smoothness of the loss and the failure of gradient descent with a constant step size.

To overcome these obstacles, we apply mild conditions and slight modifications to guarantee convergence and maintain the generality of our conclusion. We separate the phase of the target complex-

valued neuron far from $0$ and $\pi/2$ in consideration of spurious local stationary points: As $\psi_v$ becomes closer to $0$, it is more likely to obtain an initialization of the learning neuron that does not overlap with the target neuron. Then we will take the risk of falling into the spurious local minimum $(\boldsymbol{w}, \psi_w) = (\boldsymbol{0}, 0)$. As $\psi_v$ approaches $\pi/2$, we are confronted by another spurious stationary point $\psi_w = \pi/2$. We utilize gradient descent with a vanishing step size to cope with the non-smoothness of the loss function since a constant step size inevitably suffers from oscillation.

We summarize the proof idea of Theorem 2 as follows. The overall procedure is similar to that of Theorem 1 but every step is different and more challenging because of non-smoothness and more spurious stationary points. Firstly, we identify an ideal region with nice gradient properties: the gradient with respect to $\boldsymbol{w}_\perp$, the weight component perpendicular to $\boldsymbol{v}$, points to the global minimum $\boldsymbol{0}$ and maintains constant order. The gradient $\nabla_{\psi_w}$ is bounded and points towards $\psi_v$ when the angle $\theta_{\boldsymbol{w},\boldsymbol{v}}$ is small enough. Meanwhile, the gradient with respect to $\boldsymbol{w}_{\boldsymbol{v}}$ performs like a contraction mapping with fixed point $[1 - \Theta(\psi_v \psi_w^{-1})]\boldsymbol{v}$ and Lipschitz constant $1 - \Theta(\psi)$, i.e., there exists a deviation of the fixed point from the global minimum. Based on these observations, we then prove the convergence with three stages, as demonstrated in Fig 1(b): In Stage I, $\boldsymbol{w}_\perp$, the weight component perpendicular to $\boldsymbol{v}$, converges to $0$ with an inversely proportional rate, and $\psi_w$ and $\boldsymbol{w}_{\boldsymbol{v}}$ remain in the ideal region. Thus, the angle $\theta_{\boldsymbol{w},\boldsymbol{v}}$ decreases with an inversely proportional rate. When $\theta_{\boldsymbol{w},\boldsymbol{v}}$ declines below some threshold, we come to Stage II where phase $\psi_w$ converges to $\psi_v$ with rate $O(t^{-1})$. As $\psi_w$ approaches $\psi_v$, the fixed point becomes close to $\boldsymbol{v}$ and we step into Stage III where $\boldsymbol{w}$ converges to $\boldsymbol{v}$ with the same rate as $\psi_w$. Finally, we estimate the order of loss and provide a lower bound of the probability of falling into the ideal region with Gaussian initialization to complete the proof. We provide detailed proofs in Appendix C.

### 4.3 Finite-Width RVNNs Cannot Learn a Single Non-degenerate Complex-valued Neuron

We then study learning one complex-valued neuron with zReLU activation using real-valued neurons. Since a complex-valued neuron has more parameters than a real-valued neuron, it is unfair to learn a complex-valued neuron with a single real-valued neuron. Thus, we consider the problem of learning a complex-valued neuron with a two-layer RVNN. A two-layer RVNN with $n$ hidden neurons can be represented by $\boldsymbol{x} \to \boldsymbol{\alpha}^\top \tau(\mathbf{W}\boldsymbol{x})$, where $\mathbf{W} \in \mathbb{R}^{n \times 2d}$ and $\boldsymbol{\alpha} \in \mathbb{R}^n$ indicate weight parameters of the network, and $\tau$ is the ReLU activation function applied componentwisely. We still focus on the expected square loss, which takes the form

$$L_{\mathrm{rc}}(\boldsymbol{\alpha}, \mathbf{W}) = \frac{1}{2}\mathbb{E}_{\boldsymbol{x}\sim\mathcal{N}(\boldsymbol{0},\mathbf{I})}\left[\left(\boldsymbol{\alpha}^\top \tau(\mathbf{W}\boldsymbol{x}) - \sigma_\psi(\boldsymbol{v}_{\mathbb{C}}^\top \overline{\boldsymbol{x}}_{\mathbb{C}})\right)^2\right],$$

where we abbreviate the phase parameter $\psi_v$ as $\psi$ since RVNN has no phase parameter. We are mainly interested in learning a non-degenerate complex-valued neuron, which is distinct from a real-valued neuron and defined as follows.

**Definition 3.** *A complex-valued neuron is a non-degenerate one if $\psi \notin \{0, \pi/2\}$ and $\boldsymbol{v} \neq \boldsymbol{0}$.*

For a complex-valued neuron with phase $\psi = 0$ or $\boldsymbol{v} = \boldsymbol{0}$, the zReLU activation function always outputs $0$. Then the complex-valued neuron is equivalent to a real-valued neuron with all zero weights. For a complex-valued with phase $\psi = \pi/2$, the zReLU activation function is equivalent to the ReLU activation function. Thus, a non-degenerate complex-valued neuron is a non-real-valued neuron. Then we present the third theorem for complex-valued neuron learning.

**Theorem 4.** *Let $d = 1$. For any non-degenerate complex-valued neuron with phase $\psi \in (0, \pi/2)$ and non-zero weight vector $\boldsymbol{v} \in \mathbb{C}^d$, denote by $L_{\mathrm{rc}}$ the expected square loss of learning this complex-valued neuron using a one-hidden-layer RVNN with $n$ hidden neurons. Then the loss satisfies*

$$L_{\mathrm{rc}}(\boldsymbol{\alpha}, \mathbf{W}) \geqslant \frac{\|\boldsymbol{v}\|^2 \min\{2\psi, \pi - 2\psi\}^3}{24\pi(n+2)^2} > 0\,.$$

Theorem 4 provides a positive lower bound for the expected squared loss of approximating a non-degenerate complex-valued neuron using a two-layer RVNN with a fixed number of hidden neurons. This positive lower bound indicates that there always remains a positive gap between the target non-degenerate complex-valued neuron and the two-layer RVNN of fixed width no matter how the parameters of the RVNN are learned. Thus, a finite-width RVNN cannot learn a single non-degenerate complex-valued neuron.

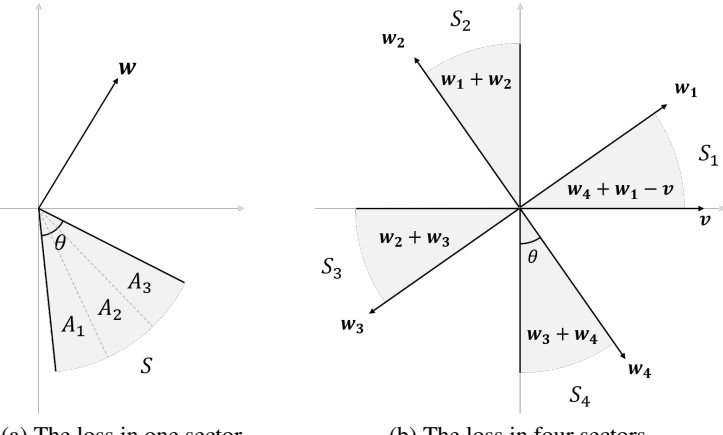

(a) The loss in one sector.    (b) The loss in four sectors.

Figure 2: An illustration of the proof idea of Theorem 4. Shaded areas represent sectors with infinite radii. (a) The expectation $\mathbb{E}_{\boldsymbol{x}\in S}[(\boldsymbol{w}^\top\boldsymbol{x})^2]$ in the sector $S$ equals the sum of the expectations on three subareas $A_1$, $A_2$, and $A_3$. The minimum expectation on a subarea can be bounded by $\Omega(\theta^2\|\boldsymbol{w}\|^2)$. (b) The expected loss of learning a complex-valued neuron using four symmetric real-valued neurons in four symmetric sectors can be bounded by $\Omega(\theta^2\|\boldsymbol{v}\|^2)$, where $\boldsymbol{w}_i$ indicates the weight vector of the $i$-th real-valued neuron, and $\boldsymbol{v}$ denotes that of a complex-valued neuron.

The lower bound decreases at rate $\Theta(\|\boldsymbol{v}\|^2\min\{2\psi,\pi-2\psi\}^3 n^{-2})$. The norm term $\|\boldsymbol{v}\|$ depicts the magnitude of the problem, which affects the expected square loss quadratically from the homogeneity of zReLU. In the extreme case of $\boldsymbol{v}=\boldsymbol{0}$, a trivial real-valued neuron with $\boldsymbol{w}=\boldsymbol{0}$ reaches the lower bound $0$. Meanwhile, the lower bound possesses a positive relation with a phase-dependent term $\{2\psi,\pi-2\psi\}$. Intuitively, this term indicates the difference between a complex-valued neuron and a real-valued neuron. A real-valued neuron corresponds to $\psi=0$ or $\psi=\pi/2$ and this term measures the distance between the phase of a complex-valued neuron and a real-valued one. Finally, the lower bound decreases with a rate inversely proportional to the square of hidden size $n$. We conjecture that this dependence is tight and cannot be improved: a two-layer RVNN with $n$ neurons divides the space into $n$ pieces, in each of which RVNN acts as a linear function. Choosing the $n$ weight vectors of RVNN suitably, the difference between the RVNN and the complex-valued neuron remains small (of order $n^{-1}$) in each piece, which leads to the expected loss of order $O(n^{-2})$.

The conditions in Theorem 4 are made for conciseness of proof and we believe the conclusion holds in more general cases. The dimension $d=1$ corresponds to the intrinsic dimension of expressing a complex-valued neuron because of the rotational invariance of the inner product and the spherical symmetry of Gaussian distributions. Thus, additional dimensions contain no information and cannot improve the efficiency of approximation when $d>1$. It is necessary to consider non-degenerate complex-valued neurons since degenerate complex-valued neurons are equivalent to real-valued ones.

We provide the central proof idea of Theorem 4 as follows. It is observed that the expected square loss $L_{\mathrm{rc}}$ is a piecewise quadratic function and each piece forms a sector centered at the origin with infinite radius. In each piece, $L_{\mathrm{rc}}$ takes the form $\mathbb{E}[(\boldsymbol{w}^\top\boldsymbol{x})^2]$. The proof mainly consists of two steps: we obtain a lower bound of $L_{\mathrm{rc}}$ in a sector and then sum over all sectors with suitable weights and order. Firstly, we consider the expected loss in a sector with a small central angle $\theta$, as shown in Fig. 2(a). We divide the sector into three identical subareas $A_1$, $A_2$, and $A_3$. Then at least one of $A_1$ and $A_3$ remains $\theta/6$ away from the vertical direction of $\boldsymbol{w}$, which leads to a lower bound $\Omega(\theta^2\|\boldsymbol{w}\|^2)$. Secondly, we consider the loss in four rotationally symmetric sectors, as shown in Fig. 2(b), where $\boldsymbol{v}$ represents a complex-valued neuron, $\boldsymbol{w}_i$ indicates a real-valued neuron, and the expression in each sector implies the activated neurons. It is observed that at least one sector possesses a weight vector with norm $\Omega(\|\boldsymbol{v}\|)$, no matter how we choose the real-valued neurons. Thus, the overall loss is bounded by $\Omega(\theta^2\|\boldsymbol{v}\|^2)$. Finally, we take the weight $\boldsymbol{\alpha}$ into consideration and sum over all sectors. For RVNN with $n$ neurons, the best choice of $\theta=\Theta(n^{-1})$ arrives at the lower bound $\Omega(n^{-2}\|\boldsymbol{v}\|^2)$. Detailed proofs are provided in Appendix D.

**Summary and simulation experiments.** We summarize the main conclusions of this section in Table 2. Both a real-valued neuron and a complex-valued neuron succeed in learning functions expressed by any one real-valued neuron. But difference occurs when learning those expressed by any non-degenerate complex-valued neuron: A complex-valued neuron can efficiently learn functions expressed by any one complex-valued neuron, but a two-layer RVNN with finite width cannot learn a single non-degenerate complex-valued neuron. Such a disagreement demonstrates that a complex-valued neuron possesses more powerful learning capability, which profits from the consideration of phase information in complex-valued operations. Our theoretical conclusions are based on the setting of low-dimensional inputs and no bias term, and the simulation results in Fig. 3 verify and extend these discoveries in more general settings. Details about the simulation experiments are available in Appendix F.

Table 2: A complex-valued neuron can learn more than a real-valued neuron.

| Target | Real-valued Neuron | Complex-valued Neuron |
|---|---|---|
| Real-valued Neuron | $O(\mathrm{e}^{-ct})$ [19] | $O(t^{-3})$ (Theorem 1) |
| Complex-valued Neuron | Cannot Learn (Theorem 4) | $O(t^{-1})$ (Theorem 2) |

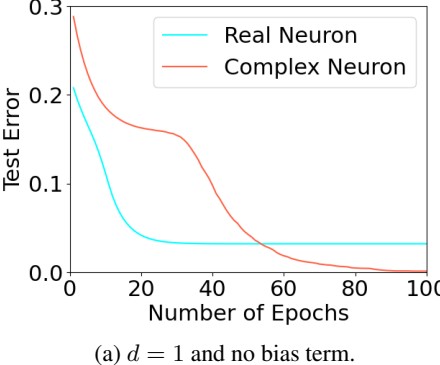

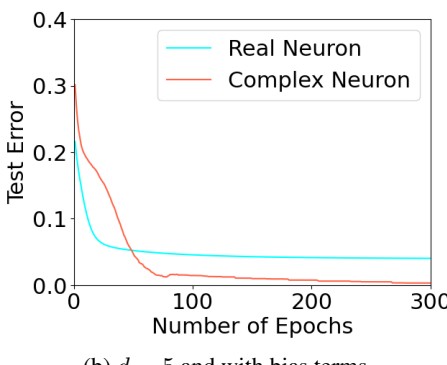

(a) $d = 1$ and no bias term.

(b) $d = 5$ and with bias terms.

Figure 3: The test error of learning a complex-valued neuron. In both the theoretical setting (Fig. 3a) and more general settings (Fig. 3b), complex-valued neurons have vanishing errors, while real-valued neurons converge to positive errors.

## 5   Complex-valued Neurons Learn Slower

In this section, we demonstrate that complex-valued neurons learn slower than real-valued neurons. To arrive at this conclusion, we first rephrase the linear convergence of learning functions expressed by any one real-valued neuron using real-valued neurons. Then we prove that a complex-valued neuron learns the same class of functions at an exponentially slower rate.

We concentrate on learning one real-valued neuron $\boldsymbol{x} \to \tau(\boldsymbol{v}^\top \boldsymbol{x})$ with $\|\boldsymbol{v}\| = 1$. When learning one real-valued neuron using a real-valued neuron, the expected square loss in Eq. (1) possesses the following simple closed form [41]

$$L_{\mathrm{rr}}(\boldsymbol{w}) = \frac{1}{4}\|\boldsymbol{w}\|^2 - \frac{1}{2\pi}\|\boldsymbol{w}\|[\sin\theta_{\boldsymbol{w},\boldsymbol{v}} + (\pi - \theta_{\boldsymbol{w},\boldsymbol{v}})\cos\theta_{\boldsymbol{w},\boldsymbol{v}}] + \frac{1}{4} .$$

It is widely known that a real-valued neuron learns a real-valued neuron with high probability and linear convergence rate [19], as reformulated by the following lemma.

**Lemma 5.** *[19, Theorem 6.4] Suppose that the weight vector $\boldsymbol{w} \in \mathbb{R}^{2d}$ is initialized by a Gaussian distribution $\mathcal{N}(0, \mathbf{I}/(2d))$. Let $L_{\mathrm{rr}}$ denote the expected square loss of learning a real-valued neuron using a real-valued neuron. Then there exist constants $c_1, c_2$ such that gradient descent with suitable step size satisfies*

$$\Pr[L_{\mathrm{rr}}(\boldsymbol{w}_t) \leqslant \mathrm{e}^{-c_1 t}] \geqslant 1 - \mathrm{e}^{-c_2 d} .$$

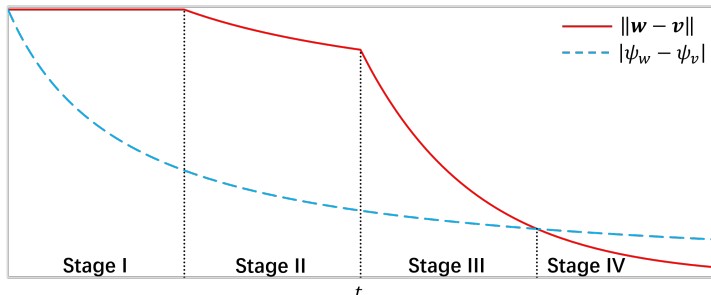

Figure 4: A demonstration of the convergence stages of Theorem 6. The horizontal axis represents the iteration index of gradient descent. The black dotted line is the separation of convergence stages.

Recalling the expected loss of learning one real-valued neuron with a complex-valued neuron in Eq. (2), then we present the fourth theorem for complex-valued neuron learning, which provides a lower bound for the convergence rate.

**Theorem 6.** *Let $d = 1$. Suppose that $\|\boldsymbol{w}_0 - \boldsymbol{v}\| < 1$. Let $\{(\boldsymbol{w}_t, \psi_t)\}_{t=0}^{\infty}$ denote the parameter sequence of the complex-valued neuron generated by projected gradient descent when optimizing $L_{\mathrm{cr}}$, the expected loss of learning a real-valued neuron using a complex-valued neuron. If the step size $\eta_t = \eta \in (0, 1/(12\pi))$, then we have*

$$L_{\mathrm{cr}}(\boldsymbol{w}_t, \psi_t) \geqslant \frac{(1 - 12\eta)^{3T_3/2}(\psi^* - \psi_0)^3}{8\pi(t - T_3 + 1)^3} - \frac{1}{2\pi}\left(1 - \frac{\eta}{48}\right)^{t-T_3},$$

*where $\psi^* = \pi/2$, and $T_3$ is a constant dependent on $\|\boldsymbol{w}_0 - \boldsymbol{v}\|$, $\eta$, and $\psi^* - \psi_0$.*

Theorem 6 presents a lower bound for the expected loss of learning one real-valued neuron with a complex-valued neuron. It is observed that the negative term in the lower bound becomes $0$ exponentially fast as $t$ increases, and the positive term decreases with order $\Omega(t^{-3})$. Thus, the expected loss possesses a lower bound $\Omega(t^{-3})$ since the positive term dominates the loss when $t$ grows sufficiently large. This lower bound matches the upper bound in Theorem 1. Thus, $O(t^{-3})$ becomes the utmost limit of learning with a complex-valued neuron via gradient descent, i.e., we cannot expect a complex-valued neuron to learn faster than this utmost limit.

The conditions in Theorem 6 are technical and reasonable. The condition on $\boldsymbol{w}_0$ is made for the conciseness of proof and can be removed. It is observed that $(\boldsymbol{w}, \psi) = (\boldsymbol{v}, \psi^*)$ is the unique global minimum with $L_{\mathrm{cr}} = 0$. Meanwhile, it is easy to verify that the loss goes to infinity when $\|\boldsymbol{w}\| \to \infty$. Thus, if we aim to obtain a small loss, the parameter sequence must fall into a small neighborhood of the global minimum, which is depicted by condition $\|\boldsymbol{w}_0 - \boldsymbol{v}\| \leqslant R < 1$. The condition $\psi_0 \neq \psi^*$ holds with probability 1 when we initialize $\psi_0$ with a continuous distribution. This condition is necessary to obtain a meaningful lower bound since the numerator of the positive term equals $0$ when $\psi_0 = \psi^*$. We emphasize that this condition is essential and cannot be removed because a complex-valued neuron with $\psi_0 = \psi^*$ is equivalent to a real-valued neuron, which enjoys linear convergence as stated in Lemma 5. The condition $d = 1$ corresponds to the simplest optimization problem of learning one real-valued neuron with a complex-valued neuron since high-dimensional optimization brings more difficulties. Thus, we cannot expect a complex-valued neuron to learn a real-valued neuron with a convergence rate faster than $O(t^{-3})$ in a higher dimension.

We summarize the proof idea of Theorem 6 as follows. The gradient with respect to $\psi$ possesses the order $(\psi^* - \psi)^2 + (\psi^* - \psi)\theta_{\boldsymbol{w},\boldsymbol{v}}$. The key intuition is that $\psi$ converges fast to the global minimum when $\theta_{\boldsymbol{w},\boldsymbol{v}}$ remains large, but $\theta_{\boldsymbol{w},\boldsymbol{v}}$ diminishes as $\boldsymbol{w}$ converges to the global minimum $\boldsymbol{v}$. The detailed proofs are complicated and consist of several stages, depicting the entangled convergence between $\boldsymbol{w}$ and $\psi$ as shown in Figure 4. In Stage I, $\psi$ increases above a positive constant, which is a necessary condition for fast convergence of $\boldsymbol{w}$ in Stage II. When the distance between $\boldsymbol{w}$ and $\boldsymbol{v}$ declines below a threshold, the angle $\theta_{\boldsymbol{w},\boldsymbol{v}}$ becomes small. Then we enter Stage III, where $\boldsymbol{w}$ converges faster than $\psi$. Stage IV begins when $\psi^* - \psi$ dominates $\theta_{\boldsymbol{w},\boldsymbol{v}}$. Then the gradient degenerates to order $(\psi^* - \psi)^2$, which implies a lower bound of convergence $\psi^* - \psi = \Omega(t^{-1})$. Finally, estimating the loss around the global minimum leads to the conclusion. Detailed proofs are provided in Appendix E.

**Summary and simulation experiments.** Table 3 summarizes the conclusions in this section, which shows that a complex-valued neuron learns slower than a real-valued one. A complex-valued neuron is more flexible since it can learn the phase. But this flexibility becomes redundant and slows down the convergence when learning a phase-independent function. Our theories are based on the setting of low-dimensional inputs and no bias term, and the simulation results in Fig. 3 verify and extend these discoveries in more general settings. Details about the experiments are available in Appendix F.

Table 3: A real-valued neuron learns faster than a complex-valued neuron.

| Target | Real-valued Neuron | Complex-valued Neuron |
|---|---|---|
| Real-valued Neuron | $O(\mathrm{e}^{-ct})$ (Lemma 5) | $\Omega(t^{-3})$ (Theorem 6) |

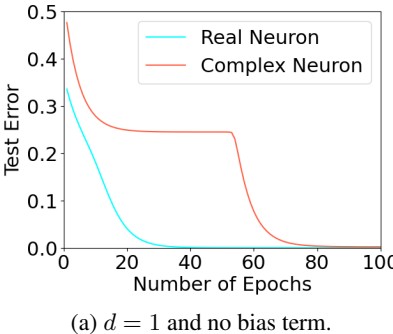
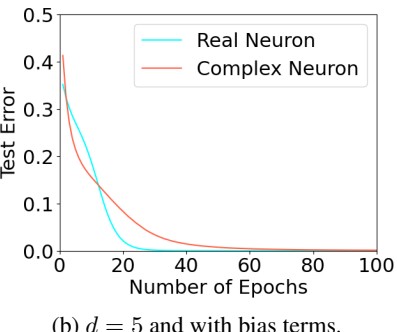

(a) $d = 1$ and no bias term.    (b) $d = 5$ and with bias terms.

Figure 5: The test error of learning a real-valued neuron. In both the theoretical setting (Fig. 5a) and more general settings (Fig. 5b), a complex-valued neuron learns a real-valued neuron slower.

## 6 Conclusions and Prospects

In this paper, we investigate the problem of learning a single neuron using another neuron by optimizing the expected square loss via gradient descent. Firstly, we prove that a complex-valued neuron can efficiently learn functions expressed by any one real-valued neuron and any one complex-valued neuron with convergence rate $O(t^{-3})$ and $O(t^{-1})$, respectively, where $t$ denotes the iteration index of gradient descent. Meanwhile, two-layer RVNNs with finite width cannot learn a single non-degenerate complex-valued neuron in a strong sense that there always exists a positive gap between a two-layer RVNN of fixed width and a non-degenerate complex-valued neuron. These conclusions suggest that complex-valued neurons can learn more than real-valued neurons since CVNNs benefit from the phase parameter, which helps CVNNs learn phase information more efficiently. Secondly, we provide a convergence lower bound $\Omega(t^{-3})$, which matches the upper bound, for learning one real-valued neuron with a complex-valued neuron. This conclusion, together with the well-known linear convergence of learning one real-valued neuron with a real-valued neuron, implies that complex-valued neurons learn slower than real-valued neurons in phase-independent tasks. This phenomenon captures the additional price for learning simpler tasks with more complicated models, where the redundant phase consideration exponentially slows down the convergence.

Our study serves as a preliminary attempt to compare the learning process of artificial neural networks with different functional operations. In the future, it is important to extend our theoretical results to more general settings, such as cases of high-dimensional inputs, equipped with bias terms, and over-parameterized architectures [42]. Meanwhile, it is prospective to investigate complex-valued neuron learning from finite samples and derive a high-probability convergence condition. Since the empirical loss is a piecewise constant function with respect to the learnable phase parameter, it might be necessary to explore new learning algorithms, which is also encouraged by the neural tangent kernel aspect [43]. Besides, it is promising to consider the more practical and challenging procedure of learning general functions with deep architectures.

## Acknowledgments

This research was supported by NSFC (61921006) and Collaborative Innovation Center of Novel Software Technology and Industrialization. We would like to thank the anonymous reviewers for their invaluable suggestions.

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

# Appendix of "Complex-valued Neurons Can Learn More but Slower than Real-valued Neurons via Gradient Descent"

## A Preliminaries

In this section, we first summarize frequently used notations in the following table.

Table 4: Frequently used notations.

| Notation | Description |
|---|---|
| $\mathbb{C}^d$ | the $d$-dimensional complex space |
| $\mathbb{E}$ | expectation |
| $\mathbb{I}(\cdot)$ | the indicator function |
| $L$ | the expected square loss of learning a neuron |
| $\mathcal{N}(\mathbf{0}, \mathbf{I})$ | the standard Gaussian distribution |
| $O, \Omega, \Theta$ | asymptotic notations |
| $\Pr$ | probability |
| $P_{\mathcal{Q}}(\boldsymbol{x})$ | the projection of $\boldsymbol{x}$ on $\mathcal{Q}$ |
| $\mathbb{R}^{2d}$ | the $2d$-dimensional real space |
| $\mathrm{Re}(z)$ | the real part of a complex number $z$ |
| $t$ | the iteration index of gradient descent |
| $\mathcal{U}(a, b)$ | the uniform distribution on the interval $[a, b]$ |
| $\boldsymbol{v}$ | the weight vector of a learning neuron |
| $\boldsymbol{w}$ | the weight vector of a target neuron |
| $\boldsymbol{x}$ | an input vector in $\mathbb{R}^{2d}$ |
| $x_i$ | the $i$-th coordinate of $\boldsymbol{x}$ |
| $\boldsymbol{x}_{\mathbb{C}}$ | $\boldsymbol{x}_{\mathbb{C}} = (x_1; \ldots; x_d) + (x_{d+1}; \ldots; x_{2d})\mathrm{i} \in \mathbb{C}^d$ |
| $\overline{\boldsymbol{x}}_{\mathbb{C}}$ | the complex conjugate of $\boldsymbol{x}_{\mathbb{C}}$ |
| $\theta_{\boldsymbol{a}, \boldsymbol{b}}$ | the angle between $\boldsymbol{a}$ and $\boldsymbol{b}$ |
| $\theta_z$ | the argument of a complex number $z$ |
| $\sigma_\psi(z)$ | the real part of the symmetrical version of zReLU activation function |
| $\eta$ | the step size of gradient descent |
| $\tau$ | the ReLU activation function $\tau(x) = \max\{0, x\}$ |
| $\psi$ | the learnable parameter of the symmetrical version of zReLU activation function |
| $\nabla$ | gradient |
| $\|\cdot\|$ | the 2-norm of a vector |

We then give some basic lemmas that help us calculate the closed form of the expected loss.

**Lemma 7.** *Let $d = 1$. For any $\boldsymbol{w}, \boldsymbol{v} \in \mathbb{R}^{2d}$, and $a \leqslant b \leqslant a + 2\pi$, we have*

$$A(\boldsymbol{w}, \boldsymbol{v}, a, b) = \mathbb{E}_{\boldsymbol{x} \sim \mathcal{N}(\mathbf{0}, \mathbf{I})} \left[ \boldsymbol{w}^\top \boldsymbol{x} \cdot \boldsymbol{v}^\top \boldsymbol{x} \cdot \mathbb{I}(\theta_x \in [a, b]) \right]$$

$$= \frac{\|\boldsymbol{w}\|\|\boldsymbol{v}\|}{4\pi} \left[ 2(b - a)\cos\theta_{\boldsymbol{w}, \boldsymbol{v}} + \sin(\theta_{\boldsymbol{w}} + \theta_{\boldsymbol{v}} - 2a) - \sin(\theta_{\boldsymbol{w}} - \theta_{\boldsymbol{v}} - 2b) \right] .$$

**Proof.** According to the probability density function of Gaussian distribution, we can calculate $A$ in the polar coordinate system as

$$A(\boldsymbol{w}, \boldsymbol{v}, a, b) = \frac{\|\boldsymbol{w}\|\|\boldsymbol{v}\|}{2\pi} \int_0^\infty \int_a^b r^3 \mathrm{e}^{-\frac{1}{2}r^2} \cos(\theta_{\boldsymbol{w}} - \phi) \cos(\theta_{\boldsymbol{v}} - \phi) \, \mathrm{d}\phi \, \mathrm{d}r$$

$$= \frac{\|\boldsymbol{w}\|\|\boldsymbol{v}\|}{\pi} \int_a^b \cos(\theta_{\boldsymbol{w}} - \phi) \cos(\theta_{\boldsymbol{v}} - \phi) \, \mathrm{d}\phi$$

$$= \frac{\|\boldsymbol{w}\|\|\boldsymbol{v}\|}{4\pi} \left[ 2(b - a)\cos\theta_{\boldsymbol{w}, \boldsymbol{v}} + \sin(\theta_{\boldsymbol{w}} + \theta_{\boldsymbol{v}} - 2a) - \sin(\theta_{\boldsymbol{w}} - \theta_{\boldsymbol{v}} - 2b) \right] ,$$

where the second and third equalities hold from integrating over $r$ and $\phi$, respectively. Thus, we have completed the proof. $\qquad \square$

**Lemma 8.** *Let $d = 1$. For any $\boldsymbol{w}, \boldsymbol{v} \in \mathbb{R}^{2d}$, denote by $\theta = \theta_{\boldsymbol{w},\boldsymbol{v}}$ the angle between $\boldsymbol{w}$ and $\boldsymbol{v}$. Then for any $\psi_w, \psi_v \in [0, \pi/2]$, define $\psi_m = \min\{\psi_w, \psi_v\}$. Then we have*

$$B(\boldsymbol{w}, \boldsymbol{v}, \psi_w, \psi_v) = \mathbb{E}_{\boldsymbol{x} \sim \mathcal{N}(\boldsymbol{0}, \mathbf{I})} \left[ \sigma_{\psi_w}(\boldsymbol{w}_{\mathbb{C}}^{\top} \overline{\boldsymbol{x}}_{\mathbb{C}}) \sigma_{\psi_v}(\boldsymbol{v}_{\mathbb{C}}^{\top} \overline{\boldsymbol{x}}_{\mathbb{C}}) \right]$$

$$= \begin{cases} \frac{\|\boldsymbol{w}\|\|\boldsymbol{v}\|}{2\pi} \cos \theta_{\boldsymbol{w},\boldsymbol{v}} [2\psi_m + \sin(2\psi_m)] , & \theta_{\boldsymbol{w},\boldsymbol{v}} \in [0, |\psi_v - \psi_w|] , \\ \frac{\|\boldsymbol{w}\|\|\boldsymbol{v}\|}{4\pi} [2(\psi_w + \psi_v - \theta_{\boldsymbol{w},\boldsymbol{v}}) \cos \theta_{\boldsymbol{w},\boldsymbol{v}} - \sin(\theta_{\boldsymbol{w},\boldsymbol{v}} - 2\psi_v) \\ \qquad - \sin(\theta_{\boldsymbol{w},\boldsymbol{v}} - 2\psi_w)] , & \theta_{\boldsymbol{w},\boldsymbol{v}} \in [|\psi_v - \psi_w|, \psi_v + \psi_w] , \\ 0 , & \theta_{\boldsymbol{w},\boldsymbol{v}} \in [\psi_v + \psi_w, \pi] . \end{cases}$$

**Proof.** We only consider the case of $\psi_w \leqslant \psi_v$. The other case $\psi_w \geqslant \psi_v$ can be proven similarly. We prove the conclusion by discussion.

1. Suppose $\theta_{\boldsymbol{w},\boldsymbol{v}} \in [0, \psi_v - \psi_w]$. Then Lemma 7 leads to

$$B(\boldsymbol{w}, \boldsymbol{v}, \psi_w, \psi_v) = A(\boldsymbol{w}, \boldsymbol{v}, \theta_{\boldsymbol{w}} - \psi_w, \theta_{\boldsymbol{w}} + \psi_w) = \frac{\|\boldsymbol{w}\|\|\boldsymbol{v}\|}{2\pi} \cos \theta_{\boldsymbol{w},\boldsymbol{v}} [2\psi_w + \sin(2\psi_w)] .$$

2. Suppose $\theta_{\boldsymbol{w},\boldsymbol{v}} \in [\psi_v - \psi_w, \psi_v + \psi_w]$ and $\theta_{\boldsymbol{w}} \leqslant \theta_{\boldsymbol{v}}$. Then one knows from Lemma 7 that

$$B(\boldsymbol{w}, \boldsymbol{v}, \psi_w, \psi_v) = A(\boldsymbol{w}, \boldsymbol{v}, \theta_{\boldsymbol{v}} - \psi_v, \theta_{\boldsymbol{w}} + \psi_w)$$

$$= \frac{\|\boldsymbol{w}\|\|\boldsymbol{v}\|}{4\pi} [2(\psi_w + \psi_v - \theta_{\boldsymbol{w},\boldsymbol{v}}) \cos \theta_{\boldsymbol{w},\boldsymbol{v}} - \sin(\theta_{\boldsymbol{w},\boldsymbol{v}} - 2\psi_v) - \sin(\theta_{\boldsymbol{w},\boldsymbol{v}} - 2\psi_w)] .$$

3. Suppose $\theta_{\boldsymbol{w},\boldsymbol{v}} \in [\psi_v - \psi_w, \psi_v + \psi_w]$ and $\theta_{\boldsymbol{w}} \geqslant \theta_{\boldsymbol{v}}$. Based on Lemma 7, we have

$$B(\boldsymbol{w}, \boldsymbol{v}, \psi_w, \psi_v) = A(\boldsymbol{w}, \boldsymbol{v}, \theta_{\boldsymbol{w}} - \psi_w, \theta_{\boldsymbol{v}} + \psi_v)$$

$$= \frac{\|\boldsymbol{w}\|\|\boldsymbol{v}\|}{4\pi} [2(\psi_w + \psi_v - \theta_{\boldsymbol{w},\boldsymbol{v}}) \cos \theta_{\boldsymbol{w},\boldsymbol{v}} - \sin(\theta_{\boldsymbol{w},\boldsymbol{v}} - 2\psi_v) - \sin(\theta_{\boldsymbol{w},\boldsymbol{v}} - 2\psi_w)] .$$

4. Suppose $\theta_{\boldsymbol{w},\boldsymbol{v}} \in [\psi_v + \psi_w, \pi]$. Then the support of $\sigma_{\psi_w}(\boldsymbol{w}_{\mathbb{C}}^{\top} \overline{\boldsymbol{x}}_{\mathbb{C}})$ does not overlap with that of $\sigma_{\psi_v}(\boldsymbol{v}_{\mathbb{C}}^{\top} \overline{\boldsymbol{x}}_{\mathbb{C}})$, which leads to $B(\boldsymbol{w}, \boldsymbol{v}, \psi_w, \psi_v) = 0$.

Combining the cases above completes the proof. □

## B  Proof of Theorem 1

In the main part of this section, we provide the closed form of the loss, definition of the ideal region, and the detailed proof of Theorem 1. Subsection B.1 presents the optimization behaviors in the ideal region. Subsection B.2 proves several convergence rate lemmas. Subsection B.3 gives some technical lemmas to bound small terms in the proof.

Let $\boldsymbol{w} = (w_1, w_2)$. According to the spherical symmetry, we assume $\boldsymbol{v} = (1, 0)$ without loss of generality. According to Lemma 8, the expected loss can be calculated by

$$L_{\mathrm{cr}}(\boldsymbol{w}, \psi) = \frac{1}{2} B(\boldsymbol{w}, \boldsymbol{w}, \psi, \psi) - B(\boldsymbol{w}, \boldsymbol{v}, \psi, \pi/2) + \frac{1}{2} B(\boldsymbol{v}, \boldsymbol{v}, \pi/2, \pi/2)$$

$$= \begin{cases} \frac{1}{4} - \frac{1}{4\pi} [\sin(2\psi) + 2\psi][1 - (w_1 - 1)^2 - w_2^2] , & \theta \in [0, \pi/2 - \psi] , \\ \frac{1}{4} - \frac{1}{2\pi} [\frac{1}{2} \sin(2\psi) w_1 - \frac{1}{2} \cos(2\psi)|w_2| + \frac{1}{2}|w_2| + (\frac{\pi}{2} + \psi - \theta) w_1] \\ \qquad + \frac{1}{4\pi} [\sin(2\psi) + 2\psi](w_1^2 + w_2^2) , & \theta \in (\pi/2 - \psi, \pi/2 + \psi) , \\ \frac{1}{4} + \frac{1}{4\pi} [2\psi + \sin(2\psi)](w_1^2 + w_2^2) , & \theta \in [\pi/2 + \psi, \pi] , \end{cases} \tag{3}$$

where $\theta = \theta_{\boldsymbol{w},\boldsymbol{v}} = \arccos(w_1/\sqrt{w_1^2 + w_2^2})$. For any $R \in (0, 1)$, define

$$D_1 = \{(\boldsymbol{w}, \psi) \mid \|\boldsymbol{w} - \boldsymbol{v}\| \leqslant R, \psi \in [0, \pi/2], \theta \in [0, \pi/2 - \psi]\} ,$$

$$D_2 = \{(\boldsymbol{w}, \psi) \mid \|\boldsymbol{w} - \boldsymbol{v}\| \leqslant R, \psi \in [0, \pi/2], \theta \in (\pi/2 - \psi, \pi/2 + \psi)\} .$$

Let $D = D_1 \cup D_2$ denote the ideal region, i.e.,

$$D = \{(\boldsymbol{w}, \psi) \mid \|\boldsymbol{w} - \boldsymbol{v}\| \leqslant R, \psi \in [0, \pi/2], \theta \in [0, \pi/2 + \psi]\} .$$

We are now ready to prove Theorem 1.

**Proof of Theorem 1.** The proof is divided into four steps.

**Step 1:** $D$ **is closed under gradient descent.** Before considering the convergence, we prove the maintenance of inclusion by mathematical induction, i.e., $(\boldsymbol{w}_0, \psi_0) \in D$ indicates $(\boldsymbol{w}_t, \psi_t) \in D$.

1. Base case. The conclusion holds for $t = 0$ from the condition.

2. Induction. Suppose that the conclusion holds for $t = k$ with $k \in \mathbb{N}$. Then based on Lemmas 11 and 12, one knows

$$-6(\psi^* - \psi_k) \leqslant \nabla_\psi L_{\mathrm{cr}}(\boldsymbol{w}_k, \psi_k) \leqslant -\frac{1 - R^2}{4\pi}(\psi^* - \psi_k)^2 \leqslant 0 \,, \tag{4}$$

where $\psi^* = \pi/2$, the first inequality holds based on the induction hypothesis and $|w_{2,k}| \leqslant 1$, and the third inequality holds from $R < 1$. Thus, the updating rule $\psi_{k+1} = \psi_k - \eta \nabla_\psi L_{\mathrm{cr}}(\boldsymbol{w}_k, \psi_k)$ with $\eta \in (0, 1/(12\pi))$ leads to

$$\frac{\pi}{2} \geqslant \psi^* - \psi_k \geqslant \psi^* - \psi_{k+1} \geqslant (1 - 6\eta)(\psi^* - \psi_k) \geqslant 0 \,, \tag{5}$$

where the first and fourth inequalities hold from the induction hypothesis. Meanwhile, Lemmas 9 and 10 imply

$$\|\boldsymbol{w}_{k+1} - \boldsymbol{v}\| \leqslant \left(1 - \frac{\eta}{24\pi}[\sin(2\psi_k) + 2\psi_k]\right)\|\boldsymbol{w}_k - \boldsymbol{v}\| \leqslant R \,. \tag{6}$$

Combining Eqs. (5) and (6), the conclusion holds for $t = k + 1$.

Therefore, mathematical induction implies $(\boldsymbol{w}_t, \psi_t) \in D$ when $(\boldsymbol{w}_0, \psi_0) \in D$.

**Step 2: parameters converge to the global minimum in $D$.** The convergence process consists of two stages. In stage I, we deal with the convergence of $\psi$ when $(\boldsymbol{w}_0, \psi_0) \in D$. Based on Eq. (4) and the updating rule $\psi_{k+1} = \psi_k - \eta \nabla_\psi L_{\mathrm{cr}}(\boldsymbol{w}_k, \psi_k)$, one knows

$$\psi^* - \psi_{t+1} \leqslant (\psi^* - \psi_t)\left[1 - \frac{\eta(1 - R^2)}{4\pi}(\psi^* - \psi_t)\right].$$

Define $a_t = \eta(1 - R^2)(\psi^* - \psi_t)/(4\pi)$. Then we obtain $a_{t+1} \leqslant a_t(1 - a_t)$. From $\psi^* - \psi_t \in [0, \pi/2]$ and $\eta < 1/(12\pi) \leqslant 4$, one knows $a_t \in [0, 1/2]$. Thus, applying Lemma 14 to $a_t$ leads to

$$\psi^* - \psi_t = \frac{4\pi a_t}{\eta(1 - R^2)} \leqslant \frac{4\pi}{\eta(1 - R^2)(t + 1)} \,. \tag{7}$$

In stage II, we consider the convergence of $\boldsymbol{w}$ when $(\boldsymbol{w}_0, \psi_0) \in D$. Based on Eq. (7), choosing $T_1 \geqslant 16\lceil \eta(1 - R^2)\rceil^{-1}$ leads to $\psi^* - \psi_t \leqslant \pi/4$ for any $t \geqslant T_1$, i.e., $\psi_t \geqslant \pi/4$ for any $t \geqslant T_1$. Thus, for any $t \geqslant T_1$, Eq. (6) indicates

$$\|\boldsymbol{w}_t - \boldsymbol{v}\| \leqslant \left(1 - \frac{\eta}{48}\right)\|\boldsymbol{w}_{t-1} - \boldsymbol{v}\| \leqslant \left(1 - \frac{\eta}{48}\right)^{t - T_1} \,, \tag{8}$$

where the first inequality holds from the monotonic increasing of $\sin(x) + x$ and $\psi_t \geqslant \pi/4$, and the second inequality holds because of $\|\boldsymbol{w}_{T_1} - \boldsymbol{v}\| \leqslant R < 1$.

**Step 3: the loss converges to $0$ in $D$.** We estimate the convergence of the expected loss when $(\boldsymbol{w}_0, \psi_0) \in D$. For any $(\boldsymbol{w}, \psi) \in D$, define non-negative quantities $\Delta_{\boldsymbol{w}} = \|\boldsymbol{w} - \boldsymbol{v}\|$ and $\Delta_\psi = \psi^* - \psi$. We provide an upper bound for $L_{\mathrm{cr}}$ by discussion.

1. Suppose $(\boldsymbol{w}, \psi) \in D_1$. Then we have

$$L_{\mathrm{cr}}(\boldsymbol{w}, \psi) \leqslant \frac{1}{4} - \frac{1}{2\pi}(\psi^* - \Delta_\psi^3)(1 - \Delta_{\boldsymbol{w}}^2) \leqslant \frac{1}{2\pi}\Delta_\psi^3 + \frac{1}{4}\Delta_{\boldsymbol{w}}^2 \,, \tag{9}$$

where the first inequality holds based on $\sin(2\psi) + 2\psi = \sin(2\Delta_\psi) + 2\psi^* - 2\Delta_\psi \geqslant 2\psi^* - 2\Delta_\psi^3$, and the second inequality holds from non-negative $\Delta_\psi$.

2. Suppose $(\boldsymbol{w}, \psi) \in D_2$. The expected loss can be rewritten as

$$
\begin{aligned}
L_{\mathrm{cr}}(\boldsymbol{w}, \psi) &= \frac{1}{4} - \frac{1}{4\pi}[\sin(2\psi) + 2\psi](1 - \Delta_{\boldsymbol{w}}^2) \\
&\quad + \frac{1}{4\pi}[(\cos(2\psi) - 1)|w_2| + (\sin(2\psi) + 2\psi + 2\theta - 2\psi^*)w_1] \\
&\leqslant \frac{1}{4} - \frac{1}{2\pi}(\psi^* - \Delta_\psi^3)(1 - \Delta_{\boldsymbol{w}}^2) + \frac{1}{4\pi}[(\pi + 2\theta - 2\psi^*)w_1] \tag{10} \\
&\leqslant \frac{1}{4} - \frac{1}{2\pi}(\psi^* - \Delta_\psi^3)(1 - \Delta_{\boldsymbol{w}}^2) + \frac{1}{2\pi}\Delta_{\boldsymbol{w}}(1 + \Delta_{\boldsymbol{w}}) \\
&\leqslant \frac{1}{2\pi}\Delta_\psi^3 + \frac{1}{2\pi}\Delta_{\boldsymbol{w}} + \frac{1}{2}\Delta_{\boldsymbol{w}}^2 \,,
\end{aligned}
$$

where the first inequality holds from $\pi \geqslant \sin(2\psi) + 2\psi \geqslant 2\psi^* - 2\Delta_\psi^3$ and $\cos(2\psi) - 1 \leqslant 0$, the second inequality holds based on $\theta \leqslant \tan\theta \leqslant \Delta_{\boldsymbol{w}}$ and $w_1 \leqslant 1 + \Delta_{\boldsymbol{w}}$, and the third inequality holds from $\Delta_\psi \geqslant 0$.

Combining Eqs. (9) and (10), one knows that the following holds for any $(\boldsymbol{w}_0, \psi_0) \in D$ and $t \geqslant T_1$

$$L_{\mathrm{cr}}(\boldsymbol{w}_t, \psi_t) \leqslant \frac{1}{2\pi}\Delta_{\psi,t}^3 + \Delta_{\boldsymbol{w},t} \leqslant \frac{32\pi^3}{\eta^3(1 - R^2)^3 t^3} + \left(1 - \frac{\eta}{48}\right)^{t - T_1} , \tag{11}$$

where the first inequality holds from $\Delta_{\boldsymbol{w}}^2 \leqslant \Delta_{\boldsymbol{w}}$, and the second inequality holds by Eqs. (7) and (8).

**Step 4: initialization falls into $D$ with constant probability.** Let $p_0 = \Pr[(\boldsymbol{w}_0, \psi_0) \in D]$ for simplicity. From $\psi_0 \sim \mathcal{U}(0, \pi/2)$, the requirement $\psi \in [0, \pi/2]$ is satisfied. Denote by $p(\boldsymbol{w})$ the probability density function of $\mathcal{N}(0, I_2)$. Then one has

$$p_0 = \Pr[\|\boldsymbol{w}_0 - \boldsymbol{v}\| \leqslant R] = \int_{\boldsymbol{w} \in B(\boldsymbol{v}, R)} p(\boldsymbol{w}) \, \mathrm{d}\boldsymbol{w} \geqslant \mu(B(\boldsymbol{v}, R)) \min_{\boldsymbol{w} \in B(\boldsymbol{v}, R)} p(\boldsymbol{w}) \geqslant \frac{R^2}{16} . \tag{12}$$

Let $R^2 = 1/2$. We obtain from Eqs. (11) and (12) that

$$\Pr\left[L_{\mathrm{cr}}(\boldsymbol{w}_t, \psi_t) \leqslant \frac{8000}{\eta^3 t^3} + \left(1 - \frac{\eta}{48}\right)^{t + 1 - 32/\eta}\right] \geqslant \frac{1}{32} ,$$

which completes the proof. $\qquad\square$

## B.1 Optimization Behaviors

The following two lemmas indicate the linear convergence of $\boldsymbol{w}$ in $D_1$ and $D_2$, respectively.

**Lemma 9.** *Let $\boldsymbol{w}' = \boldsymbol{w} - \eta\nabla_{\boldsymbol{w}} L_{\mathrm{cr}}(\boldsymbol{w}, \psi)$. If $(\boldsymbol{w}, \psi) \in D_1$ and $\eta \in (0, 4)$, then we have*

$$\|\boldsymbol{w}' - \boldsymbol{v}\| \leqslant \left(1 - \frac{\eta}{4\pi}[\sin(2\psi) + 2\psi]\right) \|\boldsymbol{w} - \boldsymbol{v}\| .$$

**Proof.** For any $(\boldsymbol{w}, \psi) \in D_1$, one has

$$\langle\nabla_{\boldsymbol{w}} L_{\mathrm{cr}}(\boldsymbol{w}, \psi), \boldsymbol{w} - \boldsymbol{v}\rangle = \left\langle\frac{1}{4\pi}[\sin(2\psi) + 2\psi](\boldsymbol{w} - \boldsymbol{v}), \boldsymbol{w} - \boldsymbol{v}\right\rangle = \frac{1}{4\pi}[\sin(2\psi) + 2\psi]\|\boldsymbol{w} - \boldsymbol{v}\|^2 .$$

Meanwhile,

$$\|\nabla_{\boldsymbol{w}} L_{\mathrm{cr}}(\boldsymbol{w}, \psi)\|^2 = \frac{1}{(4\pi)^2}[\sin(2\psi) + 2\psi]^2\|(\boldsymbol{w} - \boldsymbol{v})\|^2 .$$

Then according to Lemma 13 and $\psi \in [0, \pi/2]$, for any $\eta \in (0, 4)$, one has

$$\|\boldsymbol{w}' - \boldsymbol{v}\| \leqslant \left(1 - \frac{\eta}{4\pi}[\sin(2\psi) + 2\psi]\right) \|\boldsymbol{w} - \boldsymbol{v}\| ,$$

which completes the proof. $\qquad\square$

**Lemma 10.** *Let $\boldsymbol{w}' = \boldsymbol{w} - \eta\nabla_{\boldsymbol{w}} L_{\mathrm{cr}}(\boldsymbol{w}, \psi)$. If $(\boldsymbol{w}, \psi) \in D_2$ and $\eta \in (0, 1/(12\pi))$, then we have*

$$\|\boldsymbol{w}' - \boldsymbol{v}\| \leqslant \left(1 - \frac{\eta}{24\pi}[\sin(2\psi) + 2\psi]\right) \|\boldsymbol{w} - \boldsymbol{v}\| .$$

**Proof.** Firstly, we prove the strong convexity in $D_2$. For any $(\boldsymbol{w}, \psi) \in D_2$, one has

$$\begin{aligned}
&2\pi\langle\nabla_{\boldsymbol{w}} L_{\mathrm{cr}}(\boldsymbol{w}, \psi), \boldsymbol{w} - \boldsymbol{v}\rangle \\
&= -\left[\frac{1}{2}\sin(2\psi) + \left(\frac{\pi}{2} + \psi - \theta\right) + \frac{w_1|w_2|}{w_1^2 + w_2^2}\right](w_1 - 1) + [\sin(2\psi) + 2\psi]w_1(w_1 - 1) \\
&\quad - \left[-\frac{1}{2}\cos(2\psi) + \frac{1}{2} - \frac{w_1^2}{w_1^2 + w_2^2}\right]|w_2| + [\sin(2\psi) + 2\psi]w_2^2 \\
&= [\sin(2\psi) + 2\psi]\|\boldsymbol{w} - \boldsymbol{v}\|^2 - R_1 - R_2 ,
\end{aligned} \tag{13}$$

where

$$R_1 = \left[\left(\frac{\pi}{2} - \psi - \theta\right) - \frac{1}{2}\sin(2\psi)\right](w_1 - 1) \quad \text{and} \quad R_2 = \left[\frac{1}{2} - \frac{1}{2}\cos(2\psi) - \frac{w_1}{w_1^2 + w_2^2}\right]|w_2| .$$

According to Lemmas 15 and 16, Eq. (13) can be bounded by

$$\langle \nabla_{\boldsymbol{w}} L_{\mathrm{cr}}(\boldsymbol{w}, \psi), \boldsymbol{w} - \boldsymbol{v} \rangle \geqslant \frac{1}{2\pi} \left( \frac{1}{2} - \frac{1}{\pi} \right) [\sin(2\psi) + 2\psi] \|\boldsymbol{w} - \boldsymbol{v}\|^2 \geqslant \frac{1}{12\pi} [\sin(2\psi) + 2\psi] \|\boldsymbol{w} - \boldsymbol{v}\|^2 .$$
$$(14)$$

Secondly, we provide an upper bound of gradient in $D_2$. For any $(\boldsymbol{w}, \psi) \in D_2$, the gradient satisfies

$$4\pi^2 \|\nabla_{\boldsymbol{w}} L_{\mathrm{cr}}(\boldsymbol{w}, \psi)\|^2 = T_1 + T_2 ,$$

where

$$T_1 = \left( [\sin(2\psi) + 2\psi] w_1 - \frac{1}{2} \sin(2\psi) - \left( \frac{\pi}{2} + \psi - \theta \right) - \frac{w_1 |w_2|}{w_1^2 + w_2^2} \right)^2 ,$$

$$T_2 = \left( \left[ \frac{1}{2} \cos(2\psi) - \frac{1}{2} + \frac{w_1^2}{w_1^2 + w_2^2} \right] \mathrm{sgn}(w_2) + [\sin(2\psi) + 2\psi] w_2 \right)^2 .$$

From Lemmas 17 and 18, one knows

$$\|\nabla_{\boldsymbol{w}} L_{\mathrm{cr}}(\boldsymbol{w}, \psi)\|^2 \leqslant [\sin(2\psi) + 2\psi] \|\boldsymbol{w} - \boldsymbol{v}\|^2 . \tag{15}$$

Finally, based on Eqs. (14) and (15) and Lemma 13, we conclude

$$\|\boldsymbol{w}' - \boldsymbol{v}\| \leqslant \sqrt{1 - \left( \frac{1}{6\pi} - \eta \right) \eta [\sin(2\psi) + 2\psi]} \|\boldsymbol{w} - \boldsymbol{v}\| \leqslant \left( 1 - \frac{\eta}{24\pi} [\sin(2\psi) + 2\psi] \right) \|\boldsymbol{w} - \boldsymbol{v}\| ,$$

where the first inequality holds based on $\sqrt{1 - x} \leqslant 1 - x/2$ for any $x \in [0, 1]$ and $\eta \in (0, 1/(12\pi))$. Thus, we have completed the proof. $\qquad \square$

The following two lemmas depict the gradient with respect to $\psi$ in $D_1$ and $D_2$, respectively.

**Lemma 11.** *Let* $\psi' = \psi - \eta \nabla_\psi L_{\mathrm{cr}}(\boldsymbol{w}, \psi)$. *If* $(\boldsymbol{w}, \psi) \in D_1$, *then*

$$-\frac{1}{\pi} \left( \frac{\pi}{2} - \psi \right)^2 \leqslant \nabla_\psi L_{\mathrm{cr}}(\boldsymbol{w}, \psi) \leqslant -\frac{1 - R^2}{4\pi} \left( \frac{\pi}{2} - \psi \right)^2 .$$

**Proof.** For any $(\boldsymbol{w}, \psi) \in D_1$, one has

$$\nabla_\psi L_{\mathrm{cr}}(\boldsymbol{w}, \psi) = -\frac{1}{2\pi} [\cos(2\psi) + 1](1 - \|\boldsymbol{w} - \boldsymbol{v}\|^2) .$$

For any $\psi \in [0, \pi/2]$, we have $\frac{1}{2}(\pi/2 - \psi)^2 \leqslant \cos(2\psi) + 1 \leqslant 2(\pi/2 - \psi)^2$. Meanwhile, one has $0 \leqslant \|\boldsymbol{w}_t - \boldsymbol{v}\| \leqslant R$. Thus, the gradient with respect to $\psi$ can be bounded by

$$-\frac{1}{\pi} \left( \frac{\pi}{2} - \psi \right)^2 \leqslant \nabla_\psi L_{\mathrm{cr}}(\boldsymbol{w}, \psi) \leqslant -\frac{1 - R^2}{4\pi} \left( \frac{\pi}{2} - \psi \right)^2 ,$$

which completes the proof of the lower bound. $\qquad \square$

**Lemma 12.** *If* $(\boldsymbol{w}, \psi) \in D_2$, *then*

$$-2 \left( \frac{\pi}{2} - \psi \right)^2 - 2 \left( \frac{\pi}{2} - \psi \right) |w_2| \leqslant \nabla_\psi L_{\mathrm{cr}}(\boldsymbol{w}, \psi) \leqslant -\frac{1 - R^2}{2} \left( \frac{\pi}{2} - \psi \right)^2 .$$

**Proof.** The gradient of $L_{\mathrm{cr}}$ with respect to $\psi$ in $D_2$ can be calculated by

$$2\pi \nabla_\psi L_{\mathrm{cr}}(\boldsymbol{w}, \psi) = [1 + \cos(2\psi)] w_1^2 - [1 + \cos(2\psi)] w_1 + [1 + \cos(2\psi)] w_2^2 - \sin(2\psi) |w_2|$$
$$= [1 + \cos(2\psi)][\|\boldsymbol{w} - \boldsymbol{v}\|^2 - 1] + [1 + \cos(2\psi)] w_1 - \sin(2\psi) |w_2| . \tag{16}$$

Firstly, we prove the upper bound for $\nabla_\psi L_{\mathrm{cr}}(\boldsymbol{w}, \psi)$. It is observed that

$$[1 + \cos(2\psi)] w_1 - \sin(2\psi) |w_2| \leqslant 2 \cos \psi (w_1 \sin \theta - |w_2| \cos \theta) = 0 ,$$

where the first inequality holds based on $\pi/2 \geqslant \psi \geqslant \pi/2 - \theta \geqslant 0$, and the first equality holds from $w_1 = r \cos \theta$ and $|w_2| = r \sin \theta$. Substituting Eq. (24) into Eq. (16), we obtain

$$2\pi \nabla_\psi L_{\mathrm{cr}}(\boldsymbol{w}, \psi) \leqslant [1 + \cos(2\psi)][\|\boldsymbol{w} - \boldsymbol{v}\|^2 - 1] \leqslant -\frac{1 - R^2}{2} \left( \frac{\pi}{2} - \psi \right)^2 ,$$

where the second inequality holds according to $1 + \cos(2\psi) \geqslant \frac{1}{2}(\pi/2 - \psi)^2$ for any $\psi \in [0, \pi/2]$ and $\|\boldsymbol{w} - \boldsymbol{v}\| \leqslant R$.

Secondly, we verify the lower bound for $\nabla_\psi L_{\mathrm{cr}}(\boldsymbol{w}, \psi)$. It is observed that

$$2\pi \nabla_\psi L_{\mathrm{cr}}(\boldsymbol{w}, \psi) \geqslant -[1 + \cos(2\psi)] - \sin(2\psi)|w_2|$$
$$\geqslant -2\left(\frac{\pi}{2} - \psi\right)^2 - \sin(2\psi)|w_2|$$
$$\geqslant -2\left(\frac{\pi}{2} - \psi\right)^2 - 2\left(\frac{\pi}{2} - \psi\right)|w_2|,$$

where the first inequality holds because of $[1 + \cos(2\psi)]w_1 \geqslant 0$ and $\|\boldsymbol{w} - \boldsymbol{v}\| \geqslant 0$, the second inequality holds according to $1 + \cos(2\psi) \leqslant 2(\pi/2 - \psi)^2$, and the third inequality holds based on $\sin(2\psi) \leqslant \pi - 2\psi$ for $\psi \in [0, \pi/2]$. Thus, we have completed the proof. $\qquad\square$

## B.2 Convergence Rate Lemmas

The following lemma provides a sufficient condition for linear convergence of gradient descent.

**Lemma 13.** *If there exist two constants $c_1$ and $c_2$ such that*

$$\langle \nabla f(\boldsymbol{w}), \boldsymbol{w} - \boldsymbol{v} \rangle \geqslant c_1 \|\boldsymbol{w} - \boldsymbol{v}\|^2 \quad \text{and} \quad \|\nabla f(\boldsymbol{w})\|^2 \leqslant c_2 \|\boldsymbol{w} - \boldsymbol{v}\|^2,$$

*then $\boldsymbol{w}' = \boldsymbol{w} - \eta \nabla f(\boldsymbol{w})$ with $\eta \in (0, 2c_1/c_2)$ and $c = \sqrt{1 - 2c_1\eta + c_2\eta^2} \in (0, 1)$ satisfies*

$$\|\boldsymbol{w}' - \boldsymbol{v}\| \leqslant c\|\boldsymbol{w} - \boldsymbol{v}\|.$$

**Proof.** It is observed that

$$\|\boldsymbol{w}' - \boldsymbol{v}\|^2 = \|\boldsymbol{w} - \eta \nabla f(\boldsymbol{w}) - \boldsymbol{v}\|^2$$
$$= \|\boldsymbol{w} - \boldsymbol{v}\|^2 - 2\eta \langle \nabla f(\boldsymbol{w}), \boldsymbol{w} - \boldsymbol{v} \rangle + \eta^2 \|\nabla f(\boldsymbol{w})\|^2$$
$$\leqslant (1 - 2c_1\eta + c_2\eta^2)\|\boldsymbol{w} - \boldsymbol{v}\|^2.$$

For $\eta \in (0, 2c_1/c_2)$, the coefficient $1 - 2c_1\eta + c_2\eta^2$ is smaller than 1, which completes the proof. $\quad\square$

The following lemma gives a sufficient condition for convergence with an inversely proportional rate.

**Lemma 14.** *Let $\{a_t\}_{t=0}^\infty \subset [0, 1/2]$ represent a real-valued sequence.*

1. *If $a_{t+1} \leqslant a_t(1 - a_t)$, then $a_t \leqslant \frac{1}{t+1}$.*

2. *If $a_{t+1} \geqslant a_t(1 - a_t)$, then $a_t \geqslant \frac{a_0}{t+1}$.*

**Proof.** We prove the first conclusion by mathematical induction.

1. Base case. For $t = 0$, the conclusion holds from $a_0 \leqslant 1/2 \leqslant 1$.
2. Induction. Suppose that the conclusion holds for $t = k$ with $k \in \mathbb{N}$. Then it is observed that

$$a_{t+1} \leqslant \frac{1}{k+1}\left(1 - \frac{1}{k+1}\right) = \frac{k}{(k+1)^2} \leqslant \frac{1}{k+2},$$

where the first inequality holds from the induction hypothesis and the monotonicity of $x(1 - x)$ for $x \in [0, 1/2]$. Thus, the conclusion holds for $t = k + 1$.

Therefore, mathematical induction completes the proof of the first conclusion.

We proceed to verify the second conclusion by mathematical induction.

1. Base case. For $t = 0$, the conclusion holds from $a_0 \geqslant a_0$.
2. Induction. Suppose that the conclusion holds for $t = k$ with $k \in \mathbb{N}$. Then one has

$$a_{t+1} \geqslant \frac{a_0}{k+1}\left(1 - \frac{a_0}{k+1}\right) = \frac{a_0(k + 1 - a_0)}{(k+1)^2} \geqslant \frac{a_0}{k+2},$$

where the first inequality holds from the induction hypothesis and the monotonicity of $x(1 - x)$ for $x \in [0, 1/2]$, and the second inequality holds based on $a_0 \leqslant 1/2$. Thus, the conclusion holds for $t = k + 1$.

Therefore, mathematical induction completes the proof. $\qquad\square$

### B.3 Technical Lemmas

We present upper bounds for some small terms used in the proof.

**Lemma 15.** *Let* $R_1 = \left[\left(\frac{\pi}{2} - \psi - \theta\right) - \frac{1}{2}\sin(2\psi)\right](w_1 - 1)$. *If* $(\boldsymbol{w}, \psi) \in D_2$, *then*

$$R_1 \leqslant \frac{1}{2}[\sin(2\psi) + 2\psi]\|\boldsymbol{w} - \boldsymbol{v}\|^2 .$$

**Proof.** Let $r = \sqrt{w_1^2 + w_2^2}$ denote the norm of $\boldsymbol{w}$. Then according to the definition of $\theta$, one has $w_1 = r\cos\theta$ and $|w_2| = r\sin\theta$. Thus, we can rewrite $R_1$ as

$$R_1 = \left[\left(\frac{\pi}{2} - \psi - \theta\right) - \frac{1}{2}\sin(2\psi)\right](r\cos\theta - 1) .$$

We provide the upper bound for $R_1$ by discussion.

1. Suppose $r\cos\theta - 1 \geqslant 0$. Based on the definition of $D_2$, we have $\frac{\pi}{2} - \psi - \theta \leqslant 0$. Meanwhile, $\psi \in [0, \pi/2]$ indicates $\sin(2\psi) \geqslant 0$. Thus, one knows $R_1 \leqslant 0$.

2. Suppose $r\cos\theta - 1 < 0$. $R_1$ can be rewritten as

$$R_1 = \frac{1}{2}[\sin(2\psi) + 2\psi](1 - 2r\cos\theta + r^2) + \widetilde{R} , \tag{17}$$

   where

$$\widetilde{R} = \frac{1}{2}[\sin(2\psi) + 2\psi]r(\cos\theta - r) + \left(\frac{\pi}{2} - \theta\right)(r\cos\theta - 1) .$$

   If $\cos\theta - r \leqslant 0$, it is observed that $\widetilde{R} \leqslant 0$ because of $\psi, \theta \in [0, \pi/2]$ and $r\cos\theta - 1 < 0$. If $\cos\theta - r > 0$, then

$$\widetilde{R} \leqslant \frac{\pi}{2}r(\cos\theta - r) + \left(\frac{\pi}{2} - \theta\right)(r\cos\theta - 1) = -\frac{\pi}{2}r^2 + (\pi - \theta)\cos\theta r - \left(\frac{\pi}{2} - \theta\right) =: f(r) ,$$

   where the inequality holds since $\sin(2\psi) + 2\psi$ is monotonically increasing. The discriminant of $f$ is

$$\Delta(\theta) = (\pi - \theta)^2\cos^2\theta - \pi(\pi - 2\theta) \leqslant \frac{1}{\pi^2}\theta^2(\pi - 2\theta)(2\theta - 3\pi) ,$$

   where the first inequality holds since $\cos^2\theta \leqslant 1 - 4\theta^2/\pi^2$ on $[0, \pi/2]$. According to $\theta \in [0, \pi/2]$, one knows $\Delta(\theta) \leqslant 0$, which indicates $f(r) \leqslant 0$, and thus, $\widetilde{R} \leqslant 0$ when $\cos\theta - r \leqslant 0$. Combining the cases above, we obtain $\widetilde{R} \leqslant 0$, which, together with Eq. (17), implies $R_1 \leqslant \frac{1}{2}[\sin(2\psi) + 2\psi](1 - 2r\cos\theta + r^2)$.

Combining the cases above, one knows

$$R_1 \leqslant \frac{1}{2}[\sin(2\psi) + 2\psi](1 - 2r\cos\theta + r^2) = \frac{1}{2}[\sin(2\psi) + 2\psi]\|\boldsymbol{w} - \boldsymbol{v}\|^2 ,$$

which completes the proof. $\qquad\square$

**Lemma 16.** *Let* $R_2 = \left[\frac{1}{2} - \frac{1}{2}\cos(2\psi) - \frac{w_1}{w_1^2 + w_2^2}\right]|w_2|$. *If* $(\boldsymbol{w}, \psi) \in D_2$, *then*

$$R_2 \leqslant \frac{1}{\pi}[\sin(2\psi) + 2\psi]\|\boldsymbol{w} - \boldsymbol{v}\|^2 .$$

**Proof.** Let $r = \sqrt{w_1^2 + w_2^2}$ denote the norm of $\boldsymbol{w}$. Then according to the definition of $\theta$, one has $w_1 = r\cos\theta$ and $|w_2| = r\sin\theta$. Thus, we can rewrite $R_2$ as

$$R_2 = \left[\frac{r}{2}(1 - \cos(2\psi)) - \cos\theta\right]\sin\theta .$$

We provide the upper bound for $R_2$ by discussion.

1. Suppose $\frac{r}{2}[1 - \cos(2\psi)] - \cos\theta \leqslant 0$. From $\theta \in [0, \pi/2]$, we have $R_2 \leqslant 0$.

2. Suppose $\frac{r}{2}[1 - \cos(2\psi)] - \cos\theta > 0$. It is observed that $r < 2\cos\theta$ since $\|\boldsymbol{w} - \boldsymbol{v}\|^2 \leqslant r_0^2 < 1$ holds from the definition of $D_2$. Thus, the supposition indicates $\cos\theta < \frac{r}{2}[1 - \cos(2\psi)] < [1 - \cos(2\psi)]\cos\theta$, which, together with $\theta \in [0, \pi/2]$, implies $\psi \geqslant \pi/4$. It is observed that

$$f(r) = \frac{1}{2}(1 - 2r\cos\theta + r^2) - (r - \cos\theta)\sin\theta = \frac{1}{2}(r - \cos\theta - \sin\theta)^2 \geqslant 0 \,,$$

which indicates

$$\frac{1}{\pi}[\sin(2\psi) + 2\psi](1 - 2r\cos\theta + r^2) \geqslant \frac{1}{2}(1 - 2r\cos\theta + r^2) \geqslant (r - \cos\theta)\sin\theta \geqslant R_2 \,,$$

where the first inequality holds from $\psi \geqslant \pi/4$, and the third inequality holds because of $\cos(2\psi) \geqslant -1$.

Combining the cases above, we obtain

$$R_2 \leqslant \frac{1}{\pi}[\sin(2\psi) + 2\psi](1 - 2r\cos\theta + r^2) = \frac{1}{\pi}[\sin(2\psi) + 2\psi]\|\boldsymbol{w} - \boldsymbol{v}\|^2 \,,$$

which completes the proof. $\qquad\square$

**Lemma 17.** *Let* $T_1 = \left([\sin(2\psi) + 2\psi]w_1 - \frac{1}{2}\sin(2\psi) - \left(\frac{\pi}{2} + \psi - \theta\right) - \frac{w_1|w_2|}{w_1^2 + w_2^2}\right)^2$. *If* $(\boldsymbol{w}, \psi) \in D_2$, *then we have*

$$T_1 \leqslant 7\pi[\sin(2\psi) + 2\psi]\|\boldsymbol{w} - \boldsymbol{v}\|^2 \,.$$

**Proof.** It is observed that $T_1 = \left[[\sin(2\psi) + 2\psi](w_1 - 1) + T_{11} + T_{12}\right]^2$ with

$$T_{11} = \frac{1}{2}\sin(2\psi) + \left(\psi + \theta - \frac{\pi}{2}\right) \quad \text{and} \quad T_{12} = -\frac{w_1|w_2|}{w_1^2 + w_2^2} \,. \tag{18}$$

Firstly, denote by $r_0 \in (0, 1)$ a parameter determined later and we calculate an upper bound for $T_{11}$ by discussion.

1. Suppose $|w_1 - 1| + |w_2| \geqslant r_0$. Then one has

$$|T_{11}| \leqslant \frac{1}{2}\sin(2\psi) + \psi \leqslant \frac{1}{2r_0}[\sin(2\psi) + 2\psi][|w_1 - 1| + |w_2|] \,,$$

where the first inequality holds from $\theta \leqslant \frac{\pi}{2}$.

2. Suppose $|w_1 - 1| + |w_2| \leqslant r_0$. Then it is observed that $w_1 \geqslant 1 - r_0 + |w_2| \geqslant 0$. Thus,

$$r = \sqrt{w_1^2 + w_2^2} \geqslant \sqrt{(1 - r_0)^2 + 2|w_2|(|w_2| + 1 - r_0)} \geqslant 1 - r_0 \,,$$

where the second inequality holds because of $r_0 \leqslant 1$. Then we can bound $|w_2|$ from below as

$$|w_2| = r\sin\theta \geqslant (1 - r_0)\sin\theta \geqslant \frac{1 - r_0}{2}\theta \,, \tag{19}$$

where the second inequality holds since $\theta \leqslant 2\sin\theta$ for all $\theta \in [0, \pi/2]$. Meanwhile, we bound $\theta$ from above as

$$\theta \leqslant \tan\theta = \frac{|w_2|}{w_1} \leqslant \left(\frac{1 - r_0}{|w_2|} + 1\right)^{-1} \leqslant \left(\frac{1 - r_0}{r_0} + 1\right)^{-1} = r_0 \,, \tag{20}$$

where the second inequality holds from $w_1 \geqslant 1 - r_0 + |w_2|$, and the third inequality holds based on $|w_2| \leqslant r_0$. Then we obtain an upper bound of $T_{11}$ as follows

$$|T_{11}| \leqslant \theta \leqslant \frac{2|w_2|}{1 - r_0} \leqslant \frac{4\psi|w_2|}{(1 - r_0)(\pi - 2r_0)} \leqslant \frac{2}{(1 - r_0)(\pi - 2r_0)}[\sin(2\psi) + 2\psi][|w_1 - 1| + |w_2|] \,,$$

where the first inequality holds from the monotonicity of $\frac{1}{2}\sin(2\psi) + \psi$ and $\psi \leqslant \frac{\pi}{2}$, the second inequality holds from Eq. (19), and the third inequality holds based on $\psi \geqslant \frac{\pi}{2} - \theta$ and Eq. (20).

Combining the cases above, we have proven

$$|T_{11}| \leqslant \max\left\{\frac{1}{2r_0}, \frac{2}{(1-r_0)(\pi - 2r_0)}\right\}[\sin(2\psi) + 2\psi][|w_1 - 1| + |w_2|] .$$

Choosing $r_0 = \frac{1}{4}\left[\pi + 6 - \sqrt{pi^2 + 4\pi + 36}\right]$, we obtain an upper bound of $T_{11}$ as follows

$$|T_{11}| \leqslant \frac{3}{2}[\sin(2\psi) + 2\psi][|w_1 - 1| + |w_2|] . \tag{21}$$

Secondly, we provide an upper bound for $T_{12}$. We claim and prove by discussion that

$$|w_2| \leqslant 2\sqrt{w_1^2 + w_2^2}(|w_1 - 1| + |w_2|) . \tag{22}$$

1. Suppose $w_1 \leqslant 1/2$. Then it is observed that $|w_1 - 1| \geqslant 1/2$, which implies

$$|w_2| \leqslant \sqrt{w_1^2 + w_2^2} \leqslant \sqrt{w_1^2 + w_2^2} \cdot 2|w_1 - 1| \leqslant 2\sqrt{w_1^2 + w_2^2}(|w_1 - 1| + |w_2|) .$$

2. Suppose $w_1 \geqslant 1/2$. Then one has $\sqrt{w_1^2 + w_2^2} \geqslant 1/2$, which indicates

$$|w_2| \leqslant |w_1 - 1| + |w_2| \leqslant 2\sqrt{w_1^2 + w_2^2}(|w_1 - 1| + |w_2|) .$$

From the definition of $D_2$, one has $\frac{\pi}{2} \geqslant \psi \geqslant \frac{\pi}{2} - \theta \geqslant 0$, which indicates

$$\psi \geqslant \sin\psi \geqslant \sin\left(\frac{\pi}{2} - \theta\right) = \cos\theta = \frac{w_1}{\sqrt{w_1^2 + w_2^2}} . \tag{23}$$

Then we obtain an upper bound of $|T_{12}|$ as

$$|T_{12}| \leqslant \frac{2w_1}{\sqrt{w_1^2 + w_2^2}}(|w_1 - 1| + |w_2|) \leqslant 2\psi(|w_1 - 1| + |w_2|) \leqslant [\sin(2\psi) + 2\psi](|w_1 - 1| + |w_2|), \tag{24}$$

where the first inequality holds according to Eq. (22), and the second inequality holds based on Eq. (23). Finally, combining Eqs. (21) and (24), we conclude

$$T_1 \leqslant \left[\left|[\sin(2\psi) + 2\psi](w_1 - 1)\right| + \max\{|T_{11}|, |T_{12}|\}\right]^2 \leqslant 7\pi[\sin(2\psi) + 2\psi]\|\boldsymbol{w} - \boldsymbol{v}\|^2 ,$$

where the first inequality holds based on $T_{11} \geqslant 0$ and $T_{12} \leqslant 0$, and the second inequality holds because of $\sin(2\psi) + 2\psi \leqslant \pi$ for any $\psi \in [0, \pi/2]$. Thus, we have completed the proof. $\square$

**Lemma 18.** *Let* $T_2 = \left(\left[\frac{1}{2}\cos(2\psi) - \frac{1}{2} + \frac{w_1^2}{w_1^2 + w_2^2}\right]\text{sgn}(w_2) + [\sin(2\psi) + 2\psi]w_2\right)^2$. *If* $(\boldsymbol{w}, \psi) \in D_2$, *then we have*

$$T_2 \leqslant 7\pi[\sin(2\psi) + 2\psi]\|\boldsymbol{w} - \boldsymbol{v}\|^2 .$$

**Proof.** From $\cos\theta = w_1/\sqrt{w_1^2 + w_2^2}$, one has $\cos(\pi - 2\theta) = 1 - 2\cos^2\theta = 1 - 2w_1^2/(w_1^2 + w_2^2)$. Thus, we have

$$\left|\left[\frac{1}{2}\cos(2\psi) - \frac{1}{2} + \frac{w_1^2}{w_1^2 + w_2^2}\right]\text{sgn}(w_2)\right| = \frac{1}{2}|\cos(2\psi) - \cos(\pi - 2\theta)| \leqslant \psi + \theta - \frac{\pi}{2} \leqslant T_{11} ,$$

where the first inequality holds because of $|\cos a - \cos b| \leqslant |a - b|$, and the second inequality holds based on the definition of $T_{11}$ in Eq. (18) and $\sin(2\psi) \geqslant 0$. Recalling the upper bound of $T_{11}$ in Eq. (21), we obtain

$$T_2 \leqslant \left(\left|\left[\frac{1}{2}\cos(2\psi) - \frac{1}{2} + \frac{w_1^2}{w_1^2 + w_2^2}\right]\text{sgn}(w_2)\right| + |[\sin(2\psi) + 2\psi]w_2|\right)^2$$
$$\leqslant 7\pi[\sin(2\psi) + 2\psi]\|\boldsymbol{w} - \boldsymbol{v}\|^2 ,$$

which completes the proof. $\square$

# C  Proof of Theorem 2

In the main part of this section, we present the closed form of the loss, definition and properties of the ideal region, and the detailed proof of Theorem 2. Subsection C.1 provides the optimization behaviors. Subsection C.2 gives some convergence rate lemmas.

According to Lemma 8, the expected square loss $L_{cc}$ can be calculated by

$$L_{cc}(\boldsymbol{w}, \psi_w) = \frac{1}{2}B(\boldsymbol{w}, \boldsymbol{w}, \psi_w, \psi_w) - B(\boldsymbol{w}, \boldsymbol{v}, \psi_w, \psi_v) + \frac{1}{2}B(\boldsymbol{v}, \boldsymbol{v}, \psi_v, \psi_v) \ . \tag{25}$$

For $R \in (0, 1)$, $\psi_l \in [0, \delta_l]$, and $\psi_u \in [\pi/2 - \delta_u, \pi/2]$, define

$$D_1 = \{(\boldsymbol{w}, \psi_w) \mid \|\boldsymbol{w} - \boldsymbol{v}\|_\infty \leqslant R, \psi_w \in [\psi_l, \psi_u], \theta_{\boldsymbol{w}, \boldsymbol{v}} \in [0, |\psi_w - \psi_v|]\} \ ,$$
$$D_2 = \{(\boldsymbol{w}, \psi_w) \mid \|\boldsymbol{w} - \boldsymbol{v}\|_\infty \leqslant R, \psi_w \in [\psi_l, \psi_u], \theta_{\boldsymbol{w}, \boldsymbol{v}} \in (|\psi_w - \psi_v|, \psi_w + \psi_v)\} \ .$$

Let $D = D_1 \cup D_2$ indicate the ideal region, i.e.,

$$D = \{(\boldsymbol{w}, \psi_w) \mid \|\boldsymbol{w} - \boldsymbol{v}\|_\infty \leqslant R, \psi_w \in [\psi_l, \psi_u], \theta_{\boldsymbol{w}, \boldsymbol{v}} \in [0, \psi_w + \psi_v]\} \ .$$

By spherical symmetry, we assume $\boldsymbol{v} = (1, 0)$ without loss of generality in the rest proof. For conciseness, define $s_w = \sin(2\psi_w) + 2\psi_w$ and $s_v = \sin(2\psi_v) + 2\psi_v$. The following lemma discusses the properties of the ideal region, concerning the closeness of the region under gradient descent and the probability that an initialization falls into this region.

**Lemma 19.** *Let $\psi_v \in [7\pi/20, 2\pi/5]$. If we choose the parameters as*

$$R = \frac{1}{25} \ , \quad \psi_l = \psi_v - \frac{109}{100}R \ , \quad \psi_u = \psi_v + \frac{109}{100}R \ , \quad and \quad 0 < \eta \leqslant \frac{1}{120}R \ ,$$

*then all conditions in Lemmas 20-25 are satisfied. If $\boldsymbol{w}_0 \sim \mathcal{N}(0, I_2)$ and $\psi_{w,0} \sim \mathcal{U}(0, \pi/2)$, then*

$$\Pr[(\boldsymbol{w}_0, \psi_{w,0}) \in D] \geqslant 10^{-5} \ .$$

**Proof.** We first prove that all conditions in the lemmas are satisfied.

- Lemma 20. It is observed that the first condition holds from

$$\eta \leqslant \frac{1}{120}R = \frac{1}{120} \cdot \frac{1}{25} < 2 \ .$$

  According to $\psi_u > \psi_v > \pi/4$, we have $\psi_v \sin(2\psi_u) \leqslant \psi_u \sin(2\psi_v)$, which implies

$$s_v \geqslant \frac{\psi_v s_u}{\psi_u} = \frac{\psi_v s_u}{\psi_v + 109R/100} \geqslant \frac{7\pi s_u/20}{7\pi/20 + 109R/100} \geqslant (1 - R)s_u \geqslant (1 - R)s_w \ ,$$

  where the fourth inequality holds since $s_w$ is monotonic. Thus, the second condition is satisfied.

- Lemma 21. The first condition $\eta < 2$ has been satisfied above. It is observed that $\psi_l \geqslant 7\pi/20 - 109R/100$. Thus, The second condition holds from $\psi_l/20 \geqslant 7\pi/400 - 109R/2000 \geqslant R$. The third condition holds since

$$\max\{\psi_u - \psi_v, \psi_v - \psi_l\} = \frac{109R}{100} \leqslant \frac{5R\psi_l}{3} \ .$$

- Lemma 22. The only condition $\eta < 2$ has been satisfied.

- Lemma 23. The first condition holds because of $R = 1/25 \leqslant 1/2$. The second condition holds based on $\cos^2 \psi_v \geqslant \cos^2(2\pi/5) \geqslant 1/25$. The third condition holds from $\eta \leqslant R/120 \leqslant 3R/2$.

- Lemma 24. The first condition $R \leqslant 1/2$ has been satisfied above. The second and third conditions hold because of

$$\frac{\pi}{3}\min\{\psi_u - \psi_v, \psi_v - \psi_l\} = \frac{\pi}{3} \cdot \frac{109R}{100} \geqslant \frac{R}{120} \geqslant \eta \ .$$

- Lemma 25. The first condition $R \leqslant 1/2$ has been satisfied above. The second one holds from

$$\arcsin R + 9\eta \leqslant \frac{101R}{100} + \frac{3R}{40} \leqslant \frac{109R}{100} = \psi_u - \psi_v \ .$$

We then prove the second conclusion. Let $p_0 = \Pr[(\boldsymbol{w}_0, \psi_{w,0}) \in D]$ for simplicity. Then we have

$$
\begin{aligned}
p_0 &= \Pr[\psi_l \leqslant \psi_{w,0} \leqslant \psi_u] \cdot \Pr[1 - R \leqslant w_1 \leqslant 1 + R] \cdot \Pr[-R \leqslant w_2 \leqslant R] \\
&= \frac{109R}{50} \cdot \frac{1}{2}[\mathrm{erf}(1 + R) - \mathrm{erf}(1 - R)] \cdot \mathrm{erf}(R) \\
&\geqslant 10^{-5} \,,
\end{aligned}
$$

where $\mathrm{erf}(x)$ denotes the error function. Thus, we have completed the proof. $\qquad\square$

We are now ready to prove Theorem 2.

**Proof of Theorem 2.** Let $R$, $\psi_l$, and $\psi_u$ be the same as those in Lemma 19. Suppose that $(\boldsymbol{w}_0, \psi_{w,0}) \in D$. Then Lemma 19 implies $(\boldsymbol{w}_t, \psi_{w,t}) \in D$ for any $t \in \mathbb{N}$. The proof of convergence is divided into several stages.

**Step 1: $w_2$ converges to $0$.** In stage I, we consider the convergence of $w_{2,t}$ when $(\boldsymbol{w}_0, \psi_{w,0}) \in D$. From Lemmas 22 and 23, the optimization behaviors of $w_2$ is the combination of minimizing a contraction mapping or an almost absolute function. Thus, Lemma 26 with $r_1 = r_2 = R$, $c_3 = s_w/(2\pi)$, $g_l = (\cos^2 \psi_v - \sqrt{2}R)/(2\pi)$, and $g_u = 2/3$ implies

$$
|w_2| \leqslant \frac{c_2^2(\cos^2 \psi_v - \sqrt{2}R)}{4\pi c_1 t} \leqslant \frac{c_2^2}{4\pi c_1 t} \quad \text{for} \quad t \in \mathbb{N}^+ \,. \tag{26}
$$

**Step 2: $\psi_w$ converges to $\psi_v$.** In stage II, we prove the convergence of $\psi_{w,t}$ when $(\boldsymbol{w}_0, \psi_{w,0}) \in D$. From Lemmas 24 and 25, the convergence of $\psi_w$ is limited by that of $w_2$, i.e., $\psi_w$ tends to the global minimum with constant-order gradient when the error of $\psi_w$ is larger than that of $w_2$, while becomes far away from the global minimum otherwise. Then Lemma 27 with $r_1 = r_2 = 109R/100$, $a = c_2^2(\cos^2 \psi_v - \sqrt{2}R)/(4\pi c_1)$, $g_l = \cos^2 \psi_u/(4\pi)$, and $g_u = 9$ indicates

$$
|\psi_w - \psi_v| \leqslant \left[\frac{c_2^2(\cos^2 \psi_v - \sqrt{2}R)}{4\pi c_1} + 9c_2\right]\frac{1}{t} \leqslant \frac{10c_2^2}{c_1 t} \quad \text{for} \quad t \in \mathbb{N}^+ \,. \tag{27}
$$

**Step 3: $w_1$ converges to $1$.** In stage III, we investigate the convergence of $w_{1,t}$ when $(\boldsymbol{w}_0, \psi_{w,0}) \in D$. From Lemmas 20 and 21, the gradient points to the global minimum with a remainder controlled by the error of $w_1$ and $\psi_w$. Then Lemma 28 with $d_l = 1/4$, $d_u = 1/2$, and $e = 20c_2^2/(\pi c_1)$ leads to

$$
|w_1 - 1| \leqslant \frac{20c_2^3}{\pi c_1 t} \quad \text{for} \quad t \in \mathbb{N}^+ \,. \tag{28}
$$

**Step 3: the expected loss converges to $0$.** We now estimate the convergence of the expected square loss when $(\boldsymbol{w}_0, \psi_{w,0}) \in D$. For any $(\boldsymbol{w}, \psi_w) \in D$, define non-negative quantities $\Delta_{\boldsymbol{w}} = \|\boldsymbol{w} - \boldsymbol{v}\|$ and $\Delta_\psi = |\psi_w - \psi_v|$. We provide an upper bound for $L_{\mathrm{cc}}$ by discussion.

1. Suppose $(\boldsymbol{w}, \psi_w) \in D_1$. Then we have

$$
\begin{aligned}
4\pi L_{\mathrm{cc}}(\boldsymbol{w}, \psi_w) &= \|\boldsymbol{w}\|^2 s_w - 2\|\boldsymbol{w}\|\|\boldsymbol{v}\| \cos \theta_{\boldsymbol{w},\boldsymbol{v}} s_m + \|\boldsymbol{v}\|^2 s_v \\
&\leqslant \|\boldsymbol{w}\|^2(s_v + s_\Delta) - 2\|\boldsymbol{w}\|\|\boldsymbol{v}\|(1 - \Delta_{\boldsymbol{w}}^2)(s_v - s_\Delta) + \|\boldsymbol{v}\|^2 s_v \\
&\leqslant 4(\|\boldsymbol{w}\|^2 + 2\|\boldsymbol{w}\|\|\boldsymbol{v}\|)\Delta_\psi + (s_v + 2\|\boldsymbol{w}\|\|\boldsymbol{v}\|)\Delta_{\boldsymbol{w}}^2 \\
&\leqslant 32\Delta_\psi + 8\Delta_{\boldsymbol{w}}^2 \,,
\end{aligned}
$$

where the first inequality holds from $s_w \leqslant s_v + s_\Delta$, $\cos \theta_{\boldsymbol{w},\boldsymbol{v}} \geqslant \sqrt{1 - \Delta_{\boldsymbol{w}}^2} \geqslant 1 - \Delta_{\boldsymbol{w}}^2$, and $s_m \geqslant s_v - s_\Delta$ with $s_\Delta = 2\Delta_\psi + \sin(2\Delta_\psi)$, the second inequality holds since $|\|\boldsymbol{w}\| - \|\boldsymbol{v}\|| \leqslant \Delta_{\boldsymbol{w}}^2$ and $s_\Delta \leqslant 4\Delta_\psi$, and the third inequality holds based on $\|\boldsymbol{w}\| \leqslant 2$ and $s_v \leqslant \pi$.

2. Suppose $(\boldsymbol{w}, \psi_w) \in D_2$. Let $\theta = \theta_{\boldsymbol{w},\boldsymbol{v}}$. Then one knows

$$
\begin{aligned}
4\pi L_{\mathrm{cc}}(\boldsymbol{w}, \psi_w) &= \|\boldsymbol{w}\|^2 s_w + \|\boldsymbol{v}\|^2 s_v \\
&\quad - \|\boldsymbol{w}\|\|\boldsymbol{v}\|[2(\psi_w + \psi_v - \theta)\cos\theta + \sin(2\psi_w - \theta) + \sin(2\psi_v - \theta)] \\
&= s_v(\|\boldsymbol{w}\| - \|\boldsymbol{v}\|)^2 + (\|\boldsymbol{w}\|^2 - \|\boldsymbol{w}\|\|\boldsymbol{v}\|\cos\theta)(s_w - s_v) \\
&\quad + \|\boldsymbol{w}\|\|\boldsymbol{v}\|\theta\cos\theta + 2\|\boldsymbol{w}\|\|\boldsymbol{v}\|s_v(1 - \cos\theta) \,.
\end{aligned}
$$

Then according to $\big|\|\boldsymbol{w}\| - \|\boldsymbol{v}\|\big| \leqslant \Delta_{\boldsymbol{w}}$, $s_w - s_v \leqslant 4\Delta_\psi$, $\theta \leqslant \arcsin \Delta_{\boldsymbol{w}} \leqslant 2\Delta_{\boldsymbol{w}}$, and $\cos\theta \geqslant 1 - \Delta_{\boldsymbol{w}}^2$, we have

$$4\pi L_{\mathrm{cc}} \leqslant 4\big|\|\boldsymbol{w}\|^2 - \|\boldsymbol{w}\|\|\boldsymbol{v}\|\cos\theta\big|\Delta_\psi + 2\|\boldsymbol{w}\|\|\boldsymbol{v}\|\cos\theta\Delta_{\boldsymbol{w}} + (1 + 2\|\boldsymbol{w}\|\|\boldsymbol{v}\|)s_v\Delta_{\boldsymbol{w}}^2$$
$$\leqslant 16\Delta_\psi + 5\Delta_{\boldsymbol{w}}\,,$$

where the second inequality hodls based on $\|\boldsymbol{w}\| \leqslant 2$, $s_v \leqslant \pi$, and $\Delta_{\boldsymbol{w}} \leqslant \sqrt{2}R = \sqrt{2}/25$.

Combining the cases above, one knows from $\Delta_{\boldsymbol{w}} \leqslant 5/8$ that for any $(\boldsymbol{w}, \psi_w) \in D$, the loss satisfies

$$L_{\mathrm{cc}}(\boldsymbol{w}, \psi_w) \leqslant 32\Delta_\psi + 5\Delta_{\boldsymbol{w}}\,.$$

Then based on $(\boldsymbol{w}_t, \psi_{w,t}) \in D$ and Eqs. (26)-(28), we obtain from $c_2 \geqslant 1$ that

$$L_{\mathrm{cc}}(\boldsymbol{w}_t, \psi_{w,t}) \leqslant \frac{320c_2^2}{c_1 t} + \frac{5c_2^2}{4\pi c_1 t} + \frac{100c_2^3}{\pi c_1 t} \leqslant \frac{400c_2^3}{c_1 t}\,,$$

which holds with probability at least $10^{-5}$ from Lemma 19. Thus, we have completed the proof. $\square$

## C.1 Optimization behaviors

The following two lemmas consider the gradient with respect to $w_1$ in $D_1$ and $D_2$, respectively.

**Lemma 20.** *Let $w_1 = w_1 - \eta\nabla_{w_1} L_{\mathrm{cc}}(\boldsymbol{w}, \psi_w)$ with $(\boldsymbol{w}, \psi_w) \in D_1$. If $\eta \in (0, 2)$ and $(1-R)s_w \leqslant s_v$, then we have*

$$\nabla_{w_1} L_{\mathrm{cc}}(\boldsymbol{w}, \psi_w) = \frac{s_w}{2\pi}(w_1 - 1) + \frac{1}{2\pi}[s_w - \min\{s_w, s_v\}] \quad \text{and} \quad |w_1' - 1| \leqslant R\,.$$

**Proof.** For any $(\boldsymbol{w}, \psi_w) \in D_1$, one has

$$\nabla_{w_1} L_{\mathrm{cc}}(\boldsymbol{w}, \psi_w) = \frac{s_w}{2\pi}[w_1 - \min\{s_w, s_v\}] = \frac{s_w}{2\pi}(w_1 - 1) + r\,, \tag{29}$$

where $r$ denotes a remainder defined by $r = \frac{1}{2\pi}[s_w - \min\{s_w, s_v\}]$. Then Eq. (29) implies

$$|w_1' - 1| \leqslant \left|1 - \frac{\eta s_w}{2\pi}\right||w_1 - 1| + |\eta r| \leqslant \left(1 - \frac{\eta s_w}{2\pi}\right)R + \frac{\eta}{2\pi}[s_w - \min\{s_w, s_v\}]\,, \tag{30}$$

where the first inequality holds from the triangle inequality, and the second inequality holds based on $1 - \eta s_w/(2\pi) \geqslant 0$ and $|w_1 - 1| \leqslant R$. We proceed to complete the proof by discussion.

- Suppose that $\min\{s_w, s_v\} = s_w$. Then Eq. (30) implies

$$|w_1' - 1| \leqslant \left(1 - \frac{\eta s_w}{2\pi}\right)R \leqslant R\,,$$

where the second inequality holds from $\eta > 0$ and $s_w \geqslant 0$.

- Suppose that $\min\{s_w, s_v\} = s_v$. Then one knows from Eq. (30) that

$$|w_1' - 1| \leqslant \left(1 - \frac{\eta s_w}{2\pi}\right)R + \frac{\eta(s_w - s_v)}{2\pi} \leqslant R\,,$$

where the second inequality holds because of $(1 - R)s_w \leqslant s_v$.

Combining the cases above completes the proof. $\square$

**Lemma 21.** *Let $w_1 = w_1 - \eta\nabla_{w_1} L_{\mathrm{cc}}(\boldsymbol{w}, \psi_w)$ with $(\boldsymbol{w}, \psi_w) \in D_2$. If $\eta \in (0, 2)$, $R \leqslant \psi_l/20$ and $\max\{\psi_u - \psi_v, \psi_v - \psi_l\} \leqslant 5R\psi_l/3$, then we have*

$$\nabla_{w_1} L_{\mathrm{cc}}(\boldsymbol{w}, \psi_w) = \frac{s_w - \theta_{\boldsymbol{w},\boldsymbol{v}}}{2\pi}(w_1 - 1) + \frac{1}{4\pi}[(s_w - s_v) + 2(\theta_{\boldsymbol{w},\boldsymbol{v}} - \sin\theta_{\boldsymbol{w},\boldsymbol{v}})] \quad \text{and} \quad |w_1' - 1| \leqslant R\,.$$

**Proof.** For any $(\boldsymbol{w}, \psi_w) \in D_2$, the gradient of $L_{\mathrm{cc}}$ with respect to $w_1$ can be calculated by

$$\nabla_{w_1} L_{\mathrm{cc}} = \frac{s_w - \theta_{\boldsymbol{w},\boldsymbol{v}}}{2\pi}(w_1 - 1) + \frac{1}{4\pi}[(s_w - s_v) + 2(\theta_{\boldsymbol{w},\boldsymbol{v}} - \sin\theta_{\boldsymbol{w},\boldsymbol{v}})] = \frac{s_w - \theta_{\boldsymbol{w},\boldsymbol{v}}}{2\pi}(w_1 - 1) + r\,,$$

where $r$ denotes a remainder defined by $r = [(s_w - s_v) + 2(\theta_{\boldsymbol{w},\boldsymbol{v}} - \sin\theta_{\boldsymbol{w},\boldsymbol{v}})]/(4\pi)$. Then we have

$$|w_1' - 1| \leqslant \left|1 - \frac{\eta(s_w - \theta_{\boldsymbol{w},\boldsymbol{v}})}{2\pi}\right| |w_1 - 1| + |\eta r| \leqslant R + \eta \left[|r| - \frac{R(s_w - \theta_{\boldsymbol{w},\boldsymbol{v}})}{2\pi}\right], \qquad (31)$$

where the first inequality holds from the triangle inequality, and the second inequality holds based on $\eta(s_w - \theta_{\boldsymbol{w},\boldsymbol{v}}) \leqslant \eta s_w \leqslant 2\pi$ and $|w_1 - 1| \leqslant R$. It is observed that

$$s_w - \theta_{\boldsymbol{w},\boldsymbol{v}} \geqslant \frac{7}{2}\psi_l - \theta_{\boldsymbol{w},\boldsymbol{v}} \geqslant \frac{7}{2}\psi_l - 2R, \qquad (32)$$

where the first inequality holds based on $s_w \geqslant 2\psi_l + \sin(2\psi_l)$ and $\sin\psi_l \geqslant 3\psi_l/4$ for $\psi_l \leqslant \pi/4$, and the second inequality holds from $\theta_{\boldsymbol{w},\boldsymbol{v}} \leqslant \arcsin R \leqslant 2R$. Meanwhile, one has

$$|r| \leqslant \frac{1}{4\pi}|s_w - s_v| + \frac{1}{2\pi}|\theta_{\boldsymbol{w},\boldsymbol{v}} - \sin\theta_{\boldsymbol{w},\boldsymbol{v}}| \leqslant \frac{\max\{\psi_u - \psi_v, \psi_v - \psi_l\}}{\pi} + \frac{2R^3}{3\pi}, \qquad (33)$$

where the first inequality holds from the triangle inequality, and the second inequality holds according to the 4-Lipschitzness of $2\theta + \sin(2\theta)$, $\theta - \sin\theta \leqslant \theta^3/6$ for any $\theta \geqslant 0$, and $\theta_{\boldsymbol{w},\boldsymbol{v}} \leqslant 2R$. Substituting Eqs. (32) and (33) into Eq. (31), we obtain

$$|w_1' - 1| \leqslant R + \frac{\eta}{12\pi}\left[12\max\{\psi_u - \psi_v, \psi_v - \psi_l\} + 8R^3 + 12R^2 - 21R\psi_l\right] \leqslant R,$$

where the second inequality holds from $\max\{\psi_u - \psi_v, \psi_v - \psi_l\} \leqslant 5R\psi_l/3$ and $R \leqslant \psi_l/20 \leqslant 1$. Thus, we have completed the proof. $\qquad\square$

The following two lemmas focus on the gradient with respect to $w_2$ in $D_1$ and $D_2$, respectively.

**Lemma 22.** *Let $w_2' = w_2 - \eta\nabla_{w_2}L_{cc}(\boldsymbol{w}, \psi_w)$ with $(\boldsymbol{w}, \psi_w) \in D_1$. If $\eta \in (0, 2)$, then we have*

$$|w_2'| \leqslant \left(1 - \frac{\eta s_w}{2\pi}\right)|w_2| \quad and \quad |w_2'| \leqslant R.$$

**Proof.** For any $(\boldsymbol{w}, \psi_w) \in D_1$, one has $\nabla_{w_2}L_{cc}(\boldsymbol{w}, \psi_w) = \frac{s_w w_2}{2\pi}$. Thus, we have

$$w_2' = \left(1 - \frac{\eta s_w}{2\pi}\right)w_2. \qquad (34)$$

According to $s_w \in [0, \pi]$ and $\eta \in (0, 2)$, the coefficient $1 - \eta s_w/(2\pi)$ is positive and smaller than 1. Based on $(\boldsymbol{w}, \psi_w) \in D_1$, one knows $|w_2| \leqslant R$. Then Eq. (34) implies

$$|w_2'| = \left(1 - \frac{\eta s_w}{2\pi}\right)|w_2| \leqslant R,$$

which completes the proof. $\qquad\square$

**Lemma 23.** *Let $w_2' = w_2 - \eta\nabla_{w_2}L_{cc}(\boldsymbol{w}, \psi_w)$ with $(\boldsymbol{w}, \psi_w) \in D_2$. If $R \leqslant 1/2$, $\sqrt{2}R \leqslant \cos^2\psi_v$, and $\eta \leqslant 3R/2$, then we have*

$$\frac{\cos^2\psi_v - \sqrt{2}R}{2\pi} \leqslant \nabla_{w_2}L_{cc}(\boldsymbol{w}, \psi_w)\mathrm{sgn}(w_2) \leqslant \frac{2}{3} \quad and \quad |w_2'| \leqslant R.$$

**Proof.** For any $(\boldsymbol{w}, \psi_w) \in D_2$, the gradient of $L_{cc}$ with respect to $w_2$ can be calculated by

$$\nabla_{w_2}L_{cc}(\boldsymbol{w}, \psi_w) = \frac{1}{2\pi}s_w w_2 + \frac{1}{4\pi}\left(\cos(2\psi_w) + \cos(2\psi_v) + \frac{2w_1^2}{\sqrt{w_1^2 + w_2^2}}\right)\mathrm{sgn}(w_2). \qquad (35)$$

Since $(\boldsymbol{w}, \psi_w) \in D_2$, one knows that $|w_1 - 1| \leqslant R$ and $|w_2| \leqslant R$. Thus, we have

$$2(1 - \sqrt{2}R) \leqslant \frac{2(1 - R)^2}{\sqrt{(1 - R)^2 + R^2}} \leqslant \frac{2w_1^2}{\sqrt{w_1^2 + w_2^2}} \leqslant 2(1 + R),$$

where the first inequality holds because of $R \in [0, 1/2]$. Then we have

$$\cos(2\psi_w) + \cos(2\psi_v) + \frac{2w_1^2}{\sqrt{w_1^2 + w_2^2}} \leqslant 1 + \cos(2\psi_v) + 2(1 + R) \leqslant 5, \qquad (36)$$

where the second inequality holds based on $R \leqslant 1/2$. Meanwhile, one has

$$\cos(2\psi_w) + \cos(2\psi_v) + \frac{2w_1^2}{\sqrt{w_1^2 + w_2^2}} \geqslant -1 + \cos(2\psi_v) + 2(1 - \sqrt{2}R) = 2(\cos^2\psi_v - \sqrt{2}R) . \quad (37)$$

It is observed that $0 \leqslant s_w|w_2| \leqslant \frac{\pi}{2}$ since $s_w \in [0, \pi]$ and $|w_2| \leqslant R \leqslant \frac{1}{2}$. Then substituting Eqs. (36) and (37) into Eq. (35), we obtain

$$\frac{\cos^2\psi_v - \sqrt{2}R}{2\pi} \leqslant \nabla_{w_2} L_{\mathrm{cc}}(\boldsymbol{w}, \psi_w)\mathrm{sgn}(w_2) \leqslant \frac{1}{4} + \frac{5}{4\pi} \leqslant \frac{2}{3} .$$

Thus, one knows from Eq. (35) that

$$|w_2'| = \big||w_2| - \eta\nabla_{w_2}L_{\mathrm{cc}}(\boldsymbol{w}, \psi_w)\mathrm{sgn}(w_2)\big| \leqslant \max\{|w_2|, \eta\nabla_{w_2}L_{\mathrm{cc}}(\boldsymbol{w}, \psi_w)\mathrm{sgn}(w_2)\} \leqslant R ,$$

where the first inequality holds from $|a - b| \leqslant \max\{a, b\}$ for non-negative numbers $a$ and $b$, and the second inequality holds based on $|w_2| \leqslant R$ and $\eta \leqslant 3R/2$. Thus, we have completed the proof. $\square$

The following two lemmas investigate the gradient with respect to $\psi_w$ in $D_1$ and $D_2$, respectively.

**Lemma 24.** *Let* $\psi_w' = \psi_w - \eta\nabla_{\psi_w}L_{\mathrm{cc}}(\boldsymbol{w}, \psi_w)$ *with* $(\boldsymbol{w}, \psi_w) \in D_1$. *If* $R \leqslant 1/2$, $\eta \leqslant \pi(\psi_u - \psi_v)/3$, *and* $\eta \leqslant \pi(\psi_v - \psi_l)/3$, *then we have*

$$\frac{\cos^2\psi_u}{4\pi} \leqslant \mathrm{sgn}(\psi_w - \psi_v)\nabla_{\psi_w}L_{\mathrm{cc}}(\boldsymbol{w}, \psi_w) \leqslant \frac{3}{\pi} \quad \text{and} \quad \psi_w' \in [\psi_l, \psi_u] .$$

**Proof.** For any $(\boldsymbol{w}, \psi_w) \in D_1$, the gradient of $L_{\mathrm{cc}}$ with respect to $\psi_w$ can be calculated by

$$\nabla_{\psi_w}L_{\mathrm{cc}}(\boldsymbol{w}, \psi_w) = \begin{cases} -\frac{1}{2\pi}[1 + \cos(2\psi_w)][1 - \|\boldsymbol{w} - \boldsymbol{v}\|^2] , & \psi_w < \psi_v , \\ \frac{1}{2\pi}[1 + \cos(2\psi_w)]\|\boldsymbol{w}\|^2 , & \psi_w > \psi_v , \end{cases}$$

where the gradient at $\psi_w = \psi_v$ can be any subgradient. For any $(\boldsymbol{w}, \psi_w) \in D_2$, we have $\psi_w \in [\psi_l, \psi_u]$, which indicates $2\cos^2\psi_u \leqslant 1 + \cos(2\psi_w) \leqslant 2$. Meanwhile, all points in $D_2$ satisfies $1 - 2R^2 \leqslant 1 - \|\boldsymbol{w} - \boldsymbol{v}\|^2 \leqslant 1$ and $(1 - R)^2 \leqslant \|\boldsymbol{w}\|^2 \leqslant (1 + R)^2 + R^2$. Thus, the gradient of $L_{\mathrm{cc}}$ with respect to $\psi_w$ can be bounded by

$$\frac{\cos^2\psi_u}{4\pi} \leqslant \mathrm{sgn}(\psi_w - \psi_v)\nabla_{\psi_w}L_{\mathrm{cc}}(\boldsymbol{w}, \psi_w) \leqslant \frac{3}{\pi} ,$$

where the first and second inequalities holds based on $R \leqslant 1/2$. Then $\psi_w'$ satisfies

$$\psi_w' = \psi_w - \eta\nabla_{\psi_w}L_{\mathrm{cc}}(\boldsymbol{w}, \psi_w) \leqslant \max\left\{\psi_w, \psi_v + \frac{3\eta}{\pi}\right\} \leqslant \psi_u ,$$

where the first inequality holds from discussing the relation between $\psi_w$ and $\psi_v$, and the second inequality holds based on $\psi_w \leqslant \psi_u$ and $\eta \leqslant \pi(\psi_u - \psi_v)/3$. Meanwhile, one has

$$\psi_w' = \psi_w - \eta\nabla_{\psi_w}L_{\mathrm{cc}}(\boldsymbol{w}, \psi_w) \geqslant \min\left\{\psi_w, \psi_v - \frac{3\eta}{\pi}\right\} \geqslant \psi_l ,$$

where the first inequality holds from discussing the relation between $\psi_w$ and $\psi_v$, and the second inequality holds based on $\psi_w \geqslant \psi_l$ and $\eta \leqslant \pi(\psi_v - \psi_l)/3$. Thus, we have completed the proof. $\square$

**Lemma 25.** *Let* $\psi_w' = \psi_w - \eta\nabla_{\psi_w}L_{\mathrm{cc}}(\boldsymbol{w}, \psi_w)$ *with* $(\boldsymbol{w}, \psi_w) \in D_2$. *If* $R \leqslant 1/2$ *and* $\arcsin R + 9\eta \leqslant \psi_u - \psi_v$, *then we have*

$$-9 \leqslant -2\left(\frac{\pi}{2} - \psi_w\right)^2 - 2\left(\frac{\pi}{2} - \psi_w\right)|w_2| \leqslant \nabla_{\psi_w}L_{\mathrm{cc}} \leqslant -\frac{1}{4}\left(\frac{\pi}{2} - \psi_w\right)^2 \quad \text{and} \quad \psi_w' \in [\psi_l, \psi_u] .$$

**Proof.** For any $(\boldsymbol{w}, \psi_w) \in D_1$, the gradient of $L_{\mathrm{cc}}$ with respect to $\psi_w$ can be calculated by

$$\nabla_{\psi_w}L_{\mathrm{cc}}(\boldsymbol{w}, \psi_w) = \frac{\|\boldsymbol{w}\|^2}{2\pi}[1 + \cos(2\psi_w)] - \frac{\|\boldsymbol{w}\|}{2\pi}[\cos\theta_{\boldsymbol{w},\boldsymbol{v}} + \cos(\theta_{\boldsymbol{w},\boldsymbol{v}} - 2\psi_w)] .$$

It is observed that the above expression is the same as the gradient of $L_{\mathrm{cr}}$ with respect to $\psi$ in Eq. (16). The only difference comes from the domain of $\boldsymbol{w}$, which is $\|\boldsymbol{w} - \boldsymbol{v}\| \leqslant R$ in Lemma 12 and $\|\boldsymbol{w} - \boldsymbol{v}\|_\infty \leqslant R$ here. Then according to $\|\boldsymbol{x}\| \leqslant \sqrt{2}\|\boldsymbol{x}\|_\infty$ in $\mathbb{R}^2$, one knows from Lemma 12 that

$$-9 \leqslant -2\left(\frac{\pi}{2} - \psi_w\right)^2 - 2\left(\frac{\pi}{2} - \psi_w\right)|w_2| \leqslant \nabla_{\psi_w}L_{\mathrm{cc}}(\boldsymbol{w}, \psi_w) \leqslant -\frac{1}{4}\left(\frac{\pi}{2} - \psi_w\right)^2 ,$$

where the first inequality holds according to $|\pi/2 - \psi_w| \leqslant \pi/2$ and $|w_2| \leqslant 1$, and the third inequality holds based on $R \leqslant 1/2$. Then $\psi_w'$ satisfies

$$\psi_w' \leqslant \psi_w + 9\eta \leqslant \psi_v + \theta_{\boldsymbol{w},\boldsymbol{v}} + 9\eta \leqslant \psi_u \,,$$

where the second inequality holds from the condition $\theta_{\boldsymbol{w},\boldsymbol{v}} \geqslant |\psi_w - \psi_v|$ in the definition of $D_2$, and the third inequality holds according to

$$\theta_{\boldsymbol{w},\boldsymbol{v}} \leqslant \arcsin R \leqslant \psi_u - \psi_v - 9\eta \,.$$

Meanwhile, it is observed that the gradient is always negative, which implies $\psi_w' \geqslant \psi_w \geqslant \psi_l$. Thus, we have completed the proof. □

## C.2 Convergence Rate Lemmas

This section presents some sufficient conditions for convergence with an inversely proportional rate.

**Lemma 26.** *Let $f : K \to \mathbb{R}$ represent a function with a global minimum $x^*$, where $K \subset \mathbb{R}$ indicates the convex domain satisfying $B(x^*, r_1) \subset K \subset B(x^*, r_2)$. Suppose that there exist constants $c_1, c_3, g_l, g_u$ such that $c_1 \leqslant r_1/g_u$ and for any $x \in K$, at least one of the following holds.*

1. *$|x' - x^*| \leqslant (1 - c_3\eta)|x - x^*|$ and $(x' - x^*)(x - x^*) \geqslant 0$ with $x' = x - \eta\nabla f(x)$ and $\eta \in (0, c_1]$.*

2. *$g_l \leqslant \operatorname{sgn}(x - x^*)\nabla f(x) \leqslant g_u$ for any $x \neq x^*$ and $|\nabla f(x^*)| \leqslant g_u$.*

*Then for any $c_2 \geqslant \max\{1/c_3, 2r_2/g_l, 2c_1g_u/g_l\}$, the sequence $\{x_t\}_{t=1}^\infty$ generated by gradient descent $x_{t+1} = x_t - \eta_t\nabla f(x_t)$ with $x_0 \in K$ and $\eta_t = \min\{c_1, c_2/t\}$ satisfies*

$$x_t \in K \quad and \quad |x_t - x^*| \leqslant \frac{a}{t} \quad with \quad a = \frac{c_2^2 g_l}{2c_1} \,.$$

**Proof.** Firstly, we prove $x_t \in K$. Suppose $x_t \in K$ for $t = k$. We prove $x_{k+1} \in K$ by discussion.

1. If the first condition holds, then $x_{k+1}$ is a convex combination of $x_k$ and $x^*$. Thus, $x_{k+1} \in K$.

2. If the second condition holds and $\operatorname{sgn}(x_{k+1} - x^*) = \operatorname{sgn}(x_k - x^*)$, then $x_{k+1}$ is a convex combination of $x_k$ and $x^*$. Thus, $x_{k+1} \in K$.

3. If the third condition holds and $\operatorname{sgn}(x_{k+1} - x^*) \neq \operatorname{sgn}(x_k - x^*)$, then one knows from $\eta_t \leqslant c_1$ and $|\nabla f(x)| \leqslant g_u$ that $|x_{k+1} - x^*| \leqslant c_1 g_u \leqslant r_1$, where the second inequality holds based on $c_1 \leqslant r_1/g_u$. Thus, $B(x^*, r_1) \subset K$ leads to $x_{k+1} \in K$.

Combining the cases above, $x_0 \in K$ and mathematical induction completes the proof of $x_t \in K$.

Secondly, we prove $|x_t - x^*| \leqslant a/t$. Let $t_0 = c_2/c_1$. According to $c_2 \geqslant 2c_1 g_u/g_l \geqslant 2c_1$, one knows $t_0 \geqslant 2$. For $t < t_0$, it is observed that

$$|x_t - x^*| \leqslant r_2 \leqslant \frac{a}{t_0} \leqslant \frac{a}{t} \,,$$

where the first inequality holds based on $K \subset B(x^*, r_2)$, the second inequality holds because of $a = c_2^2 g_l/(2c_1) \geqslant r_2 t_0$. Thus, the conclusion holds for any $t < t_0$. Suppose that $|x_k - x^*| \leqslant a/k$ holds for $k \geqslant t_0 - 1$. We then prove $|x_{k+1} - x^*| \leqslant a/(k+1)$ by discussion.

1. If the first condition holds, then we have

$$|x_{k+1} - x^*| \leqslant \left(1 - \frac{c_2 c_3}{k+1}\right)\frac{a}{k} \leqslant \frac{a}{k+1} \,,$$

where the first inequality holds based on the first condition and the induction hypothesis, and the second inequality holds from $c_2 \geqslant 1/c_3$. Thus, the conclusion holds for $t = k + 1$.

2. If the second condition holds and $\operatorname{sgn}(x_{k+1} - x^*) = \operatorname{sgn}(x_k - x^*)$, then one knows

$$|x_{k+1} - x^*| \leqslant \frac{a}{k} - \frac{c_2 g_l}{k+1} \leqslant \frac{a}{k+1} \,,$$

where the first inequality holds from the induction hypothesis and the second condition, and the second inequality holds because of

$$\frac{a}{k} - \frac{c_2 g_l}{k+1} - \frac{a}{k+1} = \frac{a - c_2 g_l k}{k(k+1)} = \frac{c_2 g_l(t_0/2 - k)}{k(k+1)} \leqslant 0 \,,$$

where the first equality holds based on $c_2 \geqslant 1/c_3$, the second equality holds from the choice of $a$ and $t_0$, and the first inequality holds from $t_0 \geqslant 2$ and $k \geqslant t_0 - 1 \geqslant t_0/2$. Thus, the conclusion holds for $t = k + 1$.

3. If the second condition holds and $\mathrm{sgn}(x_{k+1} - x^*) \neq \mathrm{sgn}(x_k - x^*)$, then it is observed that

$$|x_{k+1} - x^*| \leqslant \frac{c_2 g_u}{k+1} \leqslant \frac{a}{k+1},$$

where the first inequality holds from the second condition, and the second inequality holds based on $a = c_2^2 g_l/(2c_1) \geqslant c_2 g_u$. Thus, the conclusion holds for $t = k + 1$.

Combining the cases above, we have completed the proof. □

**Lemma 27.** *Let $f : K \to \mathbb{R}$ represent a function with a global minimum $x^*$, where $K \subset \mathbb{R}$ indicates the convex domain satisfying $B(x^*, r_1) \subset K \subset B(x^*, r_2)$. Let $\{\theta_t\}_{t=0}^{\infty}$ be a positive sequence bounded by $\theta_t \leqslant a/t$. Suppose that there exist constants $g_l, g_u$ such that for any $x \in K$, the following holds*

1. *If $|x_t - x^*| \geqslant \theta_t$, then $g_l \leqslant \mathrm{sgn}(x_t - x^*)\nabla f(x_t) \leqslant g_u$.*

2. *If $|x_t - x^*| \leqslant \theta_t$, then $|\nabla f(x_t)| \leqslant g_u$.*

*Let $c_1 > 0$, and $c_2 \geqslant \max\{2r_2/g_l, 2c_1\}$. Suppose that the sequence $\{x_t\}_{t=1}^{\infty}$ generated by gradient descent $x_{t+1} = x_t - \eta_t \nabla f(x_t)$ with $x_0 \in K$ and $\eta_t = \min\{c_1, c_2/t\}$ satisfies $x_t \in K$ for any $t \in \mathbb{N}^+$. Then the following holds for any $t \in \mathbb{N}^+$*

$$|x_t - x^*| \leqslant \frac{b}{t} \quad \text{with} \quad b = \max\left\{2a + c_2 g_u, \frac{c_2^2 g_l}{2c_1}\right\}.$$

**Proof.** Let $t_0 = 2b/(c_2 g_l) \geqslant c_2/c_1 \geqslant 2$. For any $0 < t < t_0$, it is observed that

$$|x_t - x^*| \leqslant r_2 \leqslant \frac{c_2 g_l}{2} = \frac{b}{t_0} \leqslant \frac{b}{t}.$$

Thus, the conclusion holds for $0 < t < t_0$. Suppose that $|x_k - x^*| \leqslant b/k$ holds for $k \geqslant t_0 - 1$. We then prove $|x_{k+1} - x^*| \leqslant b/(k+1)$ by discussion.

1. If the first condition holds and $\mathrm{sgn}(x_{k+1} - x^*) = \mathrm{sgn}(x_k - x^*)$, then we have

$$|x_{k+1} - x^*| \leqslant |x_k - x^*| - \eta_{k+1} g_l \leqslant \frac{b}{k} - \frac{c_2 g_l}{k+1} \leqslant \frac{b}{k+1},$$

where the second inequality holds from the induction hypothesis, and the third inequality holds based on $b = c_2 g_l t_0/2$ and $t_0/2 \leqslant t_0 - 1 \leqslant k$. Thus, the conclusion holds for $t = k + 1$.

2. If the first condition holds and $\mathrm{sgn}(x_{k+1} - x^*) \neq \mathrm{sgn}(x_k - x^*)$, then we have

$$|x_{k+1} - x^*| \leqslant \eta_{k+1} g_u \leqslant \frac{c_2 g_u}{k+1} \leqslant \frac{b}{k+1},$$

which implies that the conclusion holds for $t = k + 1$.

3. If the second condition holds, then one knows

$$|x_{k+1} - x^*| \leqslant |x_k - x^*| + \eta_{k+1} g_u \leqslant \frac{a}{k} + \frac{c_2 g_u}{k+1} \leqslant \frac{b}{k+1},$$

where the second inequality holds based on $|x_{k+1} - x^*| \leqslant \theta_{k+1} \leqslant a/(k+1)$, and the third inequality holds because of $b \geqslant 2a + c_2 g_u$. Thus, the conclusion holds for $t = k + 1$.

Combining the cases above, we have completed the proof. □

**Lemma 28.** *Let $f : K \to \mathbb{R}$ represent a function with a global minimum $x^*$, where $K \subset \mathbb{R}$ indicates the convex domain satisfying $K \subset B(x^*, R)$. Let $\{x_t\}_{t=1}^{\infty}$ denote the sequence generated by gradient descent $x_{t+1} = x_t - \eta_t \nabla f(x_t)$ with $x_0 \in K$ and $\eta_t = \min\{c_1, c_2/t\}$, satisfying $x_t \in K$ for $t \in \mathbb{N}^+$. Suppose that the gradient satisfies $\nabla f(x_t) = d(x_t - x^*) + r_t$, where $d_l \leqslant d \leqslant d_u$ and $|r_t| \leqslant e/t$. If $c_1 \leqslant 1/d_u$ and $c_2 \geqslant 2/d_l$, then we have*

$$|x_t - x^*| \leqslant \frac{c}{t} \quad \text{with} \quad c = \max\left\{\frac{c_2 R}{c_1}, c_2 e\right\}.$$

**Proof.** Let $t_0 = c_2/c_1$. We prove the conclusion by mathematical induction.

1. Base case. For $0 < t \leqslant t_0$, it is observed that

$$|x_t - x^*| \leqslant R \leqslant \frac{c}{t_0} \leqslant \frac{c}{t} \,.$$

   Thus, the conclusion holds for $0 < t \leqslant t_0$.

2. Induction. Suppose that $|x_k - x^*| \leqslant c/k$ holds for $k \geqslant t_0 - 1$. Then we have

$$|x_{k+1} - x^*| = |(1 - d\eta_k)(x_k - x^*) - \eta_k r_k| \leqslant (1 - d\eta_k)|x_k - x^*| + \eta_k |r_k| \,,$$

   where the first inequality holds based on $d\eta_k \leqslant c_1 d_u \leqslant 1$. Then the induction hypothesis leads to

$$|x_{k+1} - x^*| \leqslant \left(1 - \frac{2}{k}\right)\frac{c}{k} + \frac{c_2 e}{k^2} \leqslant \frac{c}{k+1} \,,$$

   where the first inequality holds according to $c_2 d_l \geqslant 2$, and the second inequality holds based on $c \geqslant c_2 e$. Thus, the conclusion holds for $t = k + 1$.

Therefore, mathematical induction completes the proof. $\qquad\square$

## D  Proof of Theorem 4

We begin the proof with two lemmas. For any non-zero vector $\boldsymbol{a}$ in $\mathbb{R}^2$ and $\theta \in [0, \pi]$, define $S(\boldsymbol{a}, \theta) = \{\boldsymbol{x} \in \mathbb{R}^2 \mid \theta_{\boldsymbol{x}} \in [\theta_{\boldsymbol{a}} - \theta, \theta_{\boldsymbol{a}} + \theta]\}$ as the sector region with central angle $2\theta$ that is symmetric with respect to $\boldsymbol{a}$. Let $\mathcal{N}_{\boldsymbol{a}, \theta}$ represent the truncated standard Gaussian distribution on $S(\boldsymbol{a}, \theta)$, of which the probability density function is

$$p(\boldsymbol{x}) = \begin{cases} \frac{1}{2\theta} e^{-\frac{1}{2}\|\boldsymbol{x}\|^2} \,, & \boldsymbol{x} \in S(\boldsymbol{a}, \theta) \,, \\ 0 \,, & \text{otherwise} \,. \end{cases}$$

The following lemma provides a lower bound for the expected squared inner product on $S(\boldsymbol{a}, \theta)$.

**Lemma 29.** *Let $d = 1$. For any $\boldsymbol{w} \in \mathbb{R}^{2d}$, non-zero $\boldsymbol{a} \in \mathbb{R}^{2d}$, and $\theta \in [0, \pi/2]$, we have*

$$\mathbb{E}_{\boldsymbol{x} \sim \mathcal{N}_{\boldsymbol{a}, \theta}} \left[\left(\boldsymbol{w}^\top \boldsymbol{x}\right)^2\right] \geqslant \frac{\theta^2}{3} \|\boldsymbol{w}\|^2 \,.$$

**Proof.** Let $\theta_{\boldsymbol{w}}$ indicate the phase of $\boldsymbol{w}$, i.e., $\boldsymbol{w} = \|\boldsymbol{w}\|(\sin\theta_{\boldsymbol{w}} + \cos\theta_{\boldsymbol{w}} \mathrm{i})$. Then calculating the expectation in the polar coordinate system leads to

$$\begin{aligned}
\mathbb{E}_{\boldsymbol{x} \sim \mathcal{N}_{\boldsymbol{a}, \theta}} \left[\left(\boldsymbol{w}^\top \boldsymbol{x}\right)^2\right] &= \frac{\|\boldsymbol{w}\|^2}{2\theta} \int_0^{+\infty} \int_{\theta_{\boldsymbol{a}} - \theta}^{\theta_{\boldsymbol{a}} + \theta} r^3 (\cos\theta_{\boldsymbol{w}} \cos\phi + \sin\theta_{\boldsymbol{w}} \sin\phi)^2 e^{-\frac{1}{2}r^2} \, \mathrm{d}\phi \, \mathrm{d}r \\
&= \frac{\|\boldsymbol{w}\|^2}{\theta} \left[\theta + \frac{1}{2}\sin(2\theta)\cos(2\theta_{\boldsymbol{a}, \boldsymbol{w}})\right] \,,
\end{aligned} \tag{38}$$

where the second equality holds based on integrating over $r$ and $\phi$ separately, and the identity $\cos(\theta_{\boldsymbol{a}} - \theta_{\boldsymbol{w}}) = \cos\theta_{\boldsymbol{a}, \boldsymbol{w}}$. The expectation in Eq. (38) can be further bounded by

$$\begin{aligned}
\mathbb{E}_{\boldsymbol{x} \sim \mathcal{N}_{\boldsymbol{a}, \theta}} \left[\left(\boldsymbol{w}^\top \boldsymbol{x}\right)^2\right] &= \|\boldsymbol{w}\|^2 \left[\left(1 - \frac{1}{2\theta}\sin(2\theta)\right) + \frac{1}{\theta}\sin(2\theta)\cos^2\theta_{\boldsymbol{a}, \boldsymbol{w}}\right] \\
&\geqslant \left(1 - \frac{1}{2\theta}\sin(2\theta)\right)\|\boldsymbol{w}\|^2 \\
&\geqslant \frac{\theta^2}{3}\|\boldsymbol{w}\|^2 \,,
\end{aligned}$$

where the first inequality holds according to $\theta \in [0, \pi/2]$, and the second inequality holds because of $\sin(x) \leqslant x - x^3/12$ for all $\theta \in [0, \pi/2]$. Thus, we have completed the proof. $\qquad\square$

The following lemma provides a lower bound for expressing a complex-valued vector with four real-valued vectors under a symmetric constant.

**Lemma 30.** *Let $\boldsymbol{v}_k \in \mathbb{R}^d$ with $k \in [4]$ and $\boldsymbol{v} \in \mathbb{R}^d$. If $\boldsymbol{v}_1 + \boldsymbol{v}_3 = \boldsymbol{v}_2 + \boldsymbol{v}_4$, then we have*

$$\sum_{k=1}^{4} \|\boldsymbol{v}_i - \boldsymbol{v} \cdot \mathbb{I}(k=1)\|^2 \geqslant \frac{1}{4}\|\boldsymbol{v}\|^2 \ .$$

**Proof.** According to the generalized mean inequality, one knows

$$\sum_{k=1}^{4} \|\boldsymbol{v}_i - \boldsymbol{v} \cdot \mathbb{I}(k=1)\|^2 \geqslant \frac{1}{4}\left(\sum_{k=1}^{4} \|\boldsymbol{v}_i - \boldsymbol{v} \cdot \mathbb{I}(k=1)\|\right)^2 \geqslant \frac{1}{4}\|(\boldsymbol{v}_1 - \boldsymbol{v}) - \boldsymbol{v}_2 + \boldsymbol{v}_3 - \boldsymbol{v}_4\|^2 = \frac{1}{4}\|\boldsymbol{v}\|^2 \ ,$$

where the second inequality holds because of the triangle inequality, and the first equality holds based on the condition $\boldsymbol{v}_1 + \boldsymbol{v}_3 = \boldsymbol{v}_2 + \boldsymbol{v}_4$. Thus, we have completed the proof. $\qquad\square$

We are now ready to prove Theorem 4.

**Proof of Theorem 4.** We define $\mathcal{N}_{\boldsymbol{\alpha},\mathbf{W}} = \sum_{i=1}^{n} \alpha_i \tau(\boldsymbol{w}_i^\top \boldsymbol{x})$ for simplicity. From $d = 1$, the weight vector $\boldsymbol{w}_i$ is a 2-dimensional real-valued vector. Let $\theta_{\boldsymbol{w}_i} = \arctan(w_{i,1}^{-1} w_{i,2}) \in (-\psi, 2\pi - \psi]$ denote the phase of $\boldsymbol{w}_i$. We assume $\theta_{\boldsymbol{v}} = 0$ without loss of generality. Denote by $\Theta_{\mathbf{W}}$ the $\pi/2$-symmetrical phase set induced from $\mathbf{W}$ and $\psi$, i.e.,

$$\Theta_{\mathbf{W}} = \left\{ \theta_{\boldsymbol{w}_i} + \frac{(j-1)\pi}{2} \,\middle|\, i \in [n], j \in [4] \right\} \cup \left\{ i\psi + \frac{(j-1)\pi}{2} \,\middle|\, i \in \{-1, +1\}, j \in [4] \right\} \ .$$

It is observed that there is an integer $m \leqslant n + 2$ such that $|\Theta_{\mathbf{W}}| = 4m$. We sort all phases in $\Theta_{\mathbf{W}}$ as

$$\Theta_{\mathbf{W}} = \{\theta_i\}_{i=1}^{4m} \quad \text{with} \quad -\psi < \theta_1 < \cdots < \theta_{4m} = 2\pi - \psi \ .$$

Let $\mathcal{N}_{\boldsymbol{\beta},\mathbf{U}}$ represent an arbitrary two-layer RVNN with weight phases from $\Theta_{\mathbf{W}}$, i.e.,

$$\mathcal{N}_{\boldsymbol{\beta},\mathbf{U}}(\boldsymbol{x}) = \sum_{i=1}^{4m} \beta_i \tau(\boldsymbol{u}_i^\top \boldsymbol{x}) \quad \text{with} \quad \theta_{\boldsymbol{u}_i} = \theta_i \ .$$

It is observed that $\mathcal{N}_{\boldsymbol{\beta},\mathbf{U}}$ degenerates to $\mathcal{N}_{\boldsymbol{\alpha},\mathbf{W}}$ with suitable parameters. Thus, the expected square loss $L_{\mathrm{rc}}$ can be bounded as

$$
\begin{aligned}
L_{\mathrm{rc}}(\boldsymbol{\alpha}, \mathbf{W}) &\geqslant \frac{1}{2} \inf_{\boldsymbol{\beta},\mathbf{U}} \mathbb{E}_{\boldsymbol{x} \sim \mathcal{N}(\mathbf{0},\mathbf{I})} \left[ \left( \mathcal{N}_{\boldsymbol{\beta},\mathbf{U}}(\boldsymbol{x}) - \sigma_\psi(\boldsymbol{v}_\mathbb{C}^\top \overline{\boldsymbol{x}}_\mathbb{C}) \right)^2 \right] \\
&= \frac{1}{2} \inf_{\boldsymbol{\beta},\mathbf{U}} \sum_{i=1}^{4m} \frac{\Delta\theta_i}{\pi} \mathbb{E}_{\boldsymbol{x} \sim \mathcal{N}(\boldsymbol{a}_i,\Delta\theta_i)} \left[ \left( \mathcal{N}_{\boldsymbol{\beta},\mathbf{U}}(\boldsymbol{x}) - \sigma_\psi(\boldsymbol{v}_\mathbb{C}^\top \overline{\boldsymbol{x}}_\mathbb{C}) \right)^2 \right] \ ,
\end{aligned}
\tag{39}
$$

where $\Delta\theta_i = (\theta_i - \theta_{i-1})/2$ and $\boldsymbol{a}_i = \mathrm{e}^{(\theta_i - \Delta\theta_i)\mathrm{i}}$ with $\theta_0 = \theta_{4(n+1)}$. The indices can be divided into $m$ groups as $\mathcal{I}_i = \{i + (k-1)m \mid k \in [4]\}$ with $i \in [m]$. Denote by $i_\psi$ the index of $\psi$, i.e., $\theta_{i_\psi} = \psi$. Then Eq. (39) becomes

$$
\begin{aligned}
L_{\mathrm{rc}}(\boldsymbol{\alpha}, \mathbf{W}) &\geqslant \frac{1}{2} \inf_{\boldsymbol{\beta},\mathbf{U}} \sum_{i=1}^{m} \frac{\Delta\theta_i}{\pi} \sum_{j \in \mathcal{I}_i} \mathbb{E}_{\boldsymbol{x} \sim \mathcal{N}(\boldsymbol{a}_j,\Delta\theta_j)} \left[ \left( \mathcal{N}_{\boldsymbol{\beta},\mathbf{U}}(\boldsymbol{x}) - \sigma_\psi(\boldsymbol{v}_\mathbb{C}^\top \overline{\boldsymbol{x}}_\mathbb{C}) \right)^2 \right] \\
&= \frac{1}{2} \inf_{\boldsymbol{\beta},\mathbf{U}} \sum_{i=1}^{m} \frac{\Delta\theta_i}{\pi} \sum_{j \in \mathcal{I}_i} \mathbb{E}_{\boldsymbol{x} \sim \mathcal{N}(\boldsymbol{a}_j,\Delta\theta_j)} \left[ \left( (\boldsymbol{v}_j - \boldsymbol{v} \cdot \mathbb{I}(j \leqslant i_\psi))^\top \boldsymbol{x} \right)^2 \right] \ ,
\end{aligned}
\tag{40}
$$

where the first inequality holds since $\Delta\theta_j$ remains the same in $\mathcal{I}_i$, the second inequality holds based on the activation regions of ReLU and zReLU, and the definition of $\boldsymbol{v}_j$ as follows

$$\boldsymbol{v}_j = \sum_{l=j-m}^{j+m-1} \beta_{\phi(l)} \boldsymbol{u}_{\phi(l)} \quad \text{with} \quad \phi(l) = \begin{cases} l + 4m, & l \leqslant 0, \\ l, & 0 < l \leqslant 4m, \\ l - 4m, & l > 4m . \end{cases} \tag{41}$$

Applying Lemma 29 to Eq. (40), we obtain

$$
\begin{aligned}
L_{\mathrm{rc}}(\boldsymbol{\alpha}, \mathbf{W}) &\geqslant \frac{1}{2} \inf_{\boldsymbol{\beta},\mathbf{U}} \sum_{i=1}^{m} \frac{\Delta\theta_i}{\pi} \sum_{j \in \mathcal{I}_i} \frac{(\Delta\theta_j)^2}{3} \|\boldsymbol{v}_j - \boldsymbol{v} \cdot \mathbb{I}(j \leqslant i_\psi)\|^2 \\
&\geqslant \frac{1}{2} \inf_{\boldsymbol{\beta},\mathbf{U}} \sum_{i=\max\{1, i_\psi - m + 1\}}^{\min\{i_\psi, m\}} \frac{(\Delta\theta_i)^3}{3\pi} \sum_{k=1}^{4} \|\boldsymbol{v}_{i,k} - \boldsymbol{v} \cdot \mathbb{I}(k=1)\|^2 \ ,
\end{aligned}
$$

where the second inequality holds based on the definition of $\boldsymbol{v}_{i,k} = \boldsymbol{v}_{i+(k-1)(n+1)}$ and $\Delta\theta_j = \Delta\theta_i$ for any $j \in \mathcal{I}_i$. Based on Eq. (41), one has $\boldsymbol{v}_{i,1} + \boldsymbol{v}_{i,3} = \boldsymbol{v}_{i,2} + \boldsymbol{v}_{i,4}$. Then Lemma 30 implies

$$L_{\mathrm{rc}}(\boldsymbol{\alpha}, \mathbf{W}) \geqslant \frac{1}{2} \inf_{\boldsymbol{\beta}, \mathbf{U}} \sum_{i=\max\{1, i_\psi - m+1\}}^{\min\{i_\psi, m\}} \frac{(\Delta\theta_i)^3}{3\pi} \cdot \frac{1}{4} \|\boldsymbol{v}\|^2$$

$$\geqslant \frac{\|\boldsymbol{v}\|^2}{24\pi(\max\{1, i_\psi - m+1\} - \min\{i_\psi, m\})^2} \left( \sum_{i=\max\{1, i_\psi - m+1\}}^{\min\{i_\psi, m\}} \Delta\theta_i \right)^3$$

$$\geqslant \frac{\|\boldsymbol{v}\|^2 \min\{2\psi, \pi - 2\psi\}^3}{24\pi(n+2)^2} \, ,$$

where the second inequality holds based on the generalized mean inequality, and the third one holds from $\max\{1, i_\psi - m+1\} - \min\{i_\psi, m\} \leqslant m \leqslant n + 2$. Thus, we have completed the proof. $\qquad\square$

## E  Proof of Theorem 6

We begin with a lemma providing a lower bound for convergence.

**Lemma 31.** *If there exists a constant c such that*

$$\langle \nabla f(\boldsymbol{w}), \boldsymbol{w} - \boldsymbol{v} \rangle \leqslant c \|\boldsymbol{w} - \boldsymbol{v}\|^2 \, ,$$

*then $\boldsymbol{w}' = \boldsymbol{w} - \eta \nabla f(\boldsymbol{w})$ with $\eta \in (0, 1/(2c))$ satisfies*

$$\|\boldsymbol{w}' - \boldsymbol{v}\| \geqslant \sqrt{1 - 2c\eta} \|\boldsymbol{w} - \boldsymbol{v}\| \, .$$

**Proof.** From the updating rule, it is observed that

$$\|\boldsymbol{w}' - \boldsymbol{v}\|^2 \geqslant \|\boldsymbol{w} - \boldsymbol{v}\|^2 - 2\eta \langle \boldsymbol{w} - \boldsymbol{v}, \nabla f(\boldsymbol{w}) \rangle \geqslant (1 - 2c\eta)\|\boldsymbol{w} - \boldsymbol{v}\|^2 \, ,$$

which completes the proof. $\qquad\square$

We then prove Theorem 6.

**Proof of Theorem 6.** Denote by $R = \|\boldsymbol{w}_0 - \boldsymbol{v}\|$. The convergence analysis consists of several stages.

**Stage 1: the error of $\psi$ decreases below a threshold fast.** By the same arguments as those in the proof of Theorem 1, $\eta \in (0, 1/(12\pi))$ indicates $(\boldsymbol{w}_t, \psi_t) \in D$ for any $t \in \mathbb{N}$. Recalling the convergence of $\psi$ in Eq. (7), we have $\psi_t \geqslant \pi/4$ when $t \geqslant \lceil 16\eta^{-1}(1 - R^2)^{-1} \rceil$. From Eq. (4), one knows $\nabla_\psi L_{\mathrm{cr}}(\boldsymbol{w}_t, \psi_t) \geqslant -6(\psi^* - \psi_t)$. Then we have

$$\langle \nabla_\psi L_{\mathrm{cr}}(\boldsymbol{w}_t, \psi_t), \psi^* - \psi_t \rangle \geqslant -6(\psi^* - \psi_t)^2 \, .$$

Then we obtain from $\eta \in (0, 1/12)$ and Lemma 31 that

$$\psi^* - \psi_t \geqslant (1 - 12\eta)^{t/2}(\psi^* - \psi_0) \, . \tag{42}$$

Thus, one has

$$(1 - 12\eta)^{t/2}(\psi^* - \psi_0) \leqslant \psi^* - \psi_t \leqslant \frac{\pi}{4} \quad \text{with} \quad t \geqslant T_1 = 16\eta^{-1}(1 - R^2)^{-1} \, .$$

**Step 2: both errors of $w$ and $\psi$ decrease below small constants fast.** Based on Eq. (8), we have

$$\|\boldsymbol{w}_t - \boldsymbol{v}\| \leqslant \left(1 - \frac{\eta}{48}\right)^{t - T_1} \quad \text{for} \quad t \geqslant T_1 \, , \tag{43}$$

which, together with Eqs. (7) and (42), implies that

$$(1 - 12\eta)^{t/2}(\psi^* - \psi_0) \leqslant \psi^* - \psi_t \leqslant \frac{1}{384} \quad \text{and} \quad |w_2| \leqslant \|\boldsymbol{w}_t - \boldsymbol{v}\| \leqslant \frac{1}{384} \, ,$$

$$\text{with} \quad t \geqslant T_2 = \max\left\{ T_1 + \frac{\ln 384}{\ln(1 + \eta/48)}, \frac{3200\pi}{\eta(1 - R^2)} \right\} \, . \tag{44}$$

**Step 3: $w$ converges faster than $\psi$.** For any $t \geqslant T_2$, Lemmas 11 and 12 imply

$$\langle \nabla_\psi L_{\mathrm{cr}}(\boldsymbol{w}_t, \psi_t), \psi_t - \psi^* \rangle \leqslant 2(\psi^* - \psi_t)^3 + 2(\psi^* - \psi_t)^2 |w_{2,t}| \leqslant \frac{1}{96}(\psi^* - \psi_t)^2 \ ,$$

where the second inequality holds based on Eq. (44). Then Lemma 31 indicates

$$\psi^* - \psi_{t+1} \geqslant \sqrt{1 - \eta/48}(\psi^* - \psi_t) \quad \text{for} \quad t \geqslant T_2 \ ,$$

which, together with Eq. (43), indicates

$$|w_{w,t}| \leqslant \|\boldsymbol{w}_t - \boldsymbol{v}\| \leqslant \psi^* - \psi_t \quad \text{with} \quad t \geqslant T_3 = 2T_1 + \frac{T_2 \ln(1 - 12\eta) + 2\ln(\psi^* - \psi_0)}{\ln(1 - \eta/48)} \ . \quad (45)$$

**Step 4: $\psi$ converges with an inversely proportional rate.** For any $t \geqslant T_3$, it is observed from Lemmas 11, 12, and Eq. (45) that

$$\nabla_\psi L_{\mathrm{cr}}(\boldsymbol{w}_t, \psi_t) \geqslant -4(\psi^* - \psi)^2 \ .$$

Let $a_t = 4\eta(\psi^* - \psi_t)$. Then the updating rule implies $a_{t+1} \geqslant a_t(1 - a_t)$. Choosing $\eta \in (0, 1/(4\pi))$ guarantees $a_t \in [0, 1/2]$. Then Lemma 14 indicates

$$\psi^* - \psi_t \geqslant \frac{(1 - 12\eta)^{T_3/2}(\psi^* - \psi_0)}{t - T_3 + 1} \quad \text{for} \quad t \geqslant T_3 \ . \quad (46)$$

**Step 5: the loss converges to $0$ with an inversely proportional rate.** Define non-negative quantities $\Delta_{\boldsymbol{w}} = \|\boldsymbol{w} - \boldsymbol{v}\|$ and $\Delta_\psi = \psi^* - \psi$. We provide a lower bound for $L_{\mathrm{cr}}$ by discussion.

1. Suppose $(\boldsymbol{w}, \psi) \in D_1$. Then we have

$$L_{\mathrm{cr}}(\boldsymbol{w}, \psi) \geqslant \frac{1}{4} - \frac{1}{8\pi}(4\psi^* - \Delta_\psi^3)(1 - \Delta_{\boldsymbol{w}}^2) = \frac{1}{8\pi}\Delta_\psi^3 + \frac{1}{8\pi}\Delta_{\boldsymbol{w}}^2(2\pi - \Delta_\psi^3) \geqslant \frac{1}{8\pi}\Delta_\psi^3 \ , \quad (47)$$

   where the first inequality holds based on $\sin(2\psi) + 2\psi = \sin(2\Delta_\psi) + 2\psi^* - 2\Delta_\psi \leqslant 2\psi^* - \Delta_\psi^3/2$ for any $\psi \in [0, \pi/2]$, and the second inequality holds from $\Delta_\psi \leqslant \pi/2$.

2. Suppose $(\boldsymbol{w}, \psi) \in D_2$. The expected loss can be rewritten as

$$\begin{aligned}
L_{\mathrm{cr}}(\boldsymbol{w}, \psi) &= \frac{1}{4} - \frac{1}{4\pi}[\sin(2\psi) + 2\psi](1 - \Delta_{\boldsymbol{w}}^2) \\
&\quad + \frac{1}{4\pi}[(\cos(2\psi) - 1)|w_2| + (\sin(2\psi) + 2\psi + 2\theta - 2\psi^*)w_1] \\
&\geqslant \frac{1}{4} - \frac{1}{8\pi}(4\psi^* - \Delta_\psi^3)(1 - \Delta_{\boldsymbol{w}}^2) + \frac{1}{4\pi}[(\cos(2\psi) - 1)|w_2|] \quad (48) \\
&\geqslant \frac{1}{4} - \frac{1}{8\pi}(4\psi^* - \Delta_\psi^3)(1 - \Delta_{\boldsymbol{w}}^2) - \frac{1}{2\pi}\Delta_{\boldsymbol{w}} \\
&\geqslant \frac{1}{8\pi}\Delta_\psi^3 - \frac{1}{2\pi}\Delta_{\boldsymbol{w}} \ ,
\end{aligned}$$

   where the first inequality holds from $\sin(2\psi) + 2\psi \leqslant 2\psi^* - \Delta_\psi^3/2$ and $\sin(2\psi) + 2\psi + 2\theta - 2\psi^* \geqslant 0$, the second inequality holds based on $\cos(2\psi) - 1 \geqslant -2$ and $|w_2| \leqslant \Delta_{\boldsymbol{w}}$.

Combining Eqs. (47) and (48), one knows that the following holds for any $(\boldsymbol{w}_0, \psi_0) \in D$ and $t \geqslant T_3$

$$L_{\mathrm{cr}}(\boldsymbol{w}_t, \psi_t) \geqslant \frac{1}{8\pi}\Delta_{\psi,t}^3 - \frac{1}{2\pi}\Delta_{\boldsymbol{w},t} \geqslant \frac{(1 - 12\eta)^{3T_3/2}(\psi^* - \psi_0)^3}{8\pi(t - T_3 + 1)^3} - \frac{1}{2\pi}\left(1 - \frac{\eta}{48}\right)^{t - T_3} \ ,$$

where the second inequality holds from Eqs. (43) and (46). Thus, we have completed the proof. $\quad \square$

# F  Simulation Experiments

**Experimental settings.** A training set of size 7,000 and a test set of size 3,000 are generated by a randomly initialized target neuron (can be a real-valued or a complex-valued neuron). After random initialization, a complex-valued neuron and a real-valued neuron are trained by gradient descent with

the empirical mean square loss and a learning rate of 0.1 for 100 epochs (or 300 epochs when the loss does not converge).

**Experimental results.** It should be noticed that a complex-valued neuron cannot always learn a target neuron. From the theoretical formulation, our convergence rate holds with a small constant probability. From the loss landscape, there exist constant pieces in the parameter space, i.e., the complex-valued neuron does not learn anything after initialization. Thus, we cannot expect a complex-valued neuron to learn a target neuron all the time. In the experiments, we train the complex-valued neuron with several random initializations and find that our theoretical conclusions occur in experiments. This phenomenon verifies our theories and also motivates a novel learning algorithm for CVNNs, as discussed in the conclusion part.

