# OpenReview forum: "Complex-valued Neurons Can Learn More but Slower than Real-valued Neurons via Gradient Descent"
_NeurIPS.cc/2023/Conference — NeurIPS 2023 poster_

### Official Review · Reviewer_D3zC · 2023-07-02

**Soundness:** 3 good
**Presentation:** 3 good
**Contribution:** 3 good
**Rating:** 6
**Confidence:** 2

**Summary:**

The authors presents theoretical results on learning real-valued and complex-valued neurons with gradient descent. The key takeaways are that:
* complex-valued neuron learns real-values neurons and complex-valued neurons with convergence rate $O(t^{-3})$ and $O(t^{-1})$ respectively
* two-layer real-valued neural network cannot learn a single complex-valued neuron
* complex-valued neuron learns slower than real-valued neurons when learning real-value neurons

**Strengths:**

* The paper is well-written and easy to understand even for non-experts (myself)
* This appears to be a significant advance in the theory of complex-valued neuron learning, with previous work largely focusing on real-valued neuron learning. However, as someone outside this field, I defer to other reviewers' for their assessment of significance.


**Weaknesses:**

* While the main results in this paper are theoretical proofs, the paper could be furthered strengthed with empirical experiments that validate the proofs.
* table 2 appears to containa typo? the result for theorem 2 should be $O(t^{-1})$?

**Questions:**

* Do the authors expect the convergence rates to be further improved with other learning rate schedules/learning algorithms (other than projected gradient descent)?

**Limitations:**

yes

---

> ### Author Rebuttal · Authors · 2023-08-06
>
> Thanks for the comments and feedback.
>
> Q1: About empirical experiments to further strengthen the paper.
>
> A1: Thanks for your valuable suggestions. We provide toy experiments in the **global response**, where we verify our findings in more general settings.
>
> Q2: About the typo in Table 2.
>
> A2: Thanks for pointing out this problem. We will correct it in later versions.
>
> Q3: About further improving the convergence rate with other learning rate schedules and learning algorithms.
>
> A3: Absolutely yes. This paper focuses on traditional learning rate schedules and gradient descent methods since they are widely used in the analysis and early implementations of RVNNs. However, it remains an open problem whether they are the best choice for CVNNs. I prefer to believe they are not since learning CVNNs includes the learning of phase, which does not exist in learning RVNNs. It would be promising to find more suitable algorithms for CVNNs.

---

### Official Review · Reviewer_4m8w · 2023-07-07

**Soundness:** 2 fair
**Presentation:** 2 fair
**Contribution:** 3 good
**Rating:** 6
**Confidence:** 2

**Summary:**

The goal is to explore the novel approach of complex valued neural networks using gradient descent. The paper investigates when and to what extent CVNNs outperform RVNNs using gradient descent for learning tasks. The researchers prove that a single complex-valued neuron can efficiently learn functions expressed by one real-valued neuron and those expressed by one complex-valued neuron, while a two-layer RVNN with finite width cannot learn a single complex-valued neuron. However, they also show that complex-valued neurons learn more slowly than real-valued neurons, with a lower bound of Ω(t−3) for learning functions expressed by one real-valued neuron using a complex-valued neuron. Theoretical comparisons between RVNNs and CVNNs provide insights into the success of CVNNs and the potential slow convergence of CVNNs learning.

**Strengths:**

The authors appear to have thoroughly analysed a complex valued neural network and developed what appears to be rigorous theoretical results calculating the convergence rates of these complex valued neural networks.

The ideas appear to be sound

**Weaknesses:**

- While the notation appears to be self-consistent (in the main body - I did not read the appendix, which is very long and technical), the notation is very hard to parse through. If there were some tables to layout the important notation it could help lay the groundwork earlier on.

- I am not familiar with these types of reformulations, but it is not clear to me why the authors want to try to use a two layer network to model a single neuron. The authors write "Although learning one neuron is the simplest case of learning neural networks, it is sufficient to embody the learning capability and efficiency of neural networks.", which I ask the authors to motivate more. Is it not obvious a two layer network with more parameters would be able to model a single neuron of just the weighted inputs?

- I am not sure I fully understand this paper but it seems the authors only develop the proofs and theorem 1 and theorem 2 for 1d input such that they calculate the convergence rates for a two layer network to model a linear function wx where w is a vector length n and x is just a scaler number. If I understand this correctly, I am not sure how much we learn from this? Does this theorem apply for as real neuron where d is not one?

-The ideas appear to be sound but they are not very palatable to a reader.

**Questions:**

- Motivation of application - I’m surprised the authors do not mention more applications to systems in the world of physics. Fluid dynamics, beam physics, particle physics, where phase information is ubiquitously relevant to quantum materials, dynamics, and

-States that the CVNNs have shown to have some desirable properties - universal approximation - boundedness, stability → why is this desirable? Could the authors elaborate on this?

-  It is not clear to me, in Theorem 1 and theorem 2 when d=1, does this mean that x is now no longer a linear summation of inputs but truly just an input and the weight parameter? I did not work through the proof, but does the theorem generalize to inputs with more than one dimension?

**Limitations:**

It often feels like one cannot truly understand what they are discussing in the paper without referencing the very dense and long appendix.

---

> ### Author Rebuttal · Authors · 2023-08-06
>
> We thank the reviewer for the thorough feedback.
>
> Q1: About tables to layout the important notation.
>
> A1: Thanks for your constructive suggestions. The notations in this paper are self-consistent and are equipped with informal explanations when they occur in the context for the first time. Readers may forget the explanations when reading through the paper. Thus, we will consider adding some notation tables to remind readers of the definitions.
>
> Q2: About using a two-layer network to model a single neuron.
>
> A2: Thanks for your valuable feedback. Throughout this paper, we focus on learning a neuron with a neuron (not a two-layer network), which is a standard and widely studied setting [1,2,3]. But when we consider learning a complex-valued neuron with a real-valued neuron, this learning process is unfair since a complex-valued neuron has more parameters than a real-valued neuron. This motivates us to investigate learning a complex-valued neuron with a two-layer RVNN to pursue fairness. Thus, the formulation in Eq. (1) considers learning with a two-layer network to cover all settings in the paper. It seems that the formulation and explanations are too complicated for readers, and we will consider reformulating our problem in later versions.
>
> [1] Mahdi Soltanolkotabi. Learning ReLUs via Gradient Descent (NeurIPS 2017).
>
> [2] Seyed Mohammadreza Mousavi Kalan, Mahdi Soltanolkotabi, and A. Salman Avestimehr. Fitting ReLUs via SGD and Quantized SGD (ISIT 2019).
>
> [3] Gilad Yehudai and Ohad Shamir. Learning a Single Neuron with Gradient Methods (COLT 2020).
>
> Q3: About $d > 1$.
>
> A3: This is an insightful question. In our paper, $d$ represents the dimension of the complex space. From the theoretical aspect, we introduce this condition for technical reasons. In traditional neuron learning, spherical symmetry is used to reduce the analysis to 2-dimensional real space. The condition $d=1$, i.e., 1-dimensional complex space, is used to inherit the analysis framework. From the experimental aspect, we find that our findings also hold when $d$ is larger than one (Fig. 2 in the **global response**). As far as we know, this paper is the first one to analyze neuron learning in the complex domain, where symmetry becomes more complicated, and we rely on this condition to simplify the analysis. As supported by the experimental results, we believe that our findings can be extended to more general settings and we leave them for future explorations.
>
> Q4: About applications to physics.
>
> A4: Thanks for the valuable suggestions. We will consider introducing some applications related to physics in later versions.
>
> Q5: About desirable properties.
>
> A5: Thanks for raising these questions. If I have not misunderstood the questions, I need to elaborate on the motivation of these desirable properties, including universal approximation, boundedness and stability. These properties, which hold for both RVNNs and CVNNs, have been investigated for several decades in the community of neural network theory. Universal approximation roughly states that neural networks can approximate any measurable functions [4,5]. This property affirms the expressive power of neural networks and explains the phenomenon that neural networks can fit random data to perfect accuracy [6]. (Complete) stability roughly indicates that neural networks with any initialization will converge when it is optimized by gradient descent (or gradient flow) [7,8]. This property guarantees convergence and implies that the optimization is meaningful. In our paper, boundedness occurs together with stability since boundedness can be used to derive stability under some settings [8]. Although these properties may not directly reflect the performance of neural networks (i.e., generalization), these properties are still desirable since they help us theoretically understand neural networks.
>
> [4] Kurt Hornik, Maxwell Stinchcombe, and Halbert White. Multilayer feedforward networks are universal approximators (Neural Networks, 1989).
>
> [5] Felix Voigtlaender. The universal approximation theorem for complex-valued neural networks (Applied and Computational Harmonic Analysis, 2023).
>
> [6] Devansh Arpit et al. A closer look at memorization in deep networks (ICML 2017).
>
> [7] Michael A. Cohen and Stephen Grossberg. Absolute stability of global pattern formation and parallel memory storage by competitive neural networks (IEEE Transactions on Systems, Man, and Cybernetics, 1983).
>
> [8] Bo Zhou and Qiankun Song. Boundedness and complete stability of complex-valued neural networks with time delay (TNNLS 2013).
>
> Q6: About fair presentation, very long and technical appendix, hard-to-parse-through notations, and "it often feels like one cannot truly understand what they are discussing in the paper without referencing the very dense and long appendix''.
>
> A6: It seems that the reviewer pays too much attention to technical details and is trapped by mathematical formulas. It should be emphasized that we elaborate on our settings and conclusions in the introduction part, which is sufficient to understand and estimate our contributions. Meanwhile, we provide detailed and intuitive explanations after key definitions and theorems to help readers understand complicated mathematical formulas. If one hopes to delve into details of our theoretical results, it is necessary to possess some basic knowledge related to neuron learning and may require much more effort.

---

### Official Review · Reviewer_cJ2H · 2023-07-07

**Soundness:** 4 excellent
**Presentation:** 3 good
**Contribution:** 3 good
**Rating:** 7
**Confidence:** 3

**Summary:**

The authors contrast the learning and convergence properties of real-valued neural networks and complex-valued neural networks. Specifically, they study the problem of learning the function implemented by a single neuron using a 2-layer finite width network. Notably, they show that a complex valued neural network can learn functions expressed by a real-valued neuron as well as a complex-valued neuron. This result indicates the superior capacity of complex-valued neural networks as compared to their real-valued counterparts. Furthermore, they also study the convergence properties of learning a function expressed by a real-valued neuron with a complex-valued neuron and prove that its slower compared to learning with a real-valued neural network. Taken together, complex-valued neural networks have a better learning capacity at the cost of slower learning properties compared to real-valued neural networks. Overall, I think this paper makes a strong theoretical contribution to understanding the efficiency and capacity of complex-values neural networks in practice.

**Strengths:**

1. The paper has strong theoretical foundations and presents arguments through rigorous proofs.
2. Despite the involved mathematical machinery, the authors present a simplified view of the underlying learning dynamics and present the existence of phases in the learning process.
3. Despite the assumption of the MSE loss, I feel the authors present a strong theoretical framework to analyze learning in complex-valued neural networks and contrast them with real-valued neural networks.

**Weaknesses:**

1. The authors present strong theoretical results, and use specific assumptions to prove their results. However, it is unclear the extent to which these assumptions would hold in practice.
2. In addition to their (very impressive) theoretical results, I feel the current version of the paper would benefit from some empirical validation. Having some toy experiments would also be sufficient to significantly improve the readability of the paper, while also increasing the reader pool.
3. The manuscript is generally well-written, but there are certain typos or use of abbreviations/notations which make it confusing for the reader. E.g. it seems that there is a typo in Table 2, wherein the second row third column (complex-valued Neuron --> complex-valued neuron) should be $\mathcal{O}(t^{-1})$. Also, abbreviations CVNN and RVNN or the notation $t$ is not introduced in the abstract.

**Questions:**

1. The caption of Fig. 2 is currently not very descriptive. Could you please update it such that the takeaway is clear to the reader without reading in detail the text?
2. Is it possible to add a toy regression example where you can empirically demonstrate the different phases during the learning process?

---

> ### Author Rebuttal · Authors · 2023-08-06
>
> We thank the reviewer for these constructive suggestions.
>
> A1: About the extent to which these assumptions would hold in practice.
>
> Q1: The settings and assumptions used in this paper are common ones [1,2,3,4] to make the analysis tractable. Neuron learning aims to provide meaningful insights for theoretical guarantees and practical usage of neural networks, based on simple settings and assumptions. We believe that such a work is significant and necessary for the development of neural networks' theory.
>
> [1] Yuandong Tian. An analytical formula of population gradient for two-layered ReLU network and its applications in convergence and critical point analysis (ICML 2017).
>
> [2] Mahdi Soltanolkotabi. Learning ReLUs via Gradient Descent (NeurIPS 2017).
>
> [3] Seyed Mohammadreza Mousavi Kalan, Mahdi Soltanolkotabi, and A. Salman Avestimehr. Fitting ReLUs via SGD and Quantized SGD (ISIT 2019).
>
> [4] Weihang Xu and Simon S. Du. Over-parameterization exponentially slows down gradient descent for learning a single neuron (COLT 2023).
>
> Q2: The paper would benefit from some empirical validation. Is it possible to empirically demonstrate the different phases during the learning process?
>
> A2: Thanks for this valuable feedback. We provide toy experiments in the **global response**, where we verify our findings in more general settings. For empirically demonstrating learning phases, it should be emphasized that these phases only work for proof but may not exist in practice. For example, the red line remains constant during stage II in Fig. 2(b), but this line is an upper bound and the practical line may be a smooth one through all stages. In part of our experiments, the loss curve has a constant segment, which implies that such phases may exist sometimes.
>
> Q3: About typos, use of abbreviations, the caption of Fig. 2, and $t$ in the abstract.
>
> A3: Thanks for pointing out these problems. We will update them in later versions.

---

> > ### Comment · Reviewer_cJ2H · 2023-08-21
> > **Reply to the authors' rebuttal**
> >
> > I would like to thank the authors for their response and effort during the rebuttal process.
> >
> > > The settings and assumptions used in this paper are common ones [1,2,3,4] to make the analysis tractable.
> >
> > Noted. I must admit that I am not very familiar with this line of research and therefore, wasn't sure how reliable these assumptions are. But I understand that the assumptions are fairly standard in the area. Thank you for pointing it out.
> >
> > > We provide toy experiments in the global response, where we verify our findings in more general settings.
> >
> > This is great! It would be great if you could add it to the paper, along with the details of the experiment and hyperparameters (in Appendix, if space constrained).
> >
> > > For empirically demonstrating learning phases, it should be emphasized that these phases only work for proof but may not exist in practice.
> >
> > It is however, striking to see the similarity between Fig. 2 and the empirical plot Fig 1 (that you added in the general response). I guess the correspondence is weaker when the dimension conditions are softened and bias is added (Fig. 2 of your general response).
> >
> > Overall, I am convinced that this is an interesting theoretical contribution and would be interesting to the NeurIPS community. I have therefore, increased my score. I wish the authors best with their endeavours and would like to congratulate them again on their excellent work.

---

> > > ### Author Response · Authors · 2023-08-21
> > > **Rely**
> > >
> > > Thank Reviewer cJ2H for the detailed feedback and for raising the score. Have a nice day!

---

### Official Review · Reviewer_Jsv8 · 2023-07-07

**Soundness:** 2 fair
**Presentation:** 4 excellent
**Contribution:** 3 good
**Rating:** 5
**Confidence:** 3

**Summary:**

The paper develops theoretical results for the convergence rates of complex and real valued neural networks on complex-valued problems. It develops analytical models for the training regimes of complex valued NNs.

**Strengths:**

The results would be useful for those attempting to apply complex-valued networks to problem settings where they are not normally used. An intriguing result is that complex valued neurons have slower convergence rates for the same problems as real valued neurons. The theorems are thorough, and the diagrams provide excellent illustration to the proofs. They also provide a proof that real valued networks cannot learn complex valued functions, which is not surprising, but nice to have proved.


**Weaknesses:**

A fundamental flaw of the paper is that it does not have empirical verification of the theoretical results. The analysis makes many assumptions about different training regimes, but it is not obvious if complex valued neural networks actually behave like this. The complex-valued models in the theorems should be implemented and the convergence losses should be plotted alongside the theoretical predictions. Without supporting empirical evidence, the assumptions of the proof seem too strong. Fortunately, the problem settings should be simple to code up in any package, and the results would be easy to include in the author's presentation, given Figures 1 and 3.

Page 4, line 156: “whereas a complex-valued neuron may only activate a small part as controlled by the parameter psi” This line makes it sound like the authors conclude that the primary benefit is the extra parameter in zReLU, not the complex arithmetic. The use of activation functions with trainable parameters is also a major distinction between the CVNN and the RVNN. Maybe all of the identified differences can come from a network with real weights, but an activation function with a learnable parameter? zReLU can be encoded in real-valued weights, if you interpret two real weights and use a learnable ReLU. Two variations are required to ablate this confounder:
1. complex weights with a zReLU with a fixed psi (Would this be easier to study theoretically than the existing proofs? It would also be easy to demonstrate empirically.)
2. real weights with a pseudo-zReLU with a learnable psi (Possibly hard to prove, but easy to empirically study.)
I think considering case (1) is necessary to support the claim made by the paper that complex weights are the key differentiator.

**Questions:**

- In Section 4.3, could you clarify how the outputs of the RVNN are matched to the outputs of the complex valued target function?
- In Section 5, does the trainable zReLU make the network effectively deeper? Does the convergence rate change if psi is not trainable?
- Define the schema of $L_{rr}, L_{cr},$ etc., and remind the reader every time you use them. By the time I got to Lemma 4 and Lemma 5, I did not know which result was which.
- $t4 was never defined.
- Define rvnn and cvnn in the abstract
- Add equation numbers.


**Limitations:**

A limitations section is missing from the paper.

---

> ### Author Rebuttal · Authors · 2023-08-06
>
> We thank the reviewer for the detailed feedback but would like to point out a few misunderstandings in the review.
>
> 1. About assumptions on different training regimes.
>
> In the proofs and proof sketches, training regimes come from the idea of divide and conquer and we do not make any assumption about these regimes. We analyze the learning dynamics in each regime and then unify them to obtain the global dynamics. The introduction of training regimes implies complicated and challenging analysis rather than strong assumptions.
>
> 2. About a missing section of limitations.
>
> We discuss the limitations of this paper and promising future works in the second paragraph of Section 6. This is approved by Reviewer qTS2: The authors have adequately addressed the limitations of the work.
>
> Q1: About the fundamental flaw that we do not have empirical verification of the theoretical results.
>
> A1: It seems that the reviewer might not grasp the core contributions of this paper. As far as we know, this paper theoretically studies neuron learning in the complex domain for the first time, and the obtained theorems cast light on the difference between RVNNs and CVNNs. From the theoretical aspects, all theorems are clearly written and have detailed proofs in the appendix. Thus, our theoretical results are rigorous and sound, as estimated and mentioned by all other reviewers. From the experimental aspect, we admit (as suggested by other reviewers) that empirical results can further strengthen our sound theories and verify our findings in more general settings. Thus, we provide experimental results in the **global response**, where we verify our findings in more general scenarios.
>
> Q2: Can the difference come from RVNNs using activations with a learnable parameter?
>
> A2: Thanks for this important question. In a nutshell, the difference comes from both the learnable parameter and the complex arithmetic. Firstly, the learnable parameter is related to phase learning. With a fixed $\psi$, the phase prior is determined, which may cause disappointing results when the prior is unsuitable. Thus, the learnable parameter is essential for phase learning. Secondly, complex arithmetic is the basis of phase and a natural way of modeling phase information. Complex arithmetic is the basis of phase since phase does not exist in the real domain. One may argue that $C$ is isomorphic to $R^2$ (as real vector spaces), and a complex number can be modeled by two real numbers. We admit this fact but the modeling process is highly artificial. In the words of neural networks, the multiplication from $C$ to $C$ can be modeled by a complex-valued weight (2 parameters), or a 2*2 real-valued weight matrix (4 parameters). Moreover, typical activation functions used in CVNNs can be succinctly expressed by complex numbers and their arguments, while having a more complicated formulation when expressed by real and imaginary parts. Finally, it seems that the reviewer misunderstands our conclusions. Our conclusions are about complex-valued neurons rather than complex-valued weights. We never say complex-valued weights are the key differentiator. It is important to consider both the complex-valued weights and complex activation functions.
>
> Q3: How the outputs of the RVNN are matched to those of the complex-valued target function in Section 4.3?
>
> A3: Thank you for raising this question. In the literature, it is common to take the real part as the output in the readout layer [1,2]. In this paper, we adopt this choice. The activation function $\sigma_{\psi}$ contains the operation of taking the real part, as defined in Section 3. Thus, the outputs of both RVNN and CVNN are real-valued, and there is no trouble in matching the outputs. We realize that readers may forget the definition and we will add more explanations in later versions.
>
> [1] Scott Wisdom et al. Full-capacity unitary recurrent neural networks (NIPS 2016).
>
> [2] Shao-Qun Zhang, Wei Gao, and Zhi-Hua Zhou. Towards understanding theoretical advantages of complex-reaction networks (Neural Networks, 2022).
>
> Q4: Does trainable zReLU make networks deeper?
>
> A4: It is hard to measure the relation between the trainable parameter and network depth in general since the trainable parameter may have different influences on different theoretical properties. From the approximation aspect, the approximation power roughly comes from depth, width, and activation complexity [3]. The role of the trainable parameter in zReLU belongs to activation complexity and cannot be compared with depth since the effects of depth and activation complexity are incomparable. From the learning dynamics aspect which is considered in this paper, there is no proven conclusion about the effects of activation complexity and depth. We conjecture that activation complexity and depth are incomparable since a complex-valued neuron cannot represent a two-layer real-valued neural network (which can be proved by simply counting the number of linear regions).
>
> [3] Alexis Goujon, Arian Etemadi, and Michael Unser. The Role of Depth, Width, and Activation Complexity in the Number of Linear Regions of Neural Networks (2022).
>
> Q5: Does the convergence rate change if psi is not trainable?
>
> A5: The convergence rate changes if $\psi$ is not trainable since the redundant phase consideration slows down the convergence, as concluded in Section 6. The trainable parameter is important to learn phase information. With a fixed $\psi$, the phase prior is determined, and we cannot expect a complex-valued neuron to learn phase information.
>
> Q6: About notations, abbreviations, and equation numbers.
>
> A6: Thanks for your valuable suggestions. Definitions of $L_{rr}$ and $L_{cr}$ are provided in Eq. (2) and at the beginning of section 5, respectively. We will recall the meanings of notations and update the abstract in later versions. We only add a number when the equation is cited elsewhere, which is a convention in the community of neural network theory.

---

> > ### Comment · Reviewer_Jsv8 · 2023-08-17
> >
> > Q1: Thank you for adding the experimental tests. The results provide the necessary empirical evidence for the theory and its assumptions. As such, I have raised my score to a 5.
> >
> > Q2+Q4: Thank you for the discussion on these points. My point was that the model graph for the CVNN has many differences to the RVNN that we could ablate: complex values and arithmetic, complex weights, zReLU with a learnable psi parameter, etc..  To quote your response, "In a nutshell, the difference comes from both the learnable parameter and the complex arithmetic. " Indeed -- but how do both contribute independently!?
> >
> > It is possible to perform model ablation on each of these aspects. Your conclusion states "These conclusions suggest that complex-valued neurons learn more than real-valued neurons since *CVNNs benefit from the flexibility of the phase parameter*, which helps CVNNs learn phase information more efficiently." You are specifically hypothesizing that the activation function is the crucial piece. It does definitely make sense that the psi is important for phase learning, and I think you did show it was necessary. But are the complex values (and/or weights) necessary for phase learning, if you have the equivalent graph of zReLU with a psi? Figuring out which part of the CVNN in the key component would be very interesting, and make for a much stronger paper with wider applications.
> >
> > For neural network analysis, the deconstruction into real-valued float multiplies and low-level activation functions is a more fundamental representation to study the model from. The construction/deconstruction sounds artificial only when viewed from the perspective of complex function analysis. At the lowest level, both the RVNN and CVNN are just graphs of parameter multiplies and ReLUs, and maybe the CVNNs have a sine/cosine/tangent in them. See, for example, Apicella et al., "A survey on modern trainable activation functions", 2021 for a systematic study of transforming complicated activation functions into simpler graphs. By breaking down the complex arithmetic and the trainable zReLU activation function into fundamental model graphs, the RVNN and CVNN can be directly compared in terms of depth and model complexity. If you trace out the model graph of the CVNN, I think you'll see that the psi parameter makes a directly comparable graph  that is +1 layer deeper per activation layer, has a different connectivity, (might have sin/cos/tan), and might be able express more functions. Then, it will be trivially apparent that the real-valued function approximated by the CVNN low level graph can express more real-valued functions than the RVNN. Then, it will be interesting to figure what aspect of this real-valued lower level graph causes the convergence behavior you demonstrate, which would have a broader impact.
> >
> > A potential null result is that you primarily demonstrated that this particular complicated activation function is better than just a ReLU one for the class of problems, and the complex parts have no impact. Not that I think that is the case, because I think the complex arithmetic is also an important aspect, but I don't think you rigorously ruled that out. I do however think that the trainable activation adds a lot more to model expressiveness than you are giving it credit for.
> >
> > A stronger analysis would decompose the "complex valued" networks into the fundamental model graphs of basic operations, and apply basic theoretical and empirical analysis of neural network approximation theory and convergence rates to these graphs. Then, you could breakdown which aspects of the CVNN make which changes to the graph, and apply those changes independently to determine which in particular are contributing to the change in converge rate and model expressiveness. The hypothesis to test and falsify, as suggested by the comments in the paper, is that inserting the graph+parameters of psi-activation function into the RVNN graph, by itself, would make the biggest difference in function expression and convergence.
> >
> > Q3: Makes sense. Thanks for clarifying.
> >
> > Q6: Equation numbers everywhere makes it easier for readers and reviewers to cite equations :)

---

> > > ### Author Response · Authors · 2023-08-19
> > >
> > > Thank you very much for the detailed feedback and for raising the score.
> > >
> > > Q2+Q4: There might be an ambiguity in my sentence "In a nutshell, the difference comes from both the learnable parameter and the complex arithmetic". We want to say that the learnable parameter is necessary for our results, and complex arithmetic is a natural way to implement the learnable zReLU activation function. It is possible to discard complex arithmetic if one uses real arithmetic and shared weight parameters, which is artificial from the aspect of complex function analysis.
> > >
> > > It is interesting to consider the problem from the perspective of equivalent graphs. We admit that there exists a certain RVNN that is equivalent to a CVNN, but the RVNN is highly artificially designed: there are constant weights, different activation functions in one layer, and uncommon activation functions. We still think that when people talk about RVNNs, they usually mean feedforward neural networks with real-valued learnable weight parameters and one activation function (can be learnable or unlearnable) in all layers. If you would like to refer to the CVNN in this paper, we think it is more convenient to say CVNN with learnable zReLU than describing the equivalent graph.
> > >
> > > It is promising to decompose CVNN into fundamental model graphs of basic operations and analyze these graphs.  We believe that there are many interesting problems in this direction and leave them to future work.

---

### Official Review · Reviewer_qTS2 · 2023-07-18

**Soundness:** 4 excellent
**Presentation:** 4 excellent
**Contribution:** 3 good
**Rating:** 7
**Confidence:** 3

**Summary:**

The theoretical paper compares the learnability of real-valued neurons and complex-valued neurons via gradient descent. In a nutshell, the paper proves that a complex-valued neuron has a lower convergence rate than its real-valued counterpart, but learn exponentially slower. The theoretical discoveries help explain the success of CVNNs and point out the potential slow convergence of CVNNs learning.

**Strengths:**

1. The paper makes the first attempt ever to comparing the learnability of real-valued and complex-valued neurons on a theoretical level.

2. Even though the theoretical investigations are carried out on a single neuron, the findings still cast light on the success and failure of CVNNs.

3. The discoveries are presented as Theorems and proved with rigor in the appendix, in which I did not find any technical errors.

4. The paper is very well written. The implications of each Theorem are clearly explained and summarized in tables.

**Weaknesses:**

1.  The analysis is carried out on a simple type of complex neuron network, and it is hard to generalize the achieved findings to practical RVNNs and CVNNs.

2. The discoveries could be more convincing by comparing the learning curves of RVNN and CVNN on toy examples, e.g. to show that a real neuron cannot well learn a complex neuron.

3. The research questions should be reformulated. Currently they are quite ambiguous and do not directly reflect the research focus.



**Questions:**

1. It is argued in the paper that learning one neuron can embody the learning of neural networks. However, unlike RVNNs, multi-layered CVNNs may have intermediate layers with complex-valued input and outputs, a different setup from this study. Is it possible to employ a similar approach to this paper for studying these layers?

2. The bias term is removed for technical reasons. Could you please elaborate on how the presence of bias term will influence the analysis, and whether similar findings can still be obtained?


**Limitations:**

The authors have adequately addressed the limitations of the work.

---

> ### Author Rebuttal · Authors · 2023-08-06
>
> We thank the reviewer for the feedback and comments.
>
> Q1: About generalizing to practical RVNNs and CVNNs.
>
> A1: The analysis of neural networks faces the tradeoff between simple assumptions and strong conclusions in the neural network literature. It is quite difficult to analyze practical neural networks without assumptions. Our work pursues theoretical results under simple settings and assumptions. It would be promising future work to generalize our findings to deep neural networks without complicated assumptions.
>
> Q2: About toy examples to make the discoveries more convincing.
>
> A2: Thanks for the constructive comment. We believe that our findings are well supported by our theoretical conclusions. To provide a more convincing and intuitive demonstration of our discoveries, we provide experimental results in the **global response**, where we also verify our findings in the presence of bias terms.
>
> Q3: About reformulation of the research questions.
>
> A3: Thanks for your valuable feedback. The current formulation focuses on generality and includes the setting of learning a neuron with a two-layer neural network, which is used in Theorem 3. However, this generality might not directly reflect the research focus. We will formulate the problem in a more appropriate way in later versions.
>
> Q4: About studying intermediate layers in CVNNs.
>
> A4: Neuron learning focuses on the learning dynamics of neurons, which helps us understand the learning process of neural networks since a neuron is the simplest form of a neural network. This motivation is widely adopted in real-valued neuron learning [1,2,3] and benign overfitting [4]. To analyze deep CVNNs, the complex-valued intermediate layer is not the central problem since it can be modeled by distributions related to the real and imaginary parts. The fundamental difficulty is the dependence between parameters in different layers (i.e., we have to analyze the compositions of several random variables when studying neural networks), which also occurs in RVNNs. In the literature, these compositions are analyzed under different strong assumptions or highly over-parameterization [5,6,7].
>
> [1] Yuandong Tian. An analytical formula of population gradient for two-layered ReLU network and its applications in convergence and critical point analysis (ICML 2017).
>
> [2] Gilad Yehudai and Shamir Ohad. Learning a Single Neuron with Gradient Methods (COLT 2020).
>
> [3] Weihang Xu and Simon S. Du. Over-parameterization exponentially slows down gradient descent for learning a single neuron (COLT 2023).
>
> [4] Spencer Frei, Niladri S. Chatterji, and Peter L. Bartlett. Benign Overfitting without Linearity: Neural Network Classifiers Trained by Gradient Descent for Noisy Linear Data (COLT 2022).
>
> [5] Alexandr Andoni et al. Learning Polynomials with Neural Networks (ICML 2014).
>
> [6] Zeyuan Allen-Zhu, Yuanzhi Li, and Zhao Song. A Convergence Theory for Deep Learning via Over-Parameterization (ICML 2019).
>
> [7] Kwangjun Ahn, Jingzhao Zhang, and Suvrit Sra. Understanding the Unstable Convergence of Gradient Descent (ICML 2022).
>
> Q5: About the influence of the bias term.
>
> A5: This is an important question. From the theoretical aspect, although many efforts are devoted to simplifying assumptions, the bias term is still difficult to cope with and is set as 0 in the literature [1,2,3]. The bias term will change the analytical property of the loss function and make the closed-form expression unavailable, which makes our analysis infeasible. It is possible to analyze neuron learning without closed-form losses [2], but this method relies on "spread" input distributions (Assumption 4.1). If the bias term is considered, i.e., the original input concatenated with an additional "1'', the additional "1'' makes the input distribution discrete and not spread. Thus, the analysis of the bias term is still an open problem and is beyond the scope of this paper. From the experimental aspect, we verify our findings in the presence of bias terms (Fig. 2 in the **global response**).

---

> > ### Comment · Reviewer_qTS2 · 2023-08-18
> >
> > Thanks for your response. I hold my opinion that the article should be accepted due to its theoretical strength and top-quality presentations.

---

### Author Rebuttal · Authors · 2023-08-06

We thank the reviewers for their detailed feedback. We are thankful that our theoretical results are approved by most reviewers, many important questions are raised, and many constructive suggestions are proposed. A general consensus is that additional experimental verifications can help our paper become more convincing and increase the reader pool. Thus, we provide here several experimental results to further strengthen our theoretical results and verify our findings in more general settings. Detailed responses to individual questions and suggestions are provided in separate responses.

**Experimental settings.** A training set of size 7,000 and a test set of size 3,000 are generated by a randomly initialized target neuron (can be a real-valued or a complex-valued neuron). After random initialization, a complex-valued neuron and a real-valued neuron are trained by gradient descent with the empirical mean square loss and a learning rate of 0.1 for 100 epochs (or 300 epochs when the loss does not converge).

**Experimental results.** We investigate two different settings and report the test error curves in the attached pdf. In Fig. 1, we adopt the theoretical setting with $d=1$ and without bias term. In Fig. 2, we soften the dimension condition and add the bias terms. In all settings, we find that real-valued neurons fail to learn complex-valued neurons but converge faster when learning real-valued neurons. These experimental results verify our theoretical findings in general settings.

---

### Author Response · Authors · 2023-08-17

Dear Area Chair:

We appreciate your time and consideration. We would also like to thank the reviewers for their detailed comments on our paper. Meanwhile, we have tried our best to address all questions from reviewers.

Our paper theoretically investigates neuron learning in the complex domain for the first time and finds that phase learning helps complex-valued neurons learn more but slower. Four out of five reviewers highly recognize our paper, while reviewer Jsv8 still maintains negative feedback. We believe that our rebuttals fix the doubts and misunderstandings of reviewer Jsv8.

Understanding that you are managing numerous papers and that it demands lots of effort, we are kindly reaching out to inquire if it is possible for reviewer Jsv8 to provide further feedback before the author-reviewer discussion phase ends.

Once again, thank you for your devotion to ensuring a rigorous and fair review process.

Best regards,

Authors of Submission#2826

---

### Decision · Program_Chairs · 2023-09-21

**Decision:**

Accept (poster)

**Comment:**

The authors considered the learnability of real- and complex-valued neurons via gradient descent on an architecture consisting of a single real- or complex-valued neuron.  They provided a tight guarantee for the convergence rate of a complex-valued neuron learning a real-valued neuron which, in light of prior work, shows a separation between learning with complex-valued and real-valued neurons.  They also provided the first guarantee for learning a complex-valued neuron.

The reviewers had some concerns regarding the generality/applicability of the results to deeper or more general settings.  However, as the authors noted in their rebuttals, there is by now a long line of work on learning a single neuron with gradient-based methods, and the establishment of a precise understanding of the behavior of gradient descent/flow in this setting is of interest to a number of learning theorists.  I believe the authors' experiments developed during the rebuttal phase will also help broaden the paper's reception to non-learning-theorists.  Besides this concern, the reviewers were broadly supportive of the work.

In light of this, I recommend acceptance for this work.  I would also recommend that the authors revise their introduction so as to motivate the study of learning single neurons with gradient methods as a long-standing tradition in NeurIPS and related conferences, for instance the following prior works at ML all looked at similar problems:

Auer, P., Herbster, M. and Warmuth, M. K.  Exponentially many local minima for single neurons. NeurIPS 1995

S Frei, Y Cao, Q Gu. Agnostic Learning of a Single Neuron with Gradient Descent. NeurIPS 2020.

I Diakonikolas, S Goel, S Karmalkar, AR Klivans, M Soltanolkotabi. Approximation schemes for relu regression. Conference on learning theory 2020

Ilias Diakonikolas, Daniel Kane, Lisheng Ren, Yuxin Sun.  SQ Lower Bounds for Learning Single Neurons with Massart Noise.  NeurIPS 2022